# Dynamic control of enhancer activity drives stage-specific gene expression during flower morphogenesis

Wenhao Yan [1,7], Dijun Chen [1,2,7], Julia Schumacher [1,2], Diego Durantini[2,5], Julia Engelhorn[3,6], Ming Chen[4], Cristel C. Carles [3] & Kerstin Kaufmann [1]

Enhancers are critical for developmental stage-specific gene expression, but their dynamic regulation in plants remains poorly understood. Here we compare genome-wide localization of H3K27ac, chromatin accessibility and transcriptomic changes during flower development in *Arabidopsis*. H3K27ac prevalently marks promoter-proximal regions, suggesting that H3K27ac is not a hallmark for enhancers in *Arabidopsis*. We provide computational and experimental evidence to confirm that distal DNase I hypersensitive sites are predictive of enhancers. The predicted enhancers are highly stage-specific across flower development, significantly associated with SNPs for flowering-related phenotypes, and conserved across crucifer species. Through the integration of genome-wide transcription factor (TF) binding datasets, we find that floral master regulators and stage-specific TFs are largely enriched at developmentally dynamic enhancers. Finally, we show that enhancer clusters and intronic enhancers significantly associate with stage-specific gene regulation by floral master TFs. Our study provides insights into the functional flexibility of enhancers during plant development, as well as hints to annotate plant enhancers.

[1] Department for Plant Cell and Molecular Biology, Institute for Biology, Humboldt-Universität zu Berlin, 10115 Berlin, Germany. [2] Institute for Biochemistry and Biology, University of Potsdam, 14476 Potsdam, Germany. [3] Université Grenoble Alpes (UGA), CNRS, CEA, INRA, IRIG-LPCV, 38000 Grenoble, France. 38000 Grenoble, France. [4] Department of Bioinformatics, College of Life Sciences, Zhejiang University, Hangzhou 310058, China. [5]Present address: Ernst Benary Samenzucht GmbH, Friedrich-Benary-Weg, 134346 Hann, Muenden, Germany. [6]Present address: Max Planck Institute for Plant Breeding Research, Carl-von-Linné-Weg 10, 50829 Cologne, Germany/Institute for Molecular Physiology, Heinrich-Heine-Universität, 40225 Düsseldorf, Germany. [7]These authors contributed equally: Wenhao Yan, Dijun Chen. Correspondence and requests for materials should be addressed to D.C. (email: chendijun2012@gmail.com) or to K.K. (email: kerstin.kaufmann@hu-berlin.de)

Multicellular development is controlled by precise spatiotemporal regulation of gene expression, which is largely accomplished through the activation and repression via *cis*-regulatory elements within the promoter-proximal or distal regulatory regions, such as enhancers. Promoters and enhancers are noncoding DNA sequences that can be bound by multiple regulatory proteins, especially transcription factors (TFs) to activate the expression of target genes[1]. Active promoters and enhancers are devoid of nucleosomes, thereby rendering the DNA accessible for TF binding. While promoters locate near the transcription start site (TSS), enhancers can be up to several Mbs away from their target genes[1]. A single gene can be regulated by multiple enhancers with different spatiotemporal activities, resulting in a large combinatorial complexity of expression repertoires for a given set of genes[2]. Large-scale analyses in animal genomes showed that promoters and enhancers are distinguishable based on their stereotypical DNA features and chromatin signatures[3]. For example, enhancers are often associated with mono-methylation of H3 at lysine 4 (H3K4me1) and H3 acetylation marks at lysine 9 and 27 (H3K9ac and H3K27ac)[4], while gene promoters differ from enhancers by their enrichment in H3K4me3[3]. On the other hand, there is an increasing number of reports[5,6] showing that promoters and enhancers share similar structural and functional features[1].

Enhancers have recently been subjected to intensive investigation as there is increasing evidence showing their importance not only in developmental control but also in evolution and diseases, including cancer[2]. The advent of next-generation sequencing technologies and the identification of enhancer chromatin properties enable genome-wide prediction of enhancers in a high-throughput manner[3], leading to the discovery of thousands of enhancer candidates in human[6,7] and other animal genomes[8]. Enhancers appear to function as integrated platforms for binding of multiple TFs, often including lineage- and signal-determining (intrinsic or extrinsic cue-dependent signal) TFs[2,9]. Remarkably, cell identity gene loci are frequently associated with dense clusters of enhancers or super-enhancers with both overlapping and distinct spatiotemporal activities[10,11]. Genome-wide analyses of enhancer functions have revealed that enhancers are functionally complex and developmentally dynamic during cell differentiation[12]. Lineage-regulating factors, including pioneer factors and their associated TF networks, contribute to shaping enhancer repertoires during development[12–14].

Contrary to the fast growing knowledge of the regulatory function of enhancers in metazoans, knowledge of genome-wide landscapes of plant enhancers and their dynamic activities remains limited in the context of developmental stage-specific gene regulation[15]. Enhancers are predominantly found in regions of accessible chromatin[3], which is more sensitive to DNase I digestion than condensed chromatin. Genome-wide mapping of open chromatin regions has been conducted in the selected model and crop plant species[16–21]. In 2012, Zhang et al. reported the mapping of open chromatin in rice seedlings and callus using DNase-seq. They found that DNase I hypersensitivity sites (DHSs) are associated with highly expressed genes in both tissues, and that 42.3% and 44.5% of DHSs in seedlings and in calli, respectively, reside in intergenic regions[16]. The same group later identified a high overlap between TF occupancy and DHSs in *Arabidopsis thaliana* leaves and flowers and they found that around 90% of APETALA1 (AP1) and SEPALLATA3 (SEP3) binding sites overlap with DHSs[17]. Using these above-mentioned open chromatin signatures, they further carried out a genome-wide prediction and validation of *Arabidopsis thaliana* intergenic enhancers[20]. A total of 10,044 intergenic DHSs were identified as putative enhancers, of which 1644 and 2529 are leaf- and flower-tissue specific, respectively. Enrichment of both H3K27ac and

H3K27me3 was reported in the genomic regions of predicted tissue-specific enhancers, with a positive correlation for H3K27ac and a negative correlation with H3K27me3[20]. More recently, by combining the information of genome-wide DNA methylation, chromatin accessibility, and H3K9ac profiling, 1500 intergenic enhancers were predicted in maize[21]. Although these studies spark the renewed interest in the investigation of plant enhancer elements, genome-wide mechanisms of dynamic enhancer activities to trigger spatiotemporal gene expression in plant development have not yet been elucidated.

Here, we used flower development as an ideal system to study enhancer dynamics and to dissect their roles in the control of stage-specific gene expression patterns by integrative analysis of genome-wide H3K27ac locations (ChIP-seq), chromatin accessibility (DNase-seq)[18], and transcriptome dynamics (RNA-seq) in *Arabidopsis thaliana* at four representative stages. We found that H3K27ac prevalently marks promoters but rarely localizes in distal regulatory regions, suggesting that H3K27ac is not a hallmark for active enhancers in *Arabidopsis thaliana* as it is in metazoans[22], and as data from rice and maize indicate[23,24]. Examination of DHS-predicted distal intergenic enhancers reveals their high-stage specificity across flower development, their functional importance in the regulation of flowering genes, as well as their high evolutionary conservation across crucifer and dicot species.

## Results

**Distribution of H3K27ac and DHSs during flower development.** H3K27ac is a primary epigenetic mark of active enhancers in animals[1,3,22], and it was recently shown to highly correlate with active gene expression in plants[23–25]. Here, we further investigated the role of H3K27ac dynamics in stage-specific gene regulation. We generated a genomic atlas of H3K27ac modification across flower development in the model plant species *Arabidopsis thaliana*. We took advantage of a previously described floral induction system, which allows collection of synchronized flower tissue of specific developmental stages[26] and that proved to be highly suitable for depicting quantitative changes in chromatin marks, accessibility, and TF binding[18,27] (Fig. 1a). The inflorescence meristem (before induction, stage 0) as well as floral tissues at stages of meristem specification (2 days after induction (DAI), stage 2), floral whorl specification (4 DAI, stages 4–5), and organ differentiation (8 DAI, stages 7–8) were harvested for ChIP-seq to map H3K27ac modification patterns (Fig. 1b; Supplementary Fig. 1). A set of controls confirmed the high quality of the ChIP-seq data (Supplementary Fig. 2). Time-series transcriptomics (RNA-seq) and genome-wide open chromatin profiles[18] generated on the same developmental series were integrated into the analysis (Fig. 1b). Comparative analyses show that both H3K27ac and DNase I accessibility highly positively correlate with the expression of nearby genes at all four developmental stages (Supplementary Fig. 3), and highly expressed genes contained both H3K27ac sites and DHSs around their TSSs (Supplementary Fig. 4). For example, dynamic changes of H3K27ac and DHS profiles at the key floral regulatory genes APETALA1 (AP1, a floral meristem identity gene) and APETALA3 (AP3, a floral organ identity gene) correlate with their expression dynamics. The AP1 locus is enriched in H3K27ac and stays open across all four time points, in line with its steady high expression level (presumably at the transgene and/or endogenous AP1 loci that are both present in the used plant line). In contrast, the AP3 locus shows increase in H3K27ac and accessibility at later stages (4 and 8 DAI), which correlates with an increase in expression (Fig. 1b). Generally, genes with higher levels of expression displayed more elevated H3K27ac signals and higher chromatin accessibility

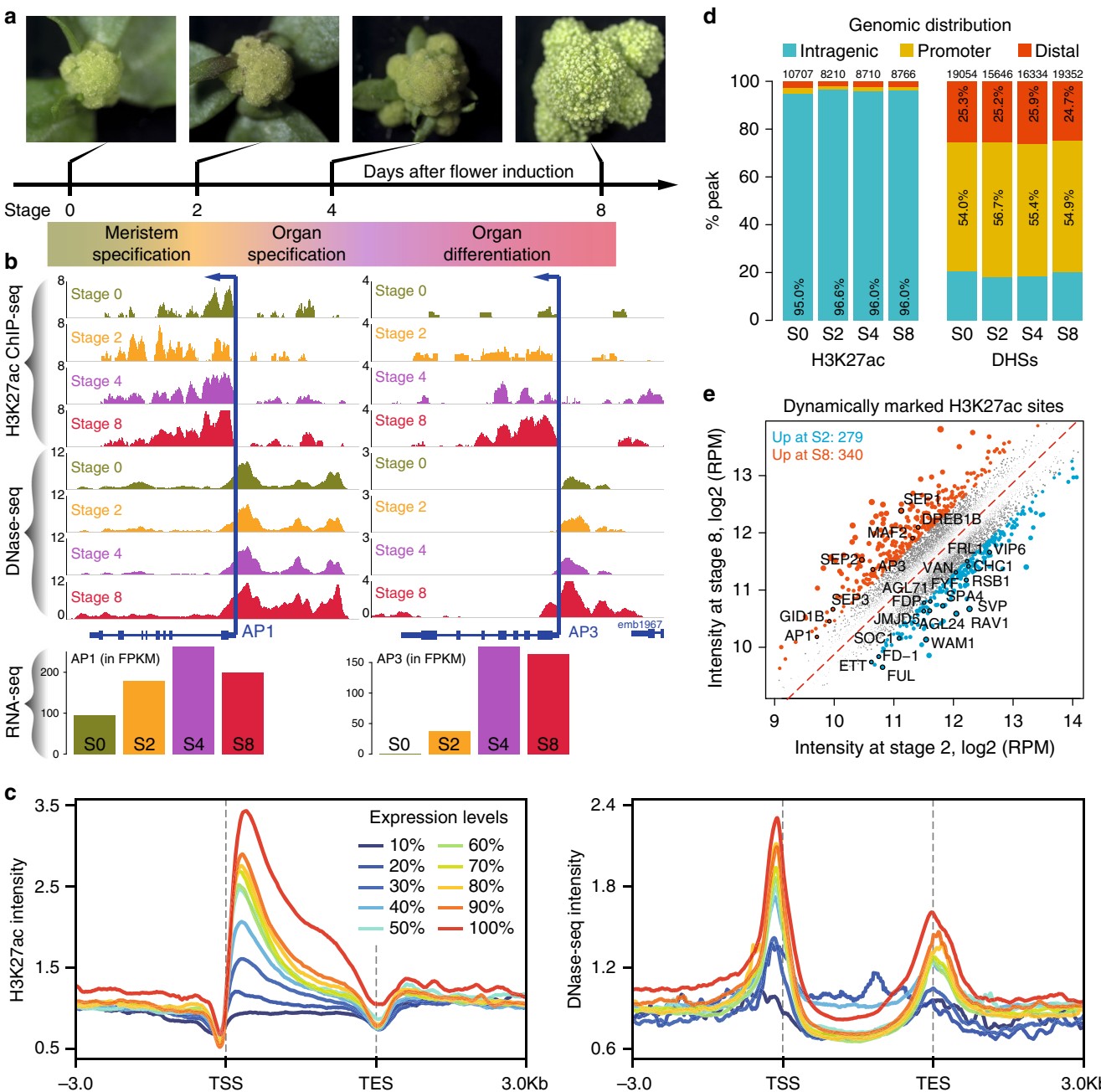

**Fig. 1** Genome-wide mapping of DHSs and H3K27ac sites across flower development. **a** Schematic representation of the floral induction system (pAP1::AP1-GR *ap1 cal*) for collecting synchronized flower tissues at four key developmental stages (stage 0, stage 2, stage 4, and stage 7/8). **b** Genome browser tracks (colored according to stages) showing H3K27ac binding signal (measured by ChIP-seq) and chromatin accessibility (by DNase-seq) around the *AP1* (left) and *AP3* (right) loci. The expression levels of these two genes measured by RNA-seq are shown in the barcharts below. FPKM, fragments per kilobase of transcript per million mapped reads. S0, stage 0. **c** The relationship between H3K27ac binding signals (left) or chromatin accessibility (right) and the levels of gene expression at stage 8. In the analysis, the expressed genes were equally divided into 10 groups (*n* = 2083 genes in each group) from low expression (10%) to high expression (100%) based on the expression levels. Mean-normalized ChIP-seq or DNase-seq densities were plotted within a 6-kb region flanking the transcription start site (TSS) or the transcription end site (TES) for each groups of genes. **d** Stacked bar plot showing the distribution of H3K27ac peaks and DNase I hypersensitive sites (DHSs) in different genomic regions at 0, 2, 4, and 8 DAI. Promoters are defined as genomic regions encompassing 1 kb upstream of the TSS. Distal intergenic regions are genomic regions except intragenic and promoter regions. Peaks were assigned to each category based on their summits. **e** Comparison of H3K27ac enrichment at stages 2 and 8 of flower development. Each point represents an H3K27ac enriched region. Regions showing more than 1.5-fold difference of enrichment are indicated in blue (more enrichment at S2) and red (more enrichment at S8). Numbers indicate the number of target genes. Examples of target genes (with black circles) are labeled. The underlying source data for **d** and **e** are provided in the Source Data file

within regions surrounding the TSS (Fig. 1c), which is probably related to the putative function of H3K27ac in transcript elongation, as has been seen in other systems[28]. These observations confirmed that H3K27ac enrichment correlates with gene activity in plants.

We observed that most of the H3K27ac sites are located in intragenic regions and directly downstream of the TSS (Fig. 1c, d —left panels; Supplementary Fig. 5a, b). In detail, out of all the high-quality H3K27ac peaks identified at different developmental stages (Supplementary Data 1), more than 95% were found in gene bodies, whereas only about 2.5% were mapped to distal intergenic regions (> 1 kb from the TSS; Fig. 1d—left panel). In contrast, the majority (~80%) of DNase I hypersensitivity sites (DHS) were found in intergenic regions and they clearly resided upstream of the TSS (Fig. 1c, d—right panels; Supplementary Fig. 5c, d). Interestingly, DNA-binding sites of floral regulatory TFs highly overlap with DHSs, but not with H3K27ac sites (Supplementary Fig. 6). However, the distance of binding sites to their closest H3K27ac peaks varies for different regulators (Supplementary Fig. 6a). Given that H3K27ac mainly marks TSS-proximal regions, the above observation indicates that different TFs have different binding preferences in genomic locations with respect to the TSS. Furthermore, we found that although most H3K27ac sites remained stably enriched during flower development (Supplementary Fig. 7), 23.2% (2913/12579) of them displayed at least a 1.5-fold change in H3K27ac peak intensity among the four time points. These 2913 sites were classified into four clusters based on their dynamic behavior (Supplementary Fig. 8a–c, Supplementary Data 2). Generally, the dynamics of H3K27ac and changes in gene expression showed a positive correlation in each cluster (Supplementary Fig. 8c). The occasionally observed differences between changes in H3K27ac and in gene activities (e.g., for genes in groups 2 and 4; Supplementary Fig. 8c) can be partly explained by the fact that the data were not generated from individual cells or cell types, but correspond to "average" signals from whole stage-specific meristems, or organ differentiation stages. For example, strong gene activation in some of the cells may be associated with an overall reduction in H3K27ac levels, if in most other cells, the gene is not activated or becomes repressed. Gene ontology (GO) enrichment analysis revealed that genes with dynamic H3K27ac activities were highly enriched in functions, such as "regulation of gene expression" and "flower development"-related processes (Supplementary Fig. 8d). We further compared H3K27ac ChIP-seq signal intensities between 2 and 8 DAI, two representative time points for early and late flower development (Fig. 1e). There are 279 genes showing elevated H3K27ac levels at 2 DAI, including several known MADS-box genes, such as *FRUITFULL* (*FUL*), *SHORT VEGETATIVE PHASE* (*SVP*), *SUPPRESSOR OF OVEREXPRESSION OF CONSTANS1* (*SOC1*), and *AGAMOUS-LIKE 24* (*AGL24*), which are important for floral transition and initiation[29]. Another set of 340 genes showed higher H3K27ac signals and increased expression levels at 8 DAI, including floral organ identity genes *AP3* and *SEPALLATA* (*SEP1*, *SEP2*, and *SEP3*). In summary, we found that H3K27ac is highly associated with gene activity during flower development. Together, our results support and refine previous research[20], showing that H3K27ac is highly correlated with gene activation during floral meristem and organ development. In addition, the difference in localization of H3K27ac sites and DHSs with respect to the TSS in distal intergenic regions indicates that they may activate gene expression via distinct mechanisms.

**H3K27ac is not a hallmark of enhancers in Arabidopsis.** Recent work suggested that H3K27ac colocalizes with distal intergenic

DHSs in *Arabidopsis thaliana*, indicating that this mark associates with enhancer elements[20]. However, we identified very few distal intergenic H3K27ac peaks (Supplementary Data 1). This difference with the previous study led us to re-examine the use of H3K27ac as a mark to map distal regulatory regions (such as enhancers) in *Arabidopsis thaliana*. We found that <3% of H3K27ac sites mapped outside core promoters (within 1 kb upstream of the TSS) and transcribed regions (with a peak summit >1 kb away from the TSS site; Fig. 1d). This observation is in line with the percentage reported in seedlings[25]. In contrast, ca. 25% of the DHSs were found in distal intergenic regions. Consistently, the center of most (>95%) H3K27ac peaks are present in intragenic regions (Fig. 1d), and these intragenic H3K27ac sites locate at ~300 bp (median value) downstream of the TSS (Fig. 2a). The centers of intragenic DHSs preferentially locate around 1 kb away from the TSS (Fig. 2b), while the centers of intergenic DHSs locate in both promoter-proximal (~120 bp upstream of the TSS, median value) and distal regions (~1.6 kb upstream of the TSS, median value; Fig. 2c), respectively. We further determined the distance between DHSs and H3K27ac peaks. Interestingly, we found that nearly all the H3K27ac sites resided within 1 kb from a nearby DHS in all four stages (Supplementary Fig. 9a). In contrast, a considerable proportion of DHSs appeared to be isolated more than 1 kb apart from H3K27ac sites (Supplementary Fig. 9b, c). These findings were further supported by the observation that the flanks of proximal DHSs were largely enriched in H3K27ac modification, while the flanks of distal intergenic DHSs showed no enrichment in H3K27ac (Supplementary Fig. 10). These observations are contradictory to the former report from Zhang et al., where the authors found H3K27ac enrichment surrounding enhancer regions using data generated from leaf and mixed flower tissues in *Arabidopsis thaliana*[20]. We thus reanalyzed these previously published H3K27ac and DNase-seq data and reached conclusions similar to those we obtained from the flower developmental time-series data (Supplementary Fig. 11), indicating that the discrepancy is not due to different tissues used or datasets generated. The differences between our report and that of Zhang et al. likely result from the differences in the sizes of regions chosen for the analysis (5 kb in Zhang et al.[20] vs. 1 kb in our study). 5 kb enhancer flanking regions may overlap with the H3K27ac signal from distal sites, because the *Arabidopsis thaliana* genome is very compact with an average size of ~3 kb for intergenic regions. In summary, these results indicate that, at least in *Arabidopsis thaliana*, H3K27ac is not a suitable mark to predict active distal regulatory elements (including enhancers), because the majority of H3K27ac sites appear in intragenic or promoter regions and hardly coexist with DHSs in distal intergenic regions.

To further investigate whether other histone modifications are hallmarks of enhancers annotated by distal DHSs, we extended our analyses by using a large catalog of histone modifications, including H3K27ac, H3K4me1, H3K4me2, H3K4me3, H3K9ac, H3K9me2, H3K27me3, and H3K36me3, for which datasets were generated from *Arabidopsis thaliana* seedlings[20,30] (Fig. 2d). We believe that the general distribution patterns of DHSs with respect to histone marks are comparable among different tissues, since we observed similar distribution patterns of H3K27ac with respect to DHSs between seedlings and flowers (Supplementary Figs. 9, 11). In the analysis, enrichment of each histone modification was plotted relative to the peak centers of proximal and distal DHSs, respectively. Moreover, to avoid bias due to the strategy used in our analysis and to compare our findings in different species, we applied the same methodology to matched datasets in rice (*Oryza sativa*) seedlings[16,23,31] (Fig. 2e) and in the human (*Homosapiens*) K562 cell line[6] (Fig. 2f). We found that H3K27ac, H3K4me3, and H3K9ac, known features for promoter

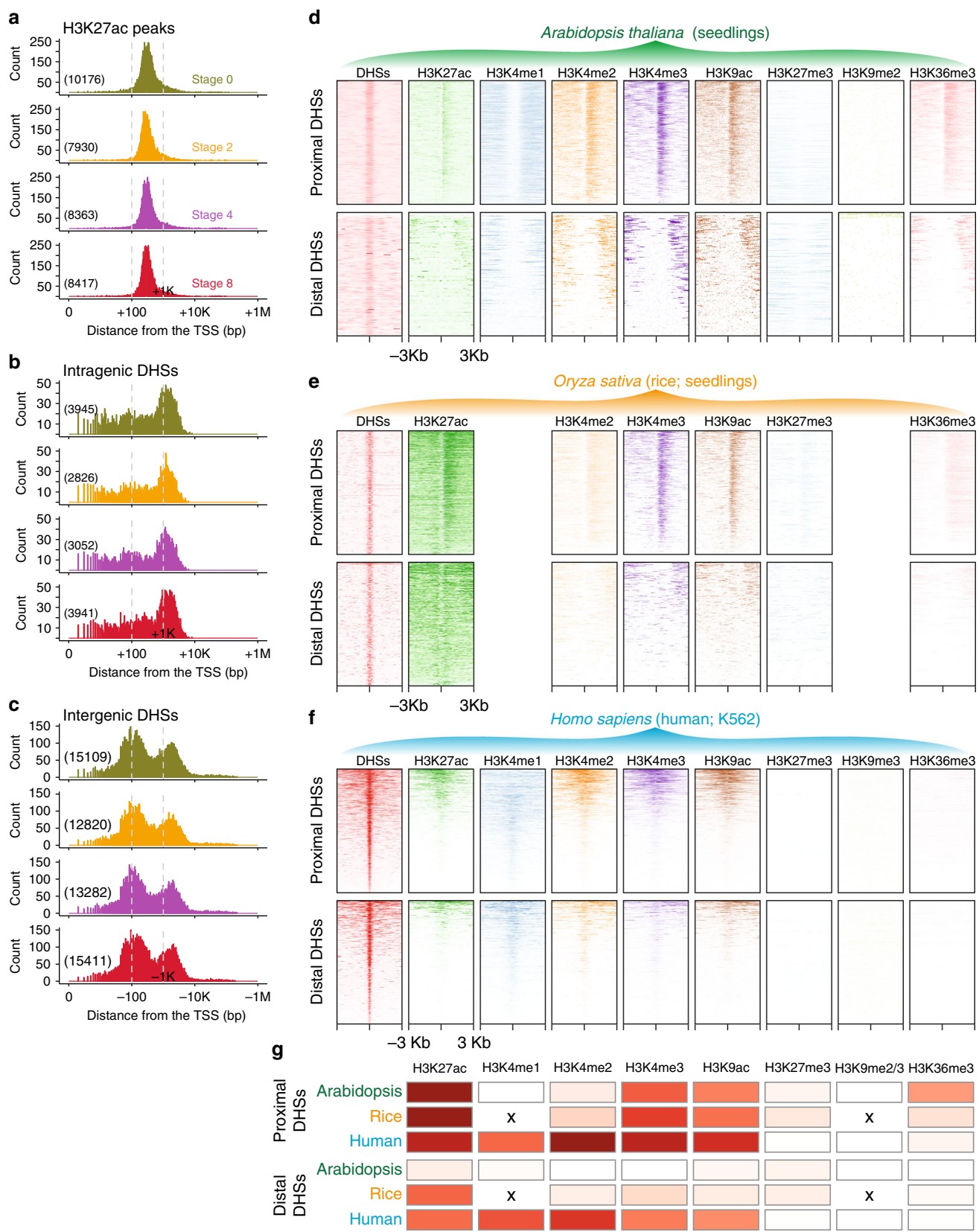

annotation[3,32], are conserved marks to predict proximal DHSs in all the three investigated species (Fig. 2d–g; Supplementary Fig. 12). The distal DHSs were largely occupied by surrounding H3K27ac in human and in rice genomes. However, the *Arabidopsis thaliana* distal DHSs barely exhibited H3K27ac

modification in their flanks. Strikingly, none of the investigated histone modifications coexisted with distal DHSs in our study, and thus none of them could be considered as chromatin signatures to predict enhancers in *Arabidopsis thaliana* (Fig. 2g). Taken together, unlike the findings in humans and rice, our

**Fig. 2** H3K27ac preferentially appears in promoter regions but not in enhancers in Arabidopsis. **a** Distribution of the distance of H3K27ac peak summits from the transcription start site (TSS). Only intragenic H3K27ac peaks were used in the analysis. **b, c** Distribution of the distance of the center of DNase I hypersensitive sites (DHSs) from the TSS. Intragenic DHSs are downstream of the TSS (**b**), while intergenic DHSs are upstream of the TSS (**c**). **a–c** The number of H3K27ac peaks or DHSs is shown in parentheses. Note that the distance for intragenic peaks is presented as positive (+) values, while the distance for intergenic peaks is shown as minus (−). The x axis is log10-transformed. **d–f** Various histone modification ChIP intensities relative to the peak centers of proximal and distal DHSs. ChIP-seq and DNase-seq data were generated from Arabidopsis seedlings (**d**), rice seedlings (**e**), and human K562 cell line (**f**). Note that there is no H3K36me3 enrichment at the gene body at the proximal DHSs in the human cell line. This is in contrast to other reports in mammals (e.g., ref. [81]), possibly due to differences in H3K36me3 antibody quality. **g** Relative enrichment of selected histone modifications in proximal and distal DHSs, respectively. The heatmap shows the Jaccard statistic between the intervals of DHSs and histone-modified sites. "x" denotes the missing matched histone modification datasets in rice. High values are shown in red and low values in white. The underlying source data for **a**, **b**, and **g** are provided in the Source Data file

results imply that H3K27ac is not a hallmark for enhancer annotation in *Arabidopsis thaliana*.

**Distal DHS sites are predictive of enhancers**. Given that H3K27ac does not significantly mark active enhancers in *Arabidopsis thaliana*, we set out to investigate whether distal DHS dynamics possess features of enhancers in our dataset. Based on the observation that TSS-proximal DHSs that coexist with H3K27ac peaks (within 1 kb from the TSS) mostly locate within 300 bp from the TSS (Fig. 2a, c; Supplementary Fig. 9), we focused on intergenic DHSs whose peaks center further than 1 kb away from the TSS. Interestingly, we found that genes with both distal intergenic and proximal DHSs showed significantly higher expression levels than those with either proximal or distal DHSs alone (Welch's *t* test *p*-value < 0.05; Fig. 3a), implying that these distal intergenic DHSs show the bona fide property of enhancers. The above observations led us to refer to distal intergenic DHSs as candidate enhancers and other TSS-proximal DHSs as promoters. Recent Hi-C-based analyses of chromatin loops revealed that the chromatin conformation in *Arabidopsis thaliana* is dominantly represented by small interactive regions (kb-sized)[33,34], which indicates that enhancers mostly target their proximal gene(s) in Arabidopsis. Based on this assumption and the fact that most of distal DHSs located less than 3 kb from the TSS of a neighboring gene (Supplementary Fig. 9d), we associated enhancers with their potential target genes using the "nearest neighbor" strategy[12], so that enhancers can be located upstream or downstream of a gene in our subsequent analyses. The time- course analysis identified 4844 putative enhancers in total, of which 3513, 2976, 3253, and 3223 enhancers were predicted for the four successive flower developmental stages, respectively (Fig. 3b; Supplementary Data 3). Interestingly, nearly two-thirds (~64%) of the predicted enhancers are upstream of their presumed target genes, while one-third of enhancers locate downstream of their target genes based on the "nearest neighbor" hypothesis (Supplementary Data 3). Computational motif mining from a collection of 763 distinct TFs (Supplementary Table 1) revealed that all predicted enhancers contained at least 10 potential TF-binding sites (Supplementary Data 3). The enrichment of predicted TF-binding sites in putative enhancers was significantly higher than in flanking regions or random intergenic non-DHS regions (Welch's *t* test *p*-value < 2.2e−16), but the enrichment is slightly lower than that in promoter regions (Welch's *t* test *p*-value < 0.002; Supplementary Fig. 13), indicating that these enhancers are subject to regulation by multiple TFs. Notably, enhancer DHSs tended to be more stage-specific (i.e., only detected at a specific stage) than promoters at all four time points after floral induction (Fig. 3c). This indicates that enhancers are more associated with stage-specific gene regulation than promoters[35,36]. To further dissect the functional importance of enhancers, we first investigated the enrichment of significantly associated SNPs occurring in

promoters defined by proximal DHSs and enhancers predicted by distal DHSs, respectively. These SNPs were associated with flowering-related phenotypes by a genome-wide association study (GWAS)[37]. We observed that enhancers showed a higher enrichment of significantly associated SNPs by GWAS than promoters (*p*-value < 0.005 by Welch's *t* test; Fig. 3d). We then examined sequence conservation between Arabidopsis and other crucifer species in promoters and enhancers based on the PhastCons conservation score[38]. We found that enhancer regions were more conserved than their flanking regions, a characteristic of regions under the evolutionary constraint (Fig. 3e). Unexpectedly, in contrast to the findings in humans (Fig. 3f), we observed that *Arabidopsis thaliana* promoters identified in this study were less conserved than their surrounding regions (Fig. 3e), due to the common existence of protein-coding genes or natural antisense transcripts[39] in promoter-surrounding regions. Overall, promoters seemed less conserved than enhancers between *Arabidopsis thaliana* and other crucifer species, although they both showed significantly higher conservation than intergenic non-DHS regions. The above results are further supported by the observation that *Arabidopsis thaliana* enhancers are more strongly enriched in evolutionarily conserved noncoding sequences (CNSs)[38,40] than promoters (Fig. 3g, h). Moreover, distal DHSs were found to overlap more strongly with long noncoding RNAs[41,42] (Fig. 3i), which might represent eRNAs (enhancer RNAs) transcribed by enhancers. In summary, our analyses highlight the efficacy of predicting *Arabidopsis thaliana* enhancers by distal DHSs. Moreover, the conservation of predicted enhancers across species indicates that enhancers may possess conserved putative functional features.

**Validation of the predicted enhancers**. To validate the predicted flower-related enhancers, we applied a well-established β-glucuronidase (GUS) reporter assay[20] with minor modifications (see the Methods section). Thirty reporter lines were produced, corresponding to 24 enhancer candidates with average fragment length of 1.07 kb neighboring flowering-related genes. Six of the enhancer candidates were verified using two independent constructs with opposite enhancer fragment orientation (Fig. 4a; Supplementary Data 4). Three candidate enhancers locating at the 3′ regions of genes (including *JAGGED* [*JAG*], *FLOWERING LOCUS T* [*FT*], and a shared region by *REDUCED SHOOT BRANCHING 1* [*RSB1*] and *SOC1*) were also chosen for validation (Supplementary Data 4). The minimal 35S basal promoter element alone and the enhancer of the 35S promoter (−200 to −39 bp) were taken to serve as negative and positive controls, respectively (Fig. 4b). To validate the enhancers' ability to increase expression, independent transgenic lines were scored for the presence of the GUS signal in floral tissues. In addition, we scored the GUS pattern to assess a potential correlation between enhancer activities and expression patterns of the proximal genes.

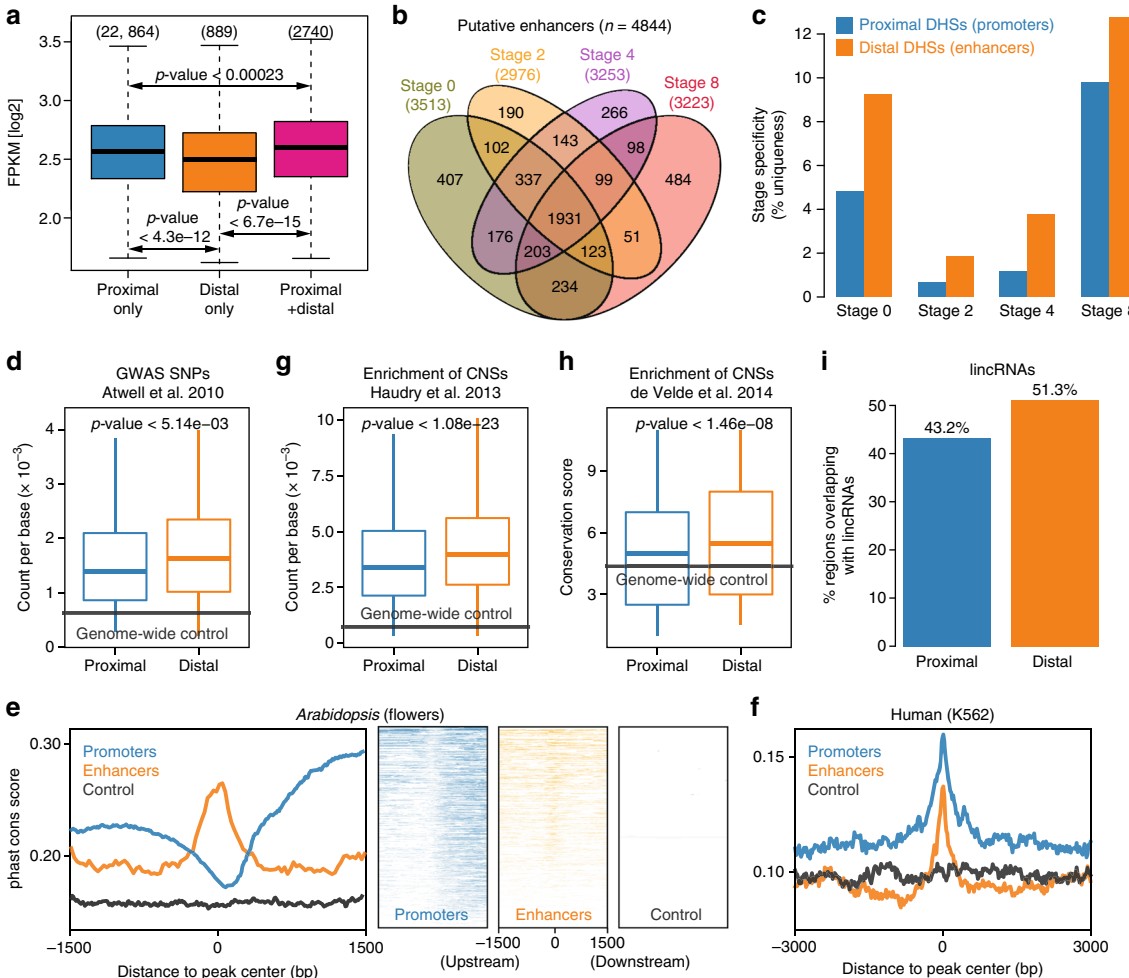

**Fig. 3** Identifying and characterizing intergenic DHS-based enhancers. **a** Boxplots of gene expression for H3K27ac-marked genes associated with only proximal DNase I hypersensitive sites (DHSs), with only distal DHSs, or with both. *P*-values were calculated based on Wilcoxon signed-rank tests. The number of genes in each category is shown in parentheses. **b** Venn diagram representing the overlap of putative enhancers identified in different stages. **c** Bar chart showing the stage specificity, i.e., the called DHSs only existed at one specific stage, of promoters and distal enhancers. **d** Enrichment of significant associations for flowering-related phenotypes from a genome-wide association study (GWAS)[37]. The *y* axis shows the number of significant SNPs per basepair. **e** Distribution of phastCons score around the peak centers of promoter (blue) and enhancer (orange) DHSs in Arabidopsis. The distribution of phastCons score for promoter and enhancer DHSs was plotted in a strand-specific manner, using the orientation of their target genes. As a control, 10,000 random intergenic non-DHS regions were used to calculate the distribution of phastCons score (gray). PhastCons scores between Arabidopsis and other crucifers were taken from ref. [38] for Arabidopsis (nine-way multiple alignment). Left panel, composite (average intensity) plots; right panel, heatmaps of conservation distribution per DHS. **f** Similar to **e** (left panel), conversation analysis of human promoter and enhancer DHSs in the K562 cell line. PhastCons scores for humans were taken from the UCSC genome browser (100-way multiple alignment). **g** Enrichment of conserved noncoding sequences (CNSs) in promoters and enhancers. The y axis shows the number of CNSs per basepair. Data from ref. [38]. **h** Similar to **g**, the *y* axis shows the conservation score. Data from ref. [40]. **a, d, g, h** Boxplots show the median (horizontal line), second to third quartiles (box), and Tukey-style whiskers (beyond the box). **d–h** Genome-wide average values were used for control (gray lines). *P*-values measure statistical significance based on a Welch's t test. **i** Percentage of promoters and enhancers overlapping with lincRNA loci. Annotation of lincRNAs were obtained from refs. [41, 42]. The underlying source data for **a**, **c**, **d**, and **g–i** are provided in the Source Data file

As expected, the GUS signal was absent in the negative control, while it was very strong in most floral organs of the positive control (Fig. 4a, b). In total, 27 of 30 (90%) reporter lines, corresponding to 22 candidate enhancers showed clear GUS signals in floral organs (Fig. 4a), indicating high reliability of the enhancer prediction. Overall, the resulting GUS signals are positively correlated (Pearson's correlation coefficient $r = 0.20$, *p*-value < 0.01) with expression profiles of the corresponding genes in the tested tissues (Fig. 4c). Five of the six enhancers, each of which was validated by two independent constructs in which the enhancer fragment was either in forward or reverse orientation showed highly similar signal patterns for both orientations (Fig. 4a), which is in line with the idea that enhancer function is

independent of orientation. Further, the GUS patterns of predicted enhancers largely correlated with the function of their proximal genes. For instance, the enhancer of the *ETTIN* (*ETT*) gene, a key regulator of gynoecium development[43], gave specific GUS signals in the tip of the gynoecium (Fig. 4d), while the enhancer of the *BLADE ON PETIOLE2* (*BOP2*) gene, which contributes to floral organ patterning along the proximal–distal axis[44], showed strong signals in the basal part of flowers as well as the junction between flowers and stems (Fig. 4e). Similar GUS patterns, but with much weaker signals, could also be observed for an enhancer element of *BOP1*, a close paralog of *BOP2* (Fig. 4f). Clear (albeit weak) GUS signals were detected in differentiating sepal organs (including receptacle) for the predicted

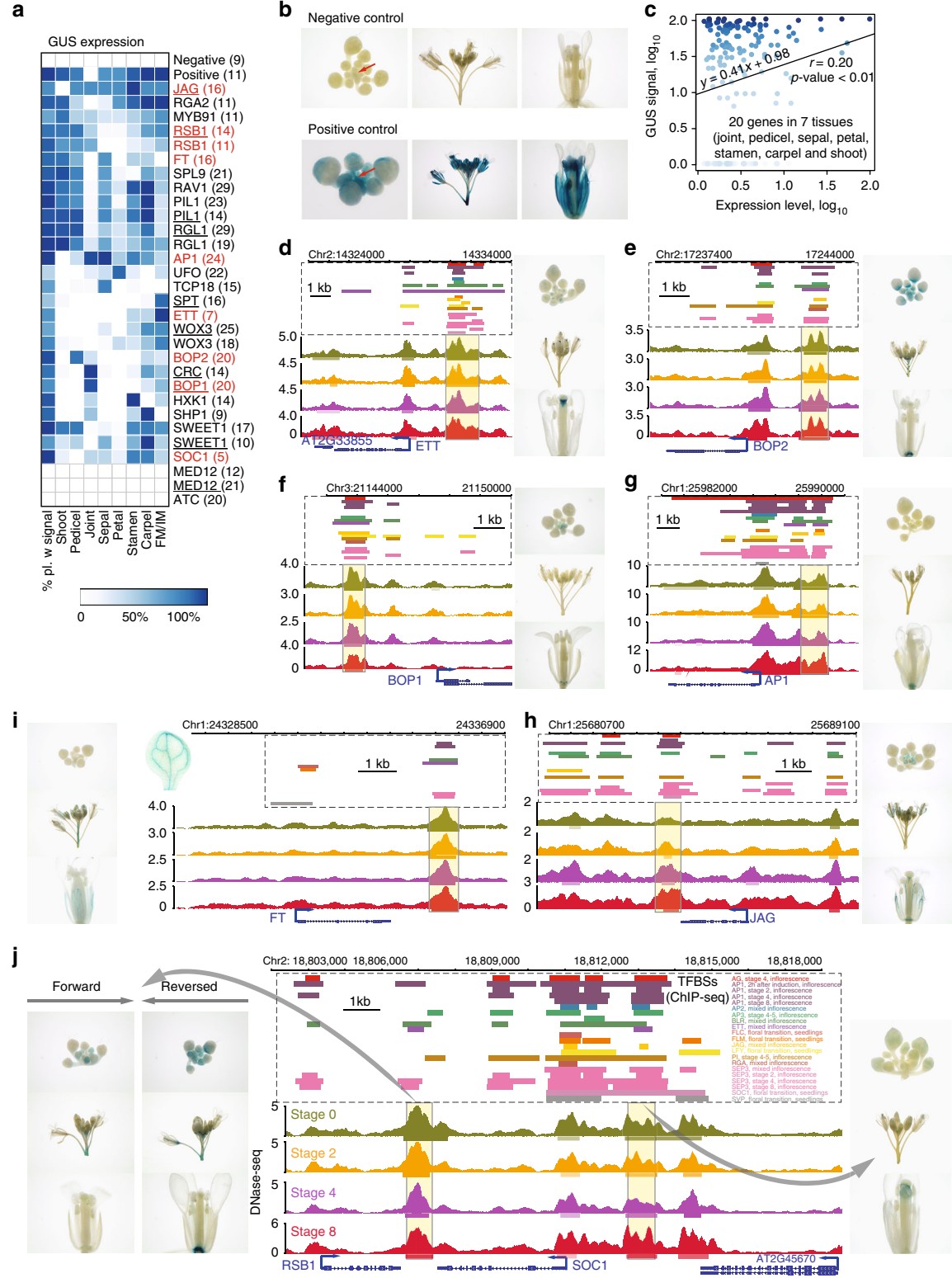

 enhancer, with the presence of GUS signal in the inner whorl organs, such as stamens and carpels (Fig. 4g). Interestingly, three of the chosen enhancers located in the 3′ distal regions of proximal genes also generated GUS signals (Fig. 4h–j). The predicted *JAG* enhancer triggered GUS expression in early floral stages (Fig. 4h). No GUS signal was observed for the predicted *FT* enhancer in early flower organs; GUS signals were instead detected in the vasculature of inflorescence stems and leaves

(Fig. 4i). This observation is consistent with the function of *FT* as a florigen[45]. Another tested locus corresponding to the 3′ region shared by *RSB1* and *SOC1*, for which GUS reporters with either orientation showed clear signals (Fig. 4j). In addition, a fragment covered by a distal DHS peak that locates upstream of the *SOC1* gene also exhibited a detectable GUS signal (Fig. 4j). These results highlight the regulatory function of enhancer elements in intergenic regions, upstream or downstream of genes. Taken together,

**Fig. 4** Functional validation of predicted enhancers in flower development via GUS reporter assay. **a** Heatmap showing GUS signals in different flower organs (columns) of T1 transgenic lines transformed with different enhancer constructs (rows). White indicates that no GUS expression was detected in any plants. Blue color intensity refers to the percentage of transgenic plants with detectable GUS expression. Names of genes that are proximal to the candidate enhancers are labeled on the right side and the numbers in parentheses represent the number of independent transgenic lines generated from the construct. Red, examples shown in **d**–**j**. Bold, the corresponding enhancer was validated with two independent constructs; underlined, construct that contains a reverse-orientated enhancer fragment. **b** Pictures of a dissected inflorescence, including a central inflorescence meristem (red arrow) and the surrounding young floral buds, of a whole inflorescence, and of a mature flower (from left to right, respectively) from plants harboring mimial35S as a negative control (left panel) or an enhancer element for a 35S promoter as a positive control (right panel). **c** Pearson correlation (*r*) between GUS signals and expression profiles of the corresponding genes in the tested tissues. Tissue-specific gene expression data can be found in Supplementary Data 4. Data points were colored according to the GUS signals in **a**. **d**–**j** Genome browser view of DHS signal profiles as well as called DHS peaks (horizontal bars below the signal profile), and transcription factor binding sites (TFBSs; data from ref. [48]) at the genomic region of predicted enhancers (in yellow-background boxes). Pictures of a dissected inflorescence, including a central inflorescence meristem and the surrounding young floral buds, of a whole inflorescence, and of a mature flower (from top to bottom, respectively) on the left or right of the genome browser view represent transgenic lines harboring the corresponding enhancer construct. The proximal genes of selected enhancers showed here are *ETT* (**d**), *BOP2* (**e**), *BOP1* (**f**), *AP1* (**g**), *JAG* (**h**), *FT* (**i**), and *RSB1* and *SOC1* (**j**). Color legends for the tracks are indicated in **j**. Note that the validated enhancers for genes *JAG* (**h**), *FT* (**i**), and *RSB1* or *SOC1* (**j**) are located downstream of the genes. The underlying source data for **a** and **c** are provided in the Source Data file

the GUS reporter assay confirmed functionality, i.e., the ability to drive expression, for the majority of the analyzed enhancers emphasizing the strengths of the enhancer prediction using distal DHSs. These data further indicate that distal enhancers can be located upstream or downstream of a gene, and that their activity is not linked to their orientation. In addition, the analyzed GUS patterns were highly consistent among the independent transgenic lines of each enhancer construct, providing evidence that enhancers drive expression in a highly organ-specific manner. Observed deviations of GUS patterns from previously reported expression patterns of proximal genes can likely be attributed to the necessity of further regulatory elements, such as the core promoter and other enhancers.

**Pervasive binding of floral regulators in enhancers**. When analyzing TF-binding patterns in promoters and enhancers, we observed that enhancers tend to harbor more binding sites for floral regulators than their promoter counterparts (Fig. 4d–j). This inspired us to investigate whether this observation is a general phenomenon for all mapped enhancers. We thus focused on 1979 genes that contained both DHS-predicted enhancers and promoters and counted the frequency of binding peaks for 15 floral regulators, corresponding to 21 ChIP-seq datasets (see the Methods section; Fig. 5a—upper panel; Supplementary Data 5). We observed that enhancers contained on average 3.1 TF-binding sites, which is significantly higher than 2.3 in promoters (Fig. 5a —bottom panel; paired Wilcoxon test, *p*-value < 2.8e–16). Accordingly, nearly all the investigated TFs preferentially bind to enhancer regions (Fig. 5b), although master regulators of development and morphogenesis such as AP1 (at the early stage), SEP3 (at late stages), ETT, and BLR also highly bind to promoters. To test whether the binding of master regulators in predicted enhancers contributes to expression dynamics, we compared the populations of genes having both proximal and distal intergenic DHSs with the genes that were differently expressed during flower development, and observed a significant (hypergeometric test, *p*-value < 1.92e–64) overlap between these two sets of genes (Fig. 5c —upper panel). Differentially expressed genes are bound by more floral regulators in their promoters and distal intergenic enhancers than genes without significant expression change during flower development. Moreover, differentially expressed genes have higher density of TF-binding sites at their predicted enhancers than at their promoters (Fig. 5c—bottom panel). The above results support the idea that enhancers function as integrated platforms for binding of multiple TFs to drive developmentally dynamic gene expression[2,9].

**Enhancer dynamics drives stage-specific gene expression**. To investigate how stage-specific enhancers, which were predicted based on distal intergenic DHSs, affect stage-specific gene-regulatory programs, we focused on the 810 distal intergenic DHSs whose accessibility varied at least 1.5 times among the four investigated floral stages. These dynamic enhancers were grouped into two distinct clusters based on their behavior in the time-course experiments (Fig. 6a; Supplementary Fig. 14 and Supplementary Data 5). Enhancers in cluster 1 were highly active at early stages but not at the later stage, while cluster 2 enhancers showed an opposite trend. For example, *UNUSUAL FLORAL ORGANS* (*UFO*) that is involved in specification of floral whorls in the meristem[46] is an enhancer-associated gene in cluster 1, while *AG* was identified to be a target gene of a cluster 2 enhancer (Fig. 6a). Overall, the dynamics of enhancers in the two clusters coincided well with the expression changes of their nearest-neighbor genes (Fig. 6b; Supplementary Fig. 15a). This observation was further supported by an analysis *per target gene*, in which the correlation between gene expression and chromatin accessibility at enhancer regions is significantly higher than that at promoters (paired Student's *t* test < 0.05; Fig. 6c; Supplementary Fig. 15b). The above results indicate that enhancer dynamics has a strong impact on stage-specific gene expression during flower development.

In order to identify candidate TFs contributing to stage-specific activity of enhancers, we searched for potential TF DNA-binding sites making use of ChIP-seq data as well as motif scanning (Fig. 6d). We focused on the distal DHSs that showed at least 1.5 times difference in accessibility between 2 and 8 DAI. We first tested whether AP1 and SEP3, two major floral regulators for which the time-course ChIP-seq data are available[18], bind to developmentally dynamic enhancers. Indeed, we found that AP1 and SEP3 bind to stage-specific enhancers in accordance with dynamic chromatin accessibility (Fig. 6e). We then searched for other TFs with known DNA-binding motifs that could bind to these stage-specific active enhancer candidates. We found that 125 and 81 TF-binding motifs showed a stronger enrichment (more than twofold in difference) in the active enhancers specific to 2 DAI and 8 DAI, respectively (Fig. 6d; Supplementary Data 5). For the highly active enhancers at 2 DAI, two LATERAL ORGAN BOUNDARIES DOMAIN (LBD) proteins, LBD18 and LBD13, were among the top-enriched binding TFs. AP2-EREBP and WRKY TF families predominantly bound at enhancers that were active at 2 DAI. Among the enhancers that were highly active at 8 DAI, the binding sites of basic helix–loop–helix proteins (bHLH) and TEOSINTE BRANCHED1, CYCLOIDEA, AND PCF TRANSCRIPTION FACTORS (TCP) proteins were

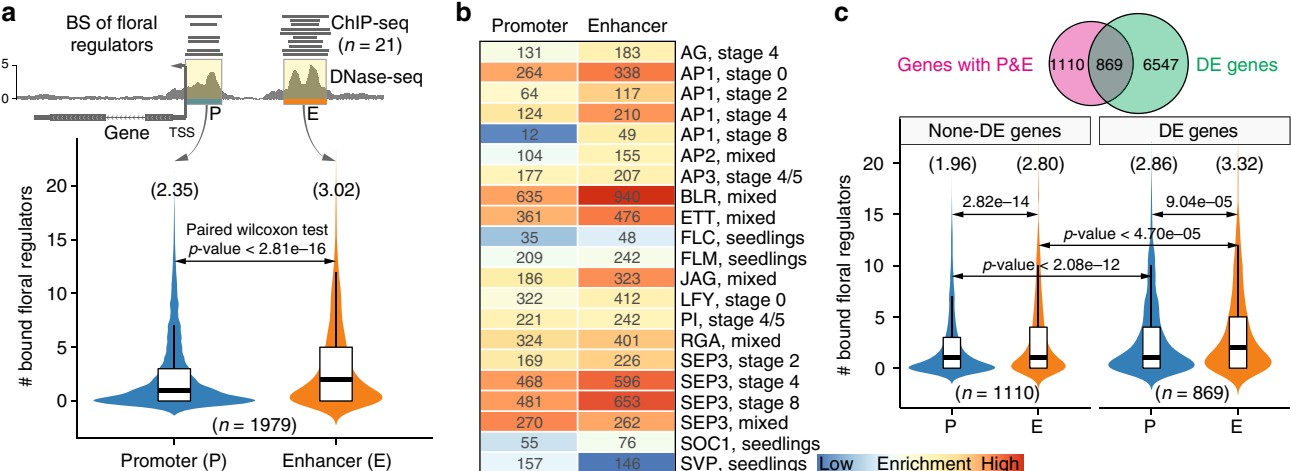

**Fig. 5** Enhancers show more enrichment in binding sites of floral regulators than promoters. **a** Violin plots (with boxplots inside) showing the number of transcription factor binding sites in promoters (P) and enhancers (E). The above diagram shows the strategy used in the analysis. Only genes with both promoters and enhancers were considered. Twenty one ChIP-seq datasets for 15 floral regulators were included in the analysis (data from ref. [48]). The numbers in the parentheses above the plots indicate the average value in each category. Boxplots show the median (horizontal line), second to third quartiles (box), and Tukey-style whiskers (beyond the box). BS binding sites. **b** Heatmap showing the enrichment of binding sites of different regulators in promoters and enhancers, respectively. Enrichment scores were calculated as log10(p-value) based on hypergeometric tests, using all the annotated protein-coding genes as the background. Numbers indicate the number of proximal genes in each category. **c** Similar to **a**, analysis was performed respectively on differentially expressed (DE) genes and non-DE genes during flower development. The Venn diagram above shows significant overlap (hypergeometric test, p-value < 1.92e−64) between genes having promoters/enhancers and genes exhibiting differential expression. The underlying source data for **a** and **c** are provided in the Source Data file

overrepresented (Fig. 6d, f). The above results indicate that dynamic enhancers provide binding sites for potentially stage-specific TFs to drive gene expression.

**Intronic enhancers and enhancer clusters**. Master regulatory genes often contain regulatory introns and some of these introns may harbor enhancers (i.e., intronic enhancers), as exemplified by the second intron of the *AG* gene[47]. However, no systematic investigation on predicted intronic enhancers exists in plants so far. To this end, we extended our analysis by focusing on intragenic DHSs that are > 1 kb from the TSS (Fig. 7a; Supplementary Data 6). We identified 2142 intragenic enhancer candidates, of which nearly one-third was derived from intronic regions and the rest mostly overlapped with 3′ untranslated regions (3′ UTRs) of protein-coding genes (Fig. 7a). Interestingly, we found that these predicted intronic enhancers (like predicted intergenic enhancers) were associated with significantly higher target gene expression dynamics than DHSs in 3′ UTR regions (Fig. 7b) and with significant enrichment in binding sites for a catalog of 15 developmental master TFs[48] (Fig. 7c). This suggests that the predicted intronic enhancers harbor important regulatory elements for the control of developmental transcriptome dynamics. We further noticed that 158 genes, including *AG*, *GOLDEN2-LIKE 2* (*GLK2*), and *SHOOT MERISTEMLESS* (*STM*), are associated with both candidates of intergenic and intronic enhancer elements (Fig. 7d). This set of genes is highly enriched in biological processes, such as "regulation of gene expression", "meristem development", "developmental growth", and "flower development" (Supplementary Data 6).

Since intergenic and intronic enhancers showed similar impact on developmental transcriptome dynamics (Fig. 7b), we asked whether combinatorial control by multiple enhancers could enhance target gene expression dynamics compared with singleton enhancers. Indeed, genes with both intergenic and intronic enhancers display significantly higher expression dynamics than intergenic or intronic enhancers alone (Fig. 7e). Accordingly, genes associated with multiple enhancer elements

(including either intergenic or intronic enhancers) tend to show significantly higher expression dynamics and higher activity of the associated enhancer elements (Fig. 7f). The above results highlight the important role of enhancers' multiplicity for combinatorial gene regulation by a set of TFs. In particular, several neighboring enhancers may form an enhancer cluster to regulate the corresponding target gene (see the Methods section). In this regard, we found that 94 genes have at least four enhancers (Fig. 7f; Supplementary Data 7), including the *TARGET OF EARLY ACTIVATION TAGGED 1* (*TOE1*) gene (Fig. 7g). Interestingly, these stretched enhancer elements are usually bound by similar sets of transcription factors (Fig. 7g), a dominant feature of superenhancers or stretched enhancers[11,49] that have so far only been described in animal systems.

## Discussion

The general features and cell-type specific activities of animal enhancers have been intensively studied. In contrast, much less is known about dynamic activities of enhancers during plant development. Our study provides insights into the features, dynamics, and role of enhancer activities in the control of gene expression during flower morphogenesis.

Only recently, enhancers were mapped in plant genomes, and the results revealed that those enhancer candidates correspond to accessible distal DNA fragments[19–21]. We mapped stage-specific enhancers based on the DHS dynamics across flower development. We predicted much less intergenic enhancers than Zhu et al.[20], mostly due to the different criteria used in peak calling. To further support the power of our prediction, we functionally validated 22 enhancer candidates in either forward or reverse, or both directions (in total, 30 constructs). The results showed that 27 of 30 (90%) reporter lines showed an ability of triggering GUS gene expression, which supports the reliability of our enhancer prediction. More than one-quarter of the H3K27ac sites locate in distal intergenic regions in animal systems[22,50]. Similarly, around 30% of H3K27ac sites were identified to locate in intergenic regions in both rice[23] and maize[24], which indicates that H3K27ac

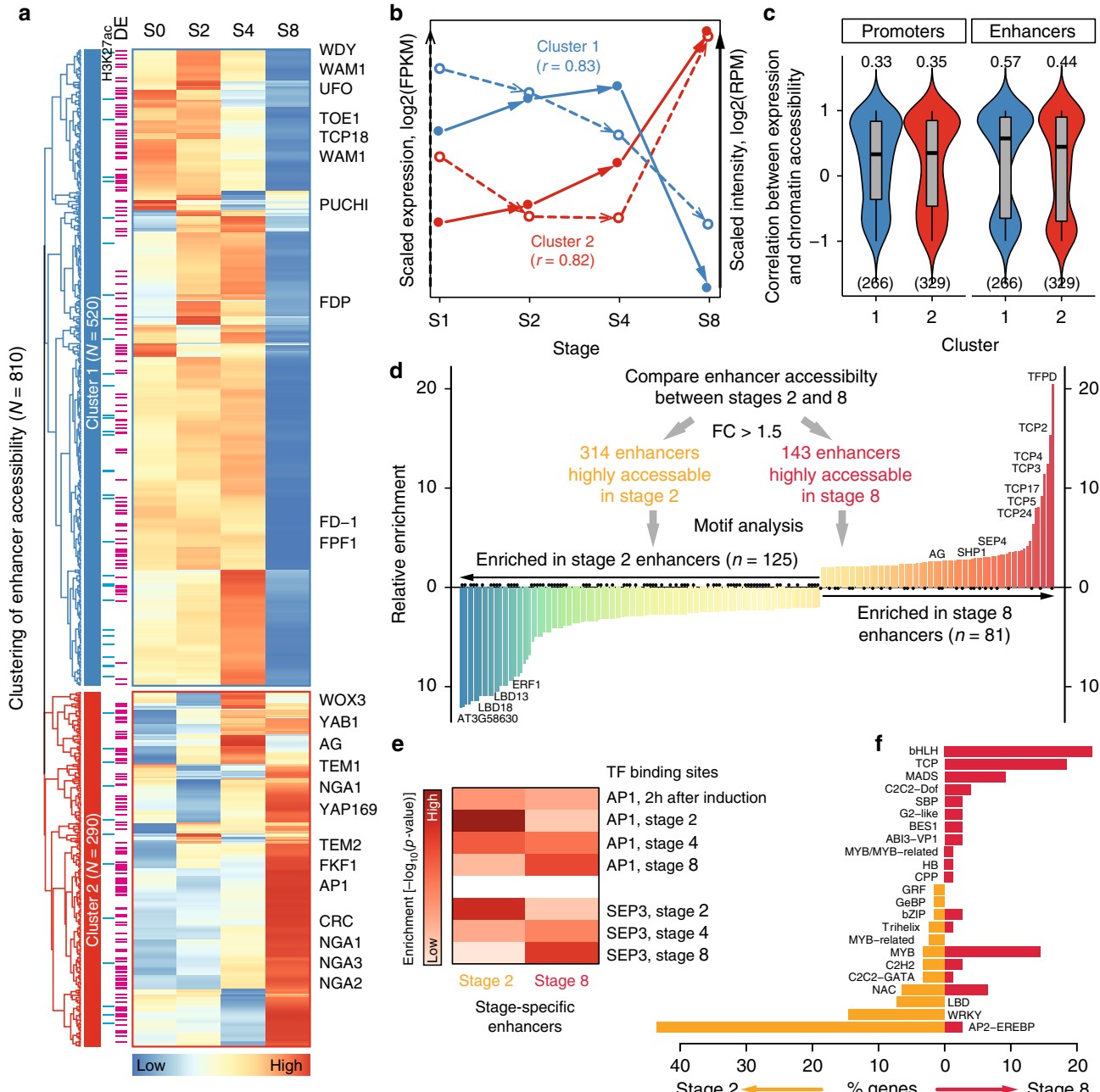

**Fig. 6** Enhancers control developmental stage-specific gene expression programs. **a** Heatmap illustrating the clustering analysis of 810 enhancers whose accessibility dynamically changes for more than 1.5-fold across the four stages (from stages 0 [S0] to 4 [S4]). The enhancers are grouped into two clusters. The blue bars on the left of the heatmap indicate that there are H3K27ac peaks for the corresponding enhancers. Pink bars indicate that the nearest genes are differentially expressed (DE). Flowering genes are labeled on the right of the heatmap. **b** The median expression level (in dashed arrow) and enhancer accessibility (in solid arrow) across developmental stages for the genes from the two clusters in **a**. Pearson correlation coefficients $r$ between gene expression and enhancer accessibility for each cluster are shown in parentheses. **c** Violin plots (with boxplots inside) showing the Pearson correlation coefficient between expression level and chromatin accessibility for each gene in different clusters. Promoters with respect to the enhancers in each cluster were used for comparison. The median correlation coefficient is shown above the violin plot. The number of genes used in the analysis is indicated in parentheses. Note that genes with missing expression data were removed from the analysis. **d** Identifying stage-specific enhancers at stages 2 and 8. Histogram shows enriched motifs in the stage-specific enhancers. Only motifs showing at least twofold changes in enrichment between stages 2 and 8 are plotted. **e** Enrichment analysis of AP1 and SEP3 binding sites (time-course ChIP-seq data from refs. [18,72]) in stage-specific enhancers. Heatmap shows log10-transformed $p$-values from hypergeometric tests. **f** Overview of TF families for TF genes with enriched motifs as shown in **d**. The underlying source data for **a**–**f** are provided in the Source Data file

could serve as an active enhancer mark in these two plant species. In contrast, in the *Arabidopsis thaliana* genome, our analysis confirms another report[25], which showed that only a minority, ranging from 0.5 to 3%, of H3K27ac sites are located more than 1

kb upstream of the TSS (Fig. 1d; Supplementary Fig. 11a; for comparison, 12.7% in rice using a reanalyzed result from ref. [23]), indicating that H3K27ac does not mark enhancers in *Arabidopsis thaliana*. However, this cannot be explained as a simple

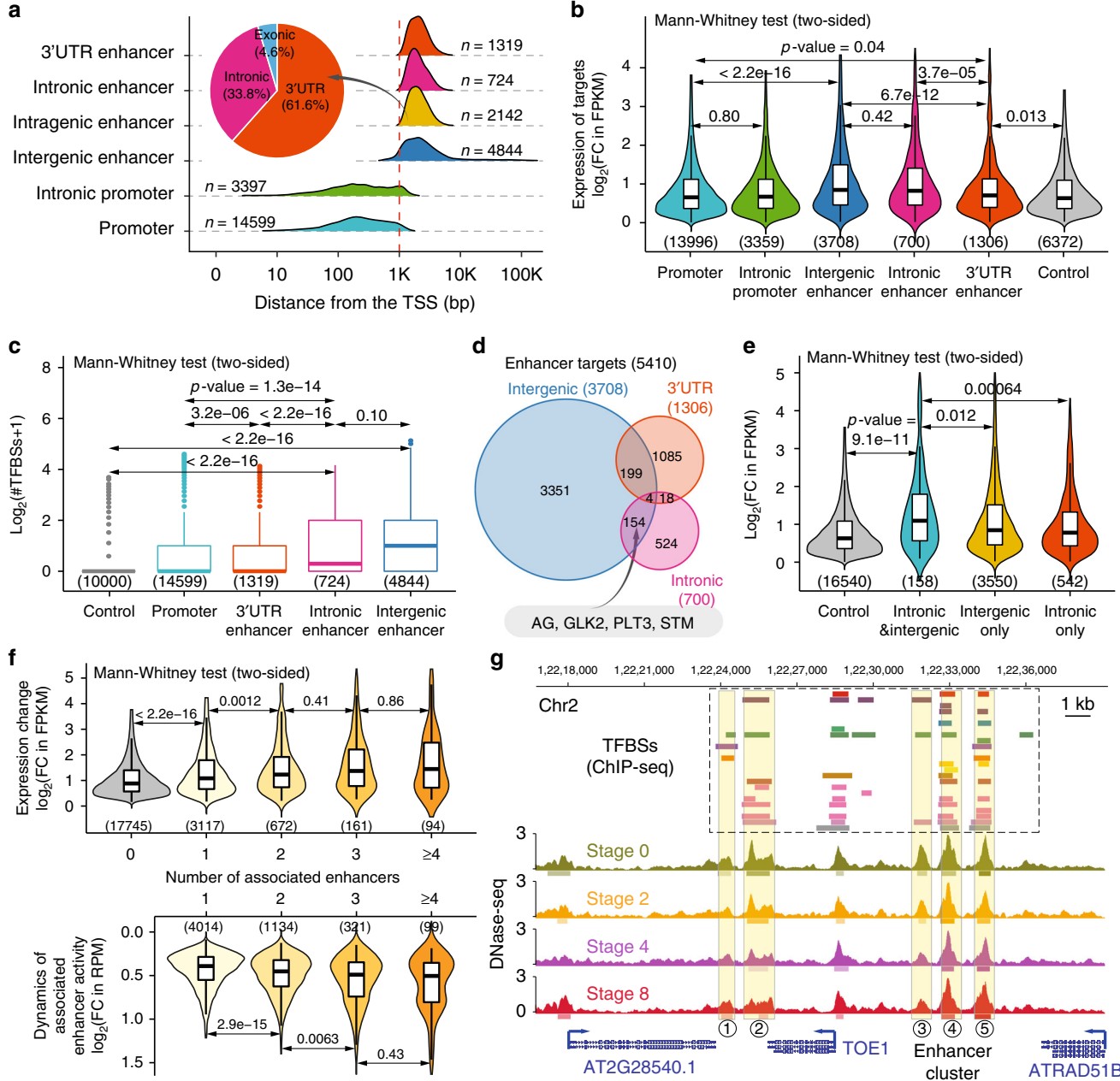

**Fig. 7** Intragenic enhancers and enhancer clusters. **a** Distribution of the distance of *cis*-regulatory elements from the TSS. Inset pie chart depicts the genomic context of intragenic enhancers. The number of elements in each category is indicated. **b** Violin plots (with boxplots inside) showing expression dynamics of predicted target genes. The y axis is the highest log2-transformed fold change (FC) of expression levels (in FPKM) during flower development. The number of predicted target genes in each category is shown in parentheses. **c** Boxplots showing the distribution of the number of transcription factor binding sites (TFBSs; data from ref. [48]) overlapping with *cis*-regulatory elements in each category. In the analysis, 10000 random genomic regions are used for control. Boxplots show the median (horizontal line), second to third quartiles (box), and Tukey-style whiskers (beyond the box). **d** Venn diagram showing the overlap of target genes for intergenic and intragenic enhancers. Example genes with both intergenic and intronic enhancers are shown in the box below. The total number of genes in each category is shown in parentheses. **e** Violin plots (with boxplots inside) showing expression dynamics of target genes with intergenic enhancers only, with intronic enhancers only, and with both intergenic and intronic enhancers. Genes without any enhancers are used for comparison. **f** Violin plots (with boxplots inside) showing expression changes of target genes (top panel) or the change of enhancer activity (in RPM, calculated based on chromatin accessibility/DNase-seq data; bottom panel) over the four developmental stages with increasing number of associated enhancers. Only intergenic and intronic enhancers were considered. The number of enhancers (bottom) and predicted target genes (top) in each category is labeled in parentheses. Genes without any enhancers are used for comparison. **b, c, e, f** Boxplots show the median (horizontal line), second to third quartiles (box), and Tukey-style whiskers (beyond the box). BS binding sites. **g** Genome browser view of annotated enhancers at the locus of *TOE1*. Putative enhancers are numbered. Please note that the fifth enhancer could belong to the next gene *ATRAD51B* based on genomic distance. See Fig. 4j for the color code of tracks. The underlying source data for **a–c** and **e, f** are provided in the Source Data file

consequence of genome compactness, since, e.g., the Drosophila genome is also compact, but H3K27ac marks its enhancers[51]. An important question is now to understand how the diverging H3K27ac patterns are established in different plant species. In animal cells, H3K27ac deposition in enhancers is mainly mediated by the CREB-binding protein (CBP) and p300 histone acetyltransferases[52,53]. An *Arabidopsis thaliana* CBP/p300 homolog was reported to promote flowering by affecting expression of *FLOWERING LOCUS C* (*FLC*), but does not appear to control acetylation levels of the H3 (H3K9ac and H3K14ac) and H4 histones within FLC chromatin[54]. Plant and animal CBP/p300 proteins share a highly conserved C-terminal domain, which is necessary for acetyltransferase activity[55]. Other domains have diverged between plant and animal homologs, suggesting divergent mechanisms of recruitment to their genomic target sites[56]. This however does not explain the differences in H3K27ac patterns observed in *Arabidopsis thaliana* and rice/maize. One possible explanation is that the interactions among different types of chromatin modifications, including histone acetylation, methylation, and DNA methylation are plant species-specific. It has been shown that in rice, DNA methylation at non-CG sites positively correlates with H3K27me3 in euchromatic regions[57], which is a different situation in *Arabidopsis thaliana*[58]. Whether H3K27ac mediates interactions of other chromatin features and whether the interaction modes differ between plant species is an open question to be addressed. In animal systems, H3K27me3 correlates with poised enhancers, whereas the H3K4me1 mark can correlate with poised and active enhancers[59]. In *Arabidopsis thaliana*, around 15–20% of H3K27me3 sites were found in intergenic regions[25,60] (20.8% in rice, using reanalyzed data from ref. [16]), whereas H3K27me3 was reported to have a positive association with poised enhancers[20]. Only 0.6% of H3K4me1 and <1% of H3K3me3 sites are located in intergenic regions[61]. Other chromatin features of active enhancers and the mechanisms underlying their characteristics thus need to be further explored in *Arabidopsis thaliana*. Interestingly, 9% of the distal DHSs we mapped carry H3K27me3, and 31% of those H3K27me3-marked distal DHSs are actively changing across flower development.

Taking advantage of the inducible synchronized flower development system, we were able to dissect the dynamics of enhancer activities in floral organ development. Our results show that a certain proportion, ranging from 2 to 12% of intergenic enhancers, were stage-specific (Fig. 3c) and their activity was surprisingly highly correlated with the expression level of proximal genes (Fig. 6b). This indicates a crucial role of enhancer dynamics in stage-specific gene expression for flower development. Similar observations have been reported in animal systems[12,62,63], which indicates that the developmental expression dynamics might be largely controlled by distal enhancers. Intriguingly, 21 of the annotated flowering-time genes[64] were associated with dynamic enhancers, and 17 (81.0%) of them were differentially expressed during flower development (Supplementary Data 5). More than half (53.3%) of the enhancers were bound by at least one of the 15 floral master regulators (Supplementary Data 5). These results highlight the importance of enhancers for re-shaping developmental transcriptome dynamics during flower development, preferentially via regulation of floral master regulators.

Enhancer activity is known to be regulated by TF-binding events[65]. We found that enhancer dynamics across flower development is highly associated with dynamic binding of floral master TFs. For instance, AP1 and SEP3 were found to bind active enhancers in a stage-specific manner (Fig. 6e). Interestingly, we observed that some master regulators of flower development preferentially bind to enhancers instead of promoters (Fig. 5b). This result suggests that developmental stage-specific gene regulation requires binding of function-specific TFs to

enhancer elements, which may then result in recruitment of the basic transcriptional machinery to core promoters.

We predicted ~2100 intragenic enhancer candidates, and about 34% of all predicted enhancers locate in introns (Fig. 7a). Intronic regulatory regions have been reported to be essential for determining the activation of several important developmental genes in distinct spatiotemporal patterns (e.g., see refs. [66–68]), which indicate that these intronic regulatory regions contain repressive or enhancing elements, as was, for example, shown for the *AG* second intron[47]. In addition, introns can enhance transcription via gene looping as was shown in yeast[69], but intronic enhancers can also attenuate gene expression through eRNA transcription[70]. Intriguingly, 158 genes are enhancer-associated candidates with both intergenic and intronic enhancer elements. This set includes many developmental TF genes, but also other signaling-related genes, such as *PIN* auxin transporters (Supplementary Data 6). Genes with both intronic and intergenic enhancers, as well as genes that show clusters of enhancers, show particularly dynamic expression changes. With these data, the question arises whether such clustered enhancers can function as so-called "super-enhancers" in plants. In animal systems, the functional relevance of "super-enhancers" is still under debate[71]. They were suggested to act as switches to determine cell fate and cell-type specific gene activities. Essentially, super-enhancers are typically defined by closely located enhancer clusters and concerted binding of specific master TFs. Indeed, our identified enhancer clusters typically serve as binding platforms of developmental master TFs, and different individual enhancers within the cluster show similar sets of TF-binding sites, suggesting that they may act in an additive or synergistic manner to establish a strong, robust gene activity in specific cell types. Further research should aim to dissect the functions of individual enhancers or individual *cis*-regulatory components within enhancer clusters in plants, for example, by making use of Cas9-based mutagenesis.

In addition to the widely accepted concept that gene transcription is triggered by binding of pioneer TFs that open the chromatin of core promoters, thereby allowing the successive binding of more regulators, our work uncovered another regulation layer of developmental stage-specific gene transcription by the dynamic activities of enhancers in plants. In conclusion, dynamics of enhancer activities, H3K27ac and TSS-proximal DHS together contribute to developmental stage-specific gene expression during flower formation.

## Methods

**Plant materials and growth conditions**. An inducible line in which the AP1 protein fused with the glucocorticoid receptor (GR) is expressed under the AP1 promoter in an *ap1-1 cal-1* double-mutant background (pAP1:AP1-GR *ap1-1 cal-1*)[72] was grown for mapping stage-specific H3 and H3K27ac, and for gene expression profiling by RNA-seq in flowers. The same material had been used previously for generating stage-specific open chromatin data in flowers by DNaseI-seq in the same group[18]. Seeds were directly sown to soil. After cold treatment for 2 days, the plants were transferred to growth chambers (Fitotron SGC 120, Weiss Technik, UK) with long-day conditions (16-h light, 8-h dark) at 20 °C, at light intensity of 100 μmol/m$^2$/s.

**RNA-seq and ChIP-seq experiments**. Plants were induced by dexamethasone as described in ref. [18]. In brief, dexamethasone (DEX) solution (2 μM dexamethasone, 0.01% (v/v) ethanol, 0.01% Silwet L-77) was applied daily to the main inflorescences from the time they reached a height of 2 cm, as drops from cut pipette tips. The first induction was applied after 8 h of light, and the following daily inductions were performed after 4 h of light. Inflorescence material was collected before the first induction for the "no DEX induced tissue", and then 2 days, 4 days, and 8 days after the first treatment, after 8 h of light. Flower tissues of specific developmental stages were collected from the first inflorescence of pAP1:AP1-GR *ap1-1 cal-1* plants. For RNA-seq, five inflorescences per replicate were harvested with liquid N$_2$. RNA was isolated with the RNeasy Plant Mini Kit (Qiagen, Germany) according to the manufacturer's recommendations, and the libraries were prepared using the TruSeq Stranded Total RNA LT Sample Prep Kit (Illumina, USA).

For ChIP samples, 0.7 g of tissue was collected for one replicate into a 50 -ml Falcon tube on ice. The ChIP experiment was performed according to the published protocol[73] with minor modifications. Briefly, the tissue was fixed with 1% formaldehyde for 20 min. After nuclei isolation, the chromatin was sonicated using a Covaris S220 sonicator to obtain the desired DNA fragment size (major band at 500 bp). The sonicated chromatin was pre-cleaned by two rounds of centrifugation with maximum speed at 4 °C. In total, 5 μl of anti-H3K27ac (Lot.2322526, Millipore, Germany) or 3 μl of anti-H3 (ab1797, Abcam, UK) were used to precipitate the chromatin fragments that contain either all H3, or K27ac-marked H3 histones. Specificity of the anti-H3K27ac antibody was verified prior to its usage (Supplementary Fig. 1). ChIP-seq libraries were prepared using ThruPLEX DNA-seq kit (RUBICON GENOMICS, USA), following the manufacturer's recommendations. Control libraries were prepared in the same way by using purified DNA from sonicated chromatin. H3 ChIP samples and ChIP-seq libraries were prepared and sequenced as described in ref. [27]. All RNA- and ChIP-seq libraries were sequenced using the NextSeq®550 sequencer (Illumina, USA).

**RNA-seq data analysis**. RNA-seq reads were mapped to *Arabidopsis thaliana* reference genome (TAIR10) using STAR[74] (version 020201). Expression levels of all annotated genes were estimated by RSEM[75] (version 1.2.22). FPKM (fragments per kilobase of transcript per million mapped reads) values as defined by RSEM were added as a pseudo-value of 1e−6 (to avoid zeros) and then log2-transformed. Differentially expressed genes across the four developmental stages were identified by analysis of variance (ANOVA) based on FPKM values. The resulting *p*-values were adjusted for multiple comparisons by false discovery rate (FDR). Genes were considered as differentially expressed if they showed at least twofold changes with FDR < 0.05.

**ChIP-seq data analysis**. Time-course H3K27ac and H3 ChIP-seq data were generated in this study. Published histone modification ChIP-seq data in *Arabidopsis thaliana* seedlings[20,30] (including H3K27ac, H3K4me1, H3K4me2, H3K4me3, H3K9ac, H3K9me2, H3K27me3, and H3K36me3) and rice seedlings[31] (including H3K4me1, H3K9me2, H3K27me3, and H3K36me3) were collected from the Sequence Read Archive (SRA) database. The reanalysis results of TF ChIP-seq data for 15 floral regulators were obtained from ref. [48]. A uniform computational pipeline (as described below) was adopted to process the raw data.

We followed the ChIP-seq data analysis guidelines[76] recommended by the ENCODE project and have developed an analysis pipeline[48] consisting of quality control, read mapping, peak calling, assessment of reproducibility among biological replicates, and peak annotation to reprocess all raw data in a standardized and uniform manner. Specially, the quality of the raw data (FASTQ files) was evaluated with the FastQC program (http://www.bioinformatics.babraham.ac.uk/projects/fastqc/). Reads were then mapped to the *Arabidopsis thaliana* genome (TAIR10) using Bowtie (version 1.1.2) with parameters "–threads 8 -n 2 -m 10 -k 1–best–chunkmbs 256 -q". Redundant reads were removed using Picard tools (v2.60; http://broadinstitute.github.io/picard/). Peak calling was performed using MACS2[77] (version 2.1.0). Duplicated reads were not considered (–keep-dup = 1) during peak calling in order to achieve a better specificity. The "–mfold" parameter was set as "2-20" to build the model. A relaxed threshold of *p*-value (*p*-value ≤ 1e−2) was suggested in order to enable the correct computation of IDR (irreproducible discovery rate) values[76]. Following the recommendations for the analysis of self-consistency and reproducibility between replicates (https://sites.google.com/site/anshulkundaje/projects/idr), control samples were combined into one single control among the replicated experiments. Peaks across replicates with an IDR ≤ 0.1 were retained.

Signal files for H3K4me2, H3K4me3, H3K9ac, and H3K27ac ChIP-seq data were downloaded from the UCSC genome browser at http://structuralbiology.cau.edu.cn/cgi-bin/hgTracks since the corresponding raw data were not available[23]. The analyzed wiggle track files for histone modification ChIP-seq in human K562 cell line were obtained from the ENCODE (Encyclopedia of DNA Elements) project[6] at https://www.encodeproject.org/.

**DNase-seq analysis**. Floral stage-specific DNase-seq data generated from the pAP1:AP1-GR *ap1-1 cal-1*) were obtained from Pajoro et al.[18]. DNase-seq data analysis was performed in a way similar to that described above for ChIP-seq. Peaks (called DNase I hypersensitive sites, DHSs) were called using MACS2 without providing an input file and with parameters "–nomodel–shift −100–extsize 200".

**Peak analysis**. All the peak-based analyses (including peak overlapping, merging, and summary) were performed using BEDTools[78] (v2.25.0). If multiple H3K27ac peak regions or DHSs at a given stage resided within 500 bp to each other, they were considered as a single peak and centered on a peak having higher H3K27ac occupancy or DNase I accessibility. An alternative approach was also used to identify nearby peaks between DHSs and H3K27ac peaks, where any peaks within 1 kb of the base peak were considered.

**Genomic distribution of peaks**. The gene annotation file (TAIR10_GFF3_genes.gff) was downloaded from the TAIR website. The genomic coordinates of

promoters were defined as 1 kb upstream of the transcription start site (TSS) of annotated protein-coding genes. Intragenic regions refer to the region between the TSS and the transcription termination site (TTS), i.e., the gene body. The rest of the genome was annotated as distal intergenic regions. Peaks were assigned to each category using their peak summits. If a peak overlaps multiple categories, only one category is used based on the following priority: intragenic > promoters > distal intergenic regions.

**Enhancer prediction**. We adopted a similar approach in Zhu et al.[20] to predict plant enhancers based on DHSs. We measured the distance from the midpoint of each DHS to the TSS of its closest genes, and we found that the distance follows two distinct distributions (Supplementary Fig. 9d). For intergenic DHSs (Fig. 2c), the first part (i.e., TSS proximal) of DHSs resided within 1 kb from the TSSs, while the second part (distal) of DHSs was 1 kb or further away from the TSSs. Therefore, intergenic enhancers were defined as intergenic DHSs > 1 kb from the nearest TSS. The intragenic DHSs (Fig. 2b) locating > 1 kb from the TSS (> 1 kb) were defined as intragenic enhancer candidates, which overlapped with either introns or 3′ untranslated regions (3′ UTRs). Note that some DHSs (about 2% of the total DHSs) can be longer than 2 kb. To avoid false positives in the intragenic enhancer prediction, the distance between DHSs and the TSS was calculated based on the whole peak region rather than the peak summit.

**Identification of enhancer clusters**. The identified DHS-based enhancer regions (regions within promoters were disregarded) were stitched together based on a specific distance. The stitching distance was determined to optimize that enhancers in the same intergenic region are combined, while enhancers from different intergenic regions avoid to stitch. In our analysis, we selected a stitching distance as 1.5 kb (Supplementary Fig. 16) to combine multiple individual enhancers into enhancer clusters. The above analytical principle is similar to that used for super-enhancer identification[11], but without a step to rank-stitched enhancers by their signal. An enhancer cluster was assigned to the "closest" gene as to the minimum distance of any individual enhancers within this cluster.

**Validation of enhancer function**. The GUS reporter system for validating enhancer function was slightly modified from ref. [20]. In detail, one DNA fragment containing multiple cloning sites (MCS) and a minimal 35 s promoter element was incorporated into TOPO sites of pENTR/D-TOPO plasmid to generate pENTR/D-TOPOmini35S. The DNA fragment containing the candidate enhancer was amplified using primers extended by sequences containing restriction enzyme cutting sites. The digested PCR product was directly inserted into the entry vector pENTR/D-TOPOmini35S via the MCS. Next, to produce the final construct, the enhancer fragment followed by the minimal 35 s promoter was transferred into the destination vector pKGWFS7.0 using the Gateway LR Clonase II Enzyme Mix (Thermo Fisher Scientific, USA). Plant transformation was performed by dipping *Arabidopsis thaliana* Col-0 plants[79] using the *Agrobacterium tumefaciens* strain GV3101. Seeds from infiltrated plants were harvested and spread on selective medium (1/2 MS including vitamins, 50 μg/ml kanamycin). Positive transformants were transferred to soil and grown under long-day conditions (18–22 °C, 16/8 h light/dark).

The main inflorescence was harvested from each transgenic line and GUS staining was performed as described[20] with minor modifications. In brief, inflorescences of independent transgenic lines produced by the same transformation were mixed in one single tube filled with GUS staining solution. Vacuum was applied for 5 min followed by 6 h of incubation at 37 °C. Tissue was cleared overnight using a 3:1 mix of ethanol and acetic acid, respectively, followed by two subsequent washes with 100% ethanol. For microscopic analysis, samples were transferred to 70% ethanol and stored at 4 °C. Pictures were taken using the Olympus LC30 microscope (Olympus, Japan).

**Assignment of target genes**. To link regulatory elements (including promoters and enhancers) with their associated genes, each regulatory element was assigned to its nearest TSS of 27,206 annotated protein-coding genes. Flowering genes were either collected from the flowering interactive database (FLOR-ID, http://www.flor-id.org/; 306 genes)[64] or selected from *Arabidopsis thaliana* transcription factors with gene ontology (GO) terms containing the keywords "floral" or "flowering" (128 genes in addition). In total, we obtained 434 "flowering" genes.

**Dynamics of DHSs or H3K27ac modification**. DHSs identified in the four developmental stages were merged into a single peak file to create a DHS superset. This DHS superset contains genomic regions that were hypersensitive in at least one of the stages. DNase I cleavages were then counted within each region in the DHS superset at each time point using BEDTools. In order to compare changes in DNase I accessibility at different genomic regions across time points, quantification was normalized to library size (reads per million, RPM). DNase-seq experiments across multiple time points are affected by global confounders related to changes in the number of sites at each time point, which is not accounted for via simple depth normalization. To remedy this issue, quantification of DHSs was further quantile normalized across all time points using the normalized quantiles function from the

preprocessCore R package. Only DHSs that displayed at least 1.5-fold enrichment above input were selected for further analysis.

The above approach was applied to the merged H3K27ac peaks to assess H3K27ac dynamics. Using H3 ChIP-seq data for S0, S2, and inflorescence[27] samples, we found that the dynamics of H3K27ac is independent on the dynamics of H3, indicating that H3K27ac changes do not simply reflect altered nucleosome occupancy (Supplementary Figs. 17, 18).

**Motif analysis**. We collected motif models for *Arabidopsis thaliana* TFs from public databases and from previous studies (see Supplementary Table 1 for a summary). Only motifs with experimental support (including protein-binding microarray [PBM], systematic evolution of ligands by exponential enrichment [SELEX], ChIP-chip, ChIP-seq, and DAP-seq) were used in the analysis. If multiple motifs existed for a given TF (e.g., motif models for the same TF from different experimental protocols or studies), the motif with the most substantial experimental support was selected.

Potential TF-binding sites were determined by scanning motif occurrences in the region of interest using the Find Individual Motif Occurrences (FIMO, version 4.10.2) with a *p*-value threshold of $10^{-4}$ and defaults for other parameters. The relative enrichment was defined by the percentage of regions supported by motif occurrences for each motif.

**Gene ontology analysis**. Significant GO terms were identified using the agriGO[80] online toolkit. All *p*-values plotted are corrected for multiple testing.

**Statistics and data visualization**. If not specified, R (https://cran.r-project.org/; version 3.2.3) was used to compute statistics and generate plots. The HTPmod online tool (http://www.epiplant.hu-berlin.de/shiny/app/HTPmod/) was partially used for the generation of figures. ChIP-seq data signal tracks were visualized in the WashU Epigenome Browser (https://epgg-test.wustl.edu/browser/).

**Reporting summary**. Further information on experimental design is available in the Nature Research Reporting Summary linked to this article.

## Data availability

ChIP-seq data generated in this study data have been submitted to the NCBI Gene Expression Omnibus (GEO; http://www.ncbi.nlm.nih.gov/geo/) under accession number GSE112965. The RNA-seq data derived during this study are available under GEO accession number GSE110500. The source data underlying Figs. 1d, e, 2a, b, g, 3a, c, d, g–i, 4a, c, 5a, c, 6, 7a–c, e and f are provided in the Source Data file.

## Code availability

The ChIP-seq data analysis pipeline is adapted from https://github.com/PlantENCODE/plantGRNs. All other custom computer code is available from the corresponding authors upon reasonable request.

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

## Acknowledgements

We would like to thank Johanna Müschner for her assistance with RNA-seq sample preparation, Dr. Johanna Paijmans from Potsdam University for the assistance with sequencing, and Liangyu Fu for her nice suggestion on data analysis. The authors acknowledge the North-German Supercomputing Alliance (HLRN) for providing HPC resources that have contributed to the research results reported in this paper. Kerstin Kaufmann wishes to thank the Alexander-von-Humboldt foundation and the Federal Ministry of Education and Research for support. Julia Schumacher is supported by the International Max Planck Research School for Primary Metabolism and Plant Growth (IMPRS-PMPG).

## Author contributions

D.C., W.Y., D.D. and K.K. conceived the study. D.C. designed the study, performed the bioinformatic analysis, and made the figures with contribution from M.C. W.Y. performed the experiments with help from J.S. D.D. verified the specificity of the antibody of H3K27ac. J.E. and C.C.C. generated the H3 ChIP-seq dataset. D.C. and W.Y. drafted the paper. C.C.C. and K.K. improved the paper. All authors reviewed and approved the submitted version.

## Additional information

**Competing interests:** The authors declare no competing interests.

