## [Peer Review File · Nature Communications]

Reviewers' comments:

Reviewer #1 (Remarks to the Author):

This study aims at identifying enhancers involved in stage specific gene expression during flower morphogenesis. Authors discuss the relationship between open chromatin, H3K27ac and gene expression and establish a correlation between proximal H3K27ac enrichment and gene expression as well as a correlation between chromatin accessibility at distal DHSs and expression of putative target genes.

The attractiveness of the study lies into the use of a flower induction system. The study shows interesting results on distal DHSs showing characteristic signs of regulatory sequences. Nevertheless, it is conceptually not novel enough to be published in Nature Communications. Especially, the presence of H3K27ac at genic regions and its absence of enrichment at enhancer candidates are results that have been reported before. Only the second part of the analysis really focuses on enhancer candidates but lacks to report new concepts on plant enhancers. We do believe that after having taking into account our comments the paper is well suited to be published elsewhere.

In addition, the text is loaded with typos of various kinds. Inappropriate use of articles, lack of articles, plural/singular, words missing, use of incorrect words and other typos. This hampers the reading and in addition gets irritating at a certain moment. It would be good to use the spelling checker and have others proofread a manuscript on typos before sending it in. That allows a reviewer to fully focus on the content. Towards the end of the sections below, only the more prominent issues are being listed. Please check the rest of the document for typos yourself.

Review

- 50-51. This sentence is not entirely correct. The binding of pioneer TFs results in chromatin remodelling that then result in nucleosome depletion and the binding of more TFs, etc
- 53. Thus indicates a causal link with the previous sentence. In this case there is no link between enhancer distances from the TSS and how many enhancers are controlling a target gene.
- 59. H3K4me1 in animals is associated with both active and inactive enhancers, not just active ones.
- 65. 'In this regard'' refers to diseases. Reformulate!
- 71. Not sure if extrinsic cues can be counted as drivers of lineage specific transcription programs. Either remove 'lineage specific' or replace by 'specific'.
- 82. The reason provided is not the reason why enhancers in plants are poorly characterized. There have hardly been any genome-wide studies focusing on the characterization of enhancers in plants. Some plant scientists, even believe(d) there are no enhancers in plants. On top of that, most studies in plants use Arabidopsis as a model system. The genome of Arabidopsis is extremely compact, hampering the dissection of 'the promoter' into promoter-proximal sequences and enhancers.
- 85-86. There are a number of studies missing focused on the prediction of enhancers in plants that should also be acknowledged:
 - o 1) Zhang W, Zhang T, Wu Y, Jiang J. Plant Cell. 2012;24:2719–31.
 - o 2) Furthermore, there is similar study in rice: Zhang W, Wu Y, Schnable JC, Zeng Z, Freeling M, Crawford GE, et al. Genome Res. 2012;22:151–62. In the latter they looked at various different histone modifications (but not H3K27ac).
 - o 3) recently published stories are missing as well: Eli Rodgers-Melnicka, Daniel L. Verab, Hank W. Bassb, and Edward S. Buckler, Open chromatin reveals the functional maize genome; doi:10.1073/pnas.1525244113
 - o Oka R, Zicola J, Weber B, Anderson SN, Hodgman C, Gent JJ, Wesselink JJ, Springer NM, Hoefsloot HCJ, Turck F, Stam M. Genome Biology 2017; 18:137. The latter used a combination of DHS, H3K9ac and DNA methylation to predict active enhancers.
- 102. H3K27ac is not the hallmark for 'active' enhancer annotations...(add 'active')

- 116: Zhu, Zhang et al, Du et al, and Charron et al do not correlate H3K27ac with predicted enhancers. Plus, saying this is in contradiction with the sentence at lines 88-90.
- What is the reason to focus on H3K27ac and not on H3K9ac or other histone marks associated with active enhancers? As mentioned in the introduction, both seem linked to enhancer activity in animals. In addition, evidence for H3K27ac not being involved with active enhancers in Arabidopsis was already there (Zhu et al). So why further investigate this situation?
- 123-124. We assume stage 4-5 corresponds to 4 days after induction instead of 2 days as indicated in the manuscript.
- 137-139. The examples in Figure 1B show highest expression in stage 4, while H3K27ac and DHS signals were highest in S8 for AP3 and relatively static for AP1. The sentence does not reflect this observation.
- 141. 'genetic' should be 'genic'. 'towards' should probably be replaced by 'directly downstream of the' as towards does not reflect any notion of distance.
- 142. Indicate Figure 1D as well.
- 143. Why focus on only those two categories (stage 2 and 0)? It suggests the situation would be different for other stages while it does not seem to be the case.
- Figure 1C. The legend does not specify the % clearly enough. Do you mean the expressed genes were divided in equal groups (equal nr of genes) or are particular thresholds taken associated with particular expression levels?
- Figure 1D. How are the peaks defined and assigned to specific genomic regions? This needs to be further detailed in M&M. What if peaks overlap multiple categories? For example if the CDS and intron are part of a region showing H3K27ac, is this region assigned to one of them or split in two categories?
- Fig1E. Is this referring to all peaks or only distal peaks? If referring to all, how does the data look like for distal peaks specifically? And for the promoter/CDS peaks?
- 152. What do you mean with 'highly induced flower tissues'? Please specify. And how does this fit with the statement that they gained more stage-specific H3K27ac sites? There are more unique peaks in S0 than in S2-S4. I would expect S8 (or S2 or S4) to be the highly induced tissues, instead of S0.
- 153-154. Is likely that the correlation between H3K27ac and expression is not just true for flower tissue. Would leave 'flower development' out in this sentence.
- 154-156. This has been observed by others as well. See e.g. papers cited above. In addition, the function of open chromatin and acetylation marks in transcription is well studied. The sentence suggests this is not the case.
- 167-168. The text indicates that the dynamic enrichment and change in gene expression levels generally follow a similar trend. This seems to be true for some examples (SVP and SEP3 for instance) but clearly not for all. Rather than showing cherry picked examples, it would be good to show, as presented in Figure 2B, the trend in expression for the different clusters. In addition we question the reasoning behind the decision to have 10 clusters. Some seem very redundant (cluster 2, 4 and 6 for instance).
- 176-184. The messages in this section are contradictory. Correlation between H3K27ac and expression is first described as positive (174-176), then questioned given the result of some specific clusters (176-181) and then positive again (182-184). In addition, we miss the data on correlation between DHS and expression. This cannot be deduced from Figure 3A.
- 178. You measured RNA steady state levels, not transcription (would need to do Gro-seq experiments or something similar). So you cannot use the word transcriptional. The use of the word transcription should be changed throughout the text.
- 189-191. Why between stages 2 and 8? You forget to mention this is done for sites that are accessible, rather than all sites.
- 193-194. The authors first have to mention where the ChIP data on the TFs is coming from.
- 194. Maser? You mean master?
- 194-198. The text mentions that the ChIP-seq datasets had been generated with plant tissues of different developmental stages. Still there is a statement saying that proteins functioning mainly at early stages are mainly bound to stage 2 specific DHSs. It is however unclear if the ChIP-seq has been done in stage 2 and stage 8, so if this statement can be made to start off with. Looking for

TF motifs is in no case an indication of TF binding and in a sense assuming binding of a TF due to the presence of a specific motifs could be seen as more bias than exploiting ChIP seq data in different tissues.

- 209-210. 'Interestingly, these stage-specifically overrepresented regulators also showed stage-specific expression patterns'. For the proteins mentioned in the previous sentence, this only holds for TFPD and TCP15. So 'These' is inappropriate. Furthermore, it is required to specify in which direction this stage-specific expression is. The overrepresentation of particular binding sites can be associated with a higher expression of the regulators, but also with a lower expression level.
- 211-212. 'Collectively, our results showed that the dynamic change of H3K27ac at the two analyzed stages largely...'
- 213. Here proximal DHS sites are mentioned. In the section this is not specified that the sites monitored in this section are proximal sites. Are they all proximal? IF so that needs to be specified at the start of the section.
- 217. Again, the message about H3K27ac being associated with active enhancers in ref 28 is not that clear.
- 225-227. From Figure 4A and B it is difficult to understand how DHS peaks can be located upstream of the TSS while they locate in similar position as H3K27ac peaks that are described as being downstream of the TSS. If the absolute value for distances was used, it should be mentioned in the legend.
- The data shown in S11 for H3K4me2 at distal DHSs in rice are different from the results shown in Zhang et al 2012 pp 151. In Fig 7 they show there is H3K4me2 enrichment around intergenic DH sites (panels D and H).
- 246-252. Fig 4D, It is unclear to which DHS sites this is plotted. Generated in the exact same tissue as the tissue in which the histone modifications have been examined? If not derived from the same tissue samples one cannot compare these data sets. Also, which DNase-seq data are shown on the vertical axis and which at the 'X-axes' (at the top)? See also comments below at Fig4 concerning the orientation of the data relative to the transcription direction. In addition, Oka et al showed that in maize, at distal DHS sites, H3K9ac is asymmetrically localized. You need to check if this is the case as well in Arabidopsis and rice and if so, orient the data such that the highest value is oriented towards only one specific site. The way the data are presented in 4D-F is in addition more suitable for a Supplemental Figure. Here data that are already published are analyses again. E.g. Zhang et al 2012 and Zhu et al 2015 already showed plots like in S11 in their paper. Luo 2013 mentions: 'This result suggests that the vast majority of histone modification peaks are found within the annotated genic regions rather than the intergenic regulatory sequences for Arabidopsis.'
- 4G. We wonder if the authors took into account that different antibodies have different kinetics, plus different antibodies will be of different quality. The figure kind of implies one can say there is less H3K4me2 in Arabidopsis and rice, than there is H3K4me3. The ChIP signal levels of different antibodies can however not be compared.
- 260, S11A. Not sure you can say 'depletion' of H3K27ac at distal DHS sites. There are small peaks visible in S11A.
- 262. Epigenetic should be chromatin signature. It is not shown that histone acetylation can be inherited through mitosis or meiosis.
- 275-276. Would be useful to mention how many of each category exist.
- 277-286. It is unclear what the added value of this analysis is. State 2 of ref 46 is also associated with coding sequences. It is associated with very low transcript levels. State 1 with high levels. Both 1 and 2 are associated with promoters. State 4, in which the distal DHSs are enriched with H3K27me3, so polycomb silenced chromatin. So those could represent silenced regulatory sequences, but this has not been tested. The authors should take this into account.
- 286. The authors identified less candidates than have been identified in Zhu et al (ref 28), despite defining shorter promoter regions (1kb instead of 1.5kb). Comments on why this is the case appear in the discussion, but an analysis of the overlap between ref 28 candidates and the ones identified in this analysis would be a good addition to the paper. Also, Zhu et al experimentally validated some of those enhancers. Having a large overlap between the two datasets would be a good way to support the quality of the prediction in this paper.

- 291. Why widespread?
- 292. Can you define precisely what the stage-specificity of promoters and distal DH sites is based on. This is missing in the main text and legend.
- 292-293. This indeed makes sense. Regulatory sequences are known to provide tissue-specificity to promoters. Also refer to such findings in literature.
- 299. Compared to which type of sequences are the GWAS SNPs enriched in the distal and proximal DH sites? Please define this.
- 302. Define in the legend or methods what the flanking regions consist of (how much of the sequence around do you consider flanking, and what type of sequence do you use/not use)? What does the control in Figure 5F consist of? Would it be possible to include such control for figure 5E-I.
- 310, 5I. Where does the lncRNA data come from (reference is missing)? How are they defined?
- 326. It is not clear if you picked two particular clusters out of more than two or if the data was clustered in 2 clusters. Also, this is not clear from the legend of Figure 6.
- 330. What does proximal genes mean? Both up- and downstream? Only downstream? Only target genes?
- 337. The distal region is indeed upstream, but not far upstream.
- Both Figure 3E and 6D are about motif enrichment when comparing stage 2 and 8. It has to be clear from the start what the differences are between those two figures.
- 350-353. Twice stage 2 is addressed, with a 'while' in between. Should second 'stage 2' be 'stage 8'?
- 366. Function is not shown, but indicated. This is true for accessibility. For H3K27ac a correlation is shown.
- 373-376, 6B. Is this about the proximal plus distal regulatory elements? How is defined that particular genes do not carry regulatory elements? They can also be in the proximal promoter, downstream, within the gene, further away; they may be active in tissues not examined, etc. If a gene is not constitutively expressed it is regulated by REs.
- Figure 6B: It is unclear to us if trends seen in gene expression and enhancer accessibility can really be considered as correlated. Especially from an average analysis. An analysis per target gene would be more relevant than a general trend comparison.

Discussion

The discussion needs to be revised based on the comments made on the main text.

We made just a few comments below:

- 396-398. There are more genome-wide studies on regulatory elements in plants than the one cited here. See also our comment above. Please place the findings in a broader perspective.
- 399. Indicate the mapping is defined on DH sites.
- 404. Intergenic should be changed to more than 1kb upstream of TSS.
- 408-413. Compactness of different genomes does not explain why a specific histone mark would be lacking an association with distal enhancers in Arabidopsis.
- 417. H3K27me3 is positively associated with poised enhancers and negatively associated with active enhancers.
- 439. The functional relevance between H3K27ac and open chromatin is quite well established. At least in other systems. The cascade of events and general mechanisms leading to transcription in Arabidopsis is expected to be similar.
- Title 451. "hidden driver" is a bit strong. Reformulate.
- 463-465. Sentence is stating a well-established concept. Please refer to relevant papers.
- 470-472. The sentence suggests this has been shown in this study while it has not been demonstrated.

Figures

- 1B. Arrow heads indicating TSS are very small.
- Figure 2A. Why are the groups not numbered in ascending/descending order? Or based on the nr of sites within a group.

- Figure 2C. Please indicate the TSS and mention what the bars below the peaks represent (we assume the peaks called by X program). Indicate in the legend that examples are shown from different groups depicted in A.
- 3C: bewteen->between
- 3E: What is the meaning of blue versus red?
- 4B. How is the density calculated? 25% of the peaks are distal. If the peaks would have the same strength, the average density of the distal peaks would be lower than those close to the TSS.
- 4C, panels on the right. H3K9ac is enriched downstream of the TSS, see Figure 1C. So when examining proximal DHSs, you should see a peak of H3K27ac on one side, rather than two sides. You may have forgotten to orient the data relative to the transcription direction. It is not clear what this panel adds to the figures/panels already shown.
- 4E. The yellow for H3K4me2 in rice is quite difficult to see.
- 5B. The first D in the legend should be B.
- 5C. Enhancers should be changed into putative enhancers, also in the Legend.
- 5F. What is meant with below and above parts in F? Scores for individual DHSs are not visible. The legend does not seem to fit the figure panel!
- 7C, how should the multiple line features be read (3, 5, 7)? What is the difference between 4 and 9, etc? It seems 8-11 only picks out particular genes? But this is not mentioned anywhere. Please add a detailed description of the figure in the legend.
-
- Suppl Fig3A. Binding intensity is not the appropriate wording. H3K27ac does not bind. 'ChIP signal intensity' is one alternative.
- S4: 'related to Fig 1' should be between brackets, and not be part of the title.
- S5, legend. Accumulative fraction of which peaks?
- S6: Peaks from all the four stages were firstly merged for reference. Then the peaks from a specific stage were counted based on the merged peaks. Legend has to be clarified.
- S8C. This panel is not very clear. It seems that numbers and % are mixed. Percentages could be added in a separate table?
- S10B is indicated as rice data in the legend. I however think it is showing Arabidopsis data. Similarly, S10C would show human data, but the panel itself indicates leaf and flower. Are the legends of S11 and S10 swapped?
- S11. Same comment as for 4C. Is the data oriented relative to transcription direction? It should.

Tables

- Suppl Table 1: explain where the score refers to.
- Suppl Table 2-Motif Analysis: why are there NA's (Not applicable?) in the RNA-seq column?

Typos/textual/reference issues

- 27. 'With' should be added before both.50. activate should be activate.
- 31. Would use another word than 'derived'. Derived does not seem appropriate. Also holds for rest of the text.
- 50.to activate
- 51.enhancers are devoided
- 53. Mbs
- 53. 'thus', should be deleted. There is no causal relationship with the previous sentence.
- 56. Similarly, 'accordingly' should be deleted.
- 61-62. When saying 'More and more reports' one should list more than 1 reference.
- 68. Leading to the discovery
- 74. Of enhancer's function
- 75. Have should be has. Or analysis should be changed into analyses.
- 93. 'that' should be 'which', or rephrase it otherwise.
- 93.Here, we took advantage of flower formation as an ideal system...

- 103. 'does' -> 'is'
- 108 largely -> highly
- 109. To drive -> , which is associated with
- 118: add 'a' before dynamic. And add an s after modification. There are a lot more of these errors in the text. Missing 'the', plural where it should be singular or vice versa, etc. Sometimes even a verb is missing. I won't indicate the rest. Please track them yourself and correct the text. It hampers smooth reading of the text quite a bit. A few more are listed below, but there are several more in the text.
- 132-133. at all of the four developmental stages
- 134. 'Regulatory architecture': we would use more explicit terminology: H3K27ac and DHS profiles.
- 137. Coordinating assumes a causal relationship, which your experiments in Figure 1 do not show.
- 139. 'the regions' should be specified.
- 141. 'genetic regions': the entire genome is 'genetic'. This sentence is not complete. It lacks a verb ("are" in between 'sites' and 'located').
- 143. Specifically, of the approximately 6000 and 11160...
- 145. Apart should be away.
- 148. Reference to 1C is not correct. No distal peaks are visible.
- 160 & 187-188. The authors state they examine the role and contribution of H3K27ac to stage specific gene expression. The experiments performed examine correlations, not more than that. Mutants in H3K27ac are not used. The text has to be adapted.
- 184-186. It is not clear why 'it was not surprising....'. Would delete this start of the sentence. Alternatively you have to explain why it was not surprising.
- 194. Maser?
- 213-214. Sentence is incomplete/incorrect.
- 217-218. Incorrect sentence.
- 224-225. H3K27ac peak centers mostly presented...? Second part sentence also lacks a verb.
- Figure 4B. Why is the upstream peak not visible in 2C? The axis extends up to -3kb and upstream peaks at -1.6kb should therefore be visible on this figure!
- 228. What is different with the data shown in FigS5? That figure already shows distances between DHSs and H3K27ac peaks.
- 229-235. Twice the same message is written down.
- 4C, legend: above parts, below parts...should be top panels, bottom panels.
- 292: at all of the four stages
- 297-298. ...occurring in promoters defin
- 298 predicated -> predicted.ed by....?
- 365. Charting?

Reviewer #2 (Remarks to the Author):

Overall the study appears to be quite strong and its conclusions fairly well supported. It will be a good resource for others to use in the future as and when ChIP-seq and expression data becomes available for other TFs. There might have been some question with inferring the TF binding through motif but the strategy to narrow down the TF motifs being observed seems sound. The major issue appear to be the clarity of some of the figures that are structured or labeled in a way that might not be immediately clear to some of the readers. It might be nice to have some more in depth ideas about why there were discrepancies between some of the paper's major findings and previous literature. You could also provide a short explanation of how your seedling assays are suitably comparable to your earlier floral data and why you didn't just use floral tissue all around. I understand that it is a lot more work in generating all these data sets in *ap1;cal1*, but at the same time, comparisons made with leaves and other tissues may lead to mis-interpretations. Would

chromatin conformation data matched to the other Seq studies you performed would be more informative as opposed to obtaining it from other studies? It would be interesting to see more possible future directions discussed. For example, why H3K27ac does not mark enhancers? H3K27ac and DHS may cooperate together in determining expression. How do they come together/communicate? Do you have any plans on how to strengthen the support for your conclusions even further or new directions on which to take your findings?

Comments

1. Even though you reference Hi-C analysis, perhaps matched chromatin structure assays could be informative along with the RNAseq, DNAaseseq, and ChipSeq.
2. Line 151-151 if most H3K27ac sites remain static throughout development and only a fraction are stage specific what could the function of most of them be?
3. Line 167 Could you be a little more specific about just how well do genes from each cluster track with its associated H3K27ac binding pattern? Do the majority of genes of every single cluster follows the H3K27ac pattern?
4. Line 184 What would be the reasons DHS and H3K27ac biologically combine together for higher expression in linked genes than DHS alone. Is the same true for H3K27ac alone?
5. Line 201 Selected motifs may be show certain patterns but do you think this is enough to generalize to developmentally important factors in general?
6. Line 217 what do you think is a reason for the discrepancy between the previous study and your data concerning H3K27ac? Maybe the discrepancy is due partly to the stages you chose to process or different tissues that you used.
7. Line 230 Perhaps the reason why there so many H3K27ac sites are near DHSs but not the other way around is simply DHSs are more widely distributed and common?
8. Line 247 Is there enough reason to believe the seedling tissue is directly comparable to the flower tissue results?
9. Line 272 Could it be clarified whether the 1kb cutoff for intergenic DHSs is arbitrary?
11. Line 276 So having both proximal and distal DHS sites implies that the distal site is an enhancer since the proximal DHS site represents open chromatin around the gene while the distal DHS is open chromatin around its associated enhancer?
12. Line 290 The correlation between motifs and open chromatin is interesting but its hard to be sure without more direct evidence. That is, stage specific CHIP data of the TF.
13. Line 299 If you later say that enhancers are under increased conservation, what would you say is the biological significance of increased SNPS in the promoters and enhancers?
14. Line 305 By surrounding regions do you mean enhancers or just surrounding DNA in general? If you mean non-enhancer DNA of little to no epigenetic function, what would be the explanation of promoters being less conserved than that?
15. Line 320 conformation instead of confirmation?
16. Line 323 Based upon the Hi-C analysis, most but not all the enhancers were associated with their nearest genes. Maybe it would be better to partition how you handle the enhancers based upon this study rather than assume all operate on the nearest neighbor.
17. Line 328 Maybe more than just two significant clusters could have been extracted if the parameters of the clustering algorithm were set differently. There seems to be a lot of variety in how those enhancers are behaving across the different stages within each of those two clusters.
18. Line 376 When you are referring to the percentages of 20.6% and 33.6% it appears to be unclear whether you are referring to flowering genes, all genes, or TF genes.
19. Line 454 corrected->correlated?
20. Line 427 Could you go into detail about how your classification of the DHSs might be more preferable than Zhu's?

Figure Comments

1. 2B Do all 10 clusters of H3K27ac behavior have biological significance? Only a fraction are discussed in the paper.
2. Fig3C Figure is confusing and could use more clear markings. They appear to show the DHS and H3K27ac peaks but what does the x axis represent? Are they just all 805 peaks lined up with the DHS results between Stage 2 and 8?

3. Fig3E Were these transcription factors the only ones examined for enrichment? How were certain ones chosen for displaying their motif logos?
4. Fig 4 The scale for D, E, and F is the same as that for C? (mentioned in text as the same)
5. Fig 5b is mislabeled in the legend as 5d. .
6. Fig 6A: would you happen to have an explanation why there are much more genes active in the later stage? (Cluster 2) rather than the active in the earlier stage? (cluster 1)
7. 6B Could you elaborate on why RPM and FPKM are ideal for the particular cases in which they are used? From the figures saying perfectly coincided may be a little too strong since there looks like there are very slight discrepancies between the trends of some of the boxplots.
8. 6c Do you have examples of several other genes where the DNAase and TF binding results line up as well as this sample that are good enough to be main figures?
9. 7C is somewhat confusing because it is difficult to understand what each of the circles represent and how they relate to the list in the middle.

Reviewer #3 (Remarks to the Author):

The paper entitled "Dynamic control of enhancer activity drives stage-specific gene expression during flower morphogenesis" describes the genome-wide localization of histone H3 lysine 27 acetylation (H3K27ac) and compares the data with DNase I hypersensitivity sites (DHSs) and RNA-seq at four representative stages of Arabidopsis flower development. They show that H3K27ac dynamics and DHSs are highly correlated with stage-specific gene expression. They also claim that DHS-derived enhancers are highly stage-specific and associated with SNPs in GWAS for flowering-related phenotypes. The same group previously reported the relationship of DHSs, gene expression and binding of two MADS-domain transcription factors APETALA1 and SEPALLATA3 in Arabidopsis flower development using the same floral induction system (Pajoro et al. *Genome Biology* 2014, 15:R41). The other group reported genome-wide analyses of DHSs and H3K27ac, and showed by the reporter assay that ten of the 14 (71%) candidate enhancers with distal DHS really function for leaf- or flower-specific expression (Zhu et al. *Plant Cell* 2015, 27: 2415–2426). This study by Yan et al. provides thorough analyses of genome-wide dynamism of H3K27ac in flower development and its correlation with DHSs, expression profiling and genome-wide binding profiles of representative transcription factors. Both papers of Zhu et al. and Yan et al. generally came to a similar conclusion, but Yan et al. further conclude that distal enhancers do not contain H3K27ac from series of genomics analyses. My major concern is that this manuscript does not contain any biological evidence to show the regions actually act "enhancers" nor any new positive correlation with other histone marks.

In summary, this manuscript brings together a useful information for some specialists in the community of flower development. I believe that more specialized genomics journals could be a better home for this paper.

Detailed comments:

- 1) It is very confusing to use the term "stage" to describe the days after dex induction, since their stage 0, 2, 4 and 8 do not seem to perfectly correlate with the conventional flower developmental stages in Arabidopsis. At least it is not clear from the images shown in Fig. 1A.
- 2) In page 5, line 123-123, "2 days after induction, stage 4-5" should be "4 days , , , ,"
- 3) In Fig. 1B, the AP1 locus is shown as a representative gene to show the correlation between H3K27ac, DHS and expression. Please justify to study the endogenous AP1 locus in the ap1 mutant background with an inducible construct of AP1.
- 4) In page 10, line 275, "that" should read "than".
- 5) In figure 1 legend, line 2 from the bottom, "Promoters were defined as 1kb upstream and downstream of the TSS." does not make sense to me.
- 6) In figure 5 legend, line 4, "D" should be "B."

Reviewer #4 (Remarks to the Author):

The manuscript by Yan et al., provides solid evidence that enhancers activity is driving stage-specific gene expression. The methodology used is adequate, the results solid and the figures neatly presented. I only have comments concerning the text itself that I believe need to be addressed:

A source of concern comes from the lack of description of the plant material used to perform the various assays. On line 484 the authors state that pAP1:AP1-GR ap1-1 cal-1 plants were used for mapping stage-specific targets of H3K27ac in flowers, while on line 120 the authors refer to a previously described floral induction system that makes use of a 35S:AP1-GR ap1-1 cal line (ref 33; I believe that reference 34 should have been used here). This needs to be clarified, as the use of a strong constitutive promoter would be inappropriate to conduct the work described in the present manuscript and would alone justify rejection. In addition, there is no mention of the plant material used for the RNA-seq and DHN-seq assays in the Plant Material and Growth Conditions paragraph. Annoyingly, the authors ended up including this information in the subsections describing the respective assays.

In its present form I don't see how this manuscript deepens our knowledge of flower morphogenesis other than providing several transcriptomes and landscapes of histone marks and DHSs at several stages of flower development. The manuscript fails to interpret the genomic data in the context of flower morphogenesis. For instance, why is it important for an Arabidopsis plant to have a high frequency of regulatory elements occurrence in flowering genes than on any other genes? On line 386 the authors reported that "Strikingly, binding sites of MADS-domain containing proteins, represented by the master regulators of flower development, were overrepresented in both the proximal and distal DHSs" but failed to discuss the implication of this result. In my opinion, the Discussion should be rewritten in order to better bridge the genomics findings reported in this study and the biological aspects behind flower development. Alternatively the title could be modified by omitting the words "during flower development", which would better reflect the focus of this manuscript on understanding the role of enhancers in regulating stage specific gene expression in plants.

Minor comment:

Line 320 confirmation should read conformation

Reviewer #5 (Remarks to the Author):

The paper by Yan is well written and the analyses performed are mostly sound.

The authors use H3K27ac signatures and RNAseq combined with prior analyses such as DNase HS to try to define possible enhancers in Arabidopsis during flower patterning. They find that H2K27ac associates only with promoter proximal DNase HS regions, while distal DNase HS regions are devoid of this (or other) histone modifications. The authors go on to suggest (based on DNA sequence conservation, presence of predicted TF binding sites and link to DE genes) that the distal DNase HS sites – but not the proximal ones- are likely to be enhancers.

The study is original, but in its current form not yet fully convincing. My main concern is that none of the newly identified proposed distal enhancers were functionally tested, this could easily have been done with reporter constructs. This needs to be remedied in a revised version.

More specific comments

1. Changes in the level of histone modifications such as H3K27ac could be triggered by addition or removal of the acetyl groups to H3, or by histone octamer removal/turnover. To avoid looking at H3K27ac changes that simply reflect altered nucleosome occupancy, the H3K27ac levels need to be normalized over an independent H3 histone ChIP. This is critical, as TF binding can lead to nucleosome loss.

2. Computationally predicted TF binding sites are not very useful for predicting factor binding. Only a very small fraction of these are bound in vivo. Most are small and occur on average every 2 kb in the genome. Because of that they are not a good metric for defining enhancers or DNase HS site function as enhancers. If they are to be used, genome-matched random regions (non DNase HS sites at similar genomic locations) should be used to test for significance of enrichment of predicted TF binding sites in the putative distal enhancers.

3. The authors use a large number of prior studies (for CNS for example), but do not always properly highlight in the main text that these were prior studies.

Reviewers' comments:

Response to all reviewers: We thank all the reviewers for your useful comments and suggestions. We have carefully considered the remarks from all reviewers and we respond to them in a point-by-point manner. In the revision, we have added experimental validation results for predicted enhancers, and extended the bioinformatics analyses to focus more on the dynamics and functions of enhancer candidates. We have re-arranged our manuscript and rewritten the discussion part. Our response begins with “Response:” in blue.

Reviewer #1 (Remarks to the Author):

This study aims at identifying enhancers involved in stage specific gene expression during flower morphogenesis. Authors discuss the relationship between open chromatin, H3K27ac and gene expression and establish a correlation between proximal H3K27ac enrichment and gene expression as well as a correlation between chromatin accessibility at distal DHSs and expression of putative target genes.

The attractiveness of the study lies into the use of a flower induction system. The study shows interesting results on distal DHSs showing characteristic signs of regulatory sequences. Nevertheless, it is conceptually not novel enough to be published in Nature Communications. Especially, the presence of H3K27ac at genic regions and its absence of enrichment at enhancer candidates are results that have been reported before. Only the second part of the analysis really focuses on enhancer candidates but lacks to report new concepts on plant enhancers. We do believe that after having taking into account our comments the paper is well suited to be published elsewhere.

In addition, the text is loaded with typos of various kinds. Inappropriate use of articles, lack of articles, plural/singular, words missing, use of incorrect words and other typos. This hampers the reading and in addition gets irritating at a certain moment. It would be good to use the spelling checker and have others proofread a manuscript on typos before sending it in. That allows a reviewer to fully focus on the content. Towards the end of the

sections below, only the more prominent issues are being listed. Please check the rest of the document for typos yourself.

Review

- 50-51. This sentence is not entirely correct. The binding of pioneer TFs results in chromatin remodelling that then result in nucleosome depletion and the binding of more TFs, etc

Response: We agree with the reviewer that pioneer TFs function in depleting nucleosomes and thereby making the chromatin accessible for other TFs. In line 50-51, we only talked about “active” promoters and enhancers, which are already accessible for TFs to bind. However, we rephrase this sentence as "*Active promoters and enhancers are devoid of nucleosomes thereby rendering the DNA accessible for TF binding*".

- 53. Thus indicates a causal link with the previous sentence. In this case there is no link between enhancer distances from the TSS and how many enhancers are controlling a target gene.

Response: We thank the reviewer for pointing out this issue. We have deleted the word “Thus” in this sentence.

- 59. H3K4me1 in animals is associated with both active and inactive enhancers, not just active ones.

Response: We thank the reviewer for pointing out this fact. To be more accurate, we have removed the word “active” in the sentence.

- 65. “In this regard” refers to diseases. Reformulate!

Response: We have deleted “In this regard” in this sentence.

- 71. Not sure if extrinsic cues can be counted as drivers of lineage specific transcription programs. Either remove ‘lineage specific’ or replace by ‘specific’.

Response: We have removed “lineage specific” in this sentence. The corresponding sentence in the revision is "*Enhancers appear to function as integrated platforms for*

binding of multiple TFs, often including lineage-determining, intrinsic cue-dependent and signal-, extrinsic cue-dependent TFs^{2,15,16},

- 82. The reason provided is not the reason why enhancers in plants are poorly characterized. There have hardly been any genome-wide studies focusing on the characterization of enhancers in plants. Some plant scientists, even believe (d) there are no enhancers in plants. On top of that, most studies in plants use Arabidopsis as a model system. The genome of Arabidopsis is extremely compact, hampering the dissection of ‘the promoter’ into promoter-proximal sequences and enhancers.

Response: Thank you for this comment. We removed the sentence from the text. Regarding the question about existence of plant enhancers: indeed, due to the compactness of the genome, it is difficult to identify enhancers in Arabidopsis. However, we hope that both the previous study by Bao et al. and our study here provide evidence that there are distal regulatory elements existing in Arabidopsis and these elements are important for spatiotemporal gene regulation, especially with our time series data provided here.

- 85-86. There are a number of studies missing focused on the prediction of enhancers in plants that should also be acknowledged:

- o 1) Zhang W, Zhang T, Wu Y, Jiang J. Plant Cell. 2012;24:2719–31.

- o 2) Furthermore, there is similar study in rice: Zhang W, Wu Y, Schnable JC, Zeng Z, Freeling M, Crawford GE, et al. Genome Res. 2012;22:151–62. In the latter they looked at various different histone modifications (but not H3K27ac).

- o 3) recently published stories are missing as well: Eli Rodgers-Melnicka, Daniel L. Verab, Hank W. Bassb, and Edward S. Buckler, Open chromatin reveals the functional maize genome; doi:10.1073/pnas.1525244113

- o Oka R, Zicola J, Weber B, Anderson SN, Hodgman C, Gent JI, Wesselink JJ, Springer NM, Hoefsloot HCJ, Turck F, Stam M. Genome Biology 2017; 18:137. The latter used a combination of DHS, H3K9ac and DNA methylation to predict active enhancers.

Response: Thank you for this remark. We have added the information from these publications as well as other related reports in the third paragraph of the introduction part. Please see page 4 (line 7) in the manuscript text for details.

- 102. H3K27ac is not the hallmark for ‘active’ enhancer annotations...(add ‘active’)

Response: Thank you for the suggestion. We have added the word of “active” to make the description more precise.

- 116: Zhu, Zhang et al, Du et al, and Charron et al do not correlate H3K27ac with predicted enhancers. Plus, saying this is in contradiction with the sentence at lines 88-90.

Response: We intended to say that 1) H3K27ac is correlated with active gene transcription, which had been shown by Zhang et al, Du et al, and Charron et al, and 2) H3K27ac is associated with predicted enhancers in Arabidopsis, which had been declared by Zhu et al. We do agree that the way we cited the references in the previous version was misleading. We have changed this sentence as “*H3K27ac is a primary epigenetic mark for active enhancers in animals^{1,3,38} and it was recently shown to highly correlate with active gene transcription in plants³⁹⁻⁴¹*”. Please see the reply to the next reviewer comment for more details.

- What is the reason to focus on H3K27ac and not on H3K9ac or other histone marks associated with active enhancers? As mentioned in the introduction, both seem linked to enhancer activity in animals. In addition, evidence for H3K27ac not being involved with active enhancers in Arabidopsis was already there (Zhu et al). So why further investigate this situation?

Response: The reasons for which we took H3K27ac but no other histone marks were that H3K27ac was described to be associated with Arabidopsis enhancers and surrounding regions and to be positively correlated with tissue-specific enhancers by Zhu et al.

We got this information from Figure 2B-E and Supplemental Figure 1 as well as by reading interpretation of these results at page 2 of Zhu et al. [Zhu et al, 2015 Plant Cell]: “To investigate the possible association of these histone modification marks with enhancers in *A. thaliana*, we developed chromatin immunoprecipitation followed by sequencing (ChIP-seq) data

sets for H3K27ac and H3K27me3 using both leaf and flower tissues that were at the same developmental stages as the tissues used for DNase-seq (see Methods). We examined ChIP-seq profiles along 5-kb regions flanking each predicted enhancer. These surrounding regions were clearly enriched in both histone modification marks (Supplemental Figure 1A). In contrast, the same regions were depleted of H3K27me1 and H3K9me2, two classical marks associated with heterochromatin in *Arabidopsis* (Luo et al., 2013) (Supplemental Figure 1A)... We then examined the histone modification patterns around tissue specific enhancers. The predicted leaf-specific enhancers and flanking regions were more enriched in leaf tissue-derived H3K27ac than in flower tissue-derived H3K27ac (Figure 2B). Similarly, the predicted flower-specific enhancers and flanking regions were more enriched in flower tissue-derived H3K27ac than leaf-tissue-derived H3K27ac (Figure 2D). Thus, H3K27ac was positively correlated with tissue-specific enhancers (Supplemental Figure 1B).”

Since we aimed at studying the control of gene expression by dynamics of enhancer activity in *Arabidopsis* flowers, inspired by the results provided by Zhu et al, we chose H3K27ac for facilitated mapping of active enhancers. Not only our temporal analysis of this mark in the developing flower drove insights into its dynamics (which Zhu et al study could not bring, dealing with mixture of tissue types and stages), but we also further found that H3K27ac hardly locates at distal intergenic regions. Zhu et al. found a correlation (actually a very weak correlation; Fig. 2B and C) between enhancers and H3K27ac mostly due to a broad window (in size of 5 kb) used in their analysis. The window size may not be specific enough because the genome of *Arabidopsis thaliana* is very compact with the average size of ~3kb in intergenic regions. In fact, our analysis revealed that less than 3% of H3K27ac sites were mapped >1kb away from the TSS and there is no enrichment of H3K27ac at distal DHSs (Fig. 2a-c).

Fig. 2a-c. (a) Distribution of the distance of H3K27ac peaks from the transcript start site (TSS). (b) Distribution of the distance of DNase I hypersensitive sites (DHSs) from the TSS. (a,b) Note that absolute value was used to calculate the distance and the x-axis is log10-transformed. (c) H3K27ac and DNase-seq read intensity relative to proximal and distal DHS midpoints. All data were obtained at stage 0. The direction of the nearest genes was taken as the DHS's orientation. Top parts, composite (average intensity) plots; bottom parts, heatmaps of read distribution per DHS.

- 123-124. We assume stage 4-5 corresponds to 4 days after induction instead of 2 days as indicated in the manuscript.

Response: Yes, it should be “4 days after induction”. We apologize for the mistake and have corrected it.

- 137-139. The examples in Figure 1B show highest expression in stage 4, while H3K27ac and DHS signals were highest in S8 for AP3 and relatively static for AP1. The sentence does not reflect this observation.

Response: Thank you for the suggestion. To make it more clear, we re-wrote the sentences: “*For example, the dynamic changes of H3K27ac and DHS profiles of the key floral regulator genes APETALA1 (AP1, a floral meristem identity gene) and APETALA3 (AP3, a floral organ identity gene), across flower developmental stages are correlated with their expression dynamics. The AP1 locus is enriched in H3K27ac and stays open across all of the four timepoints, in link with its continuously high expression, while the AP3 locus shows more H3K27ac-enriched and accessible chromatin at later stages (4 and 8 days after induction), which correlates with increase in AP3 expression (Fig. 1b).*”

- 141. ‘genetic’ should be ‘genic’. ‘towards’ should probably be replaced by ‘directly downstream of the’ as towards does not reflect any notion of distance.

Response: Thank you for pointing out the mistakes which we have corrected. We have changed “genetic” into “intragenic” and “towards” into “directly downstream of”.

- 142. Indicate Figure 1D as well.

Response: We have included the citation of Figure 1D in this text location.

- 143. Why focus on only those two categories (stage 2 and 0)? It suggests the situation would be different for other stages while it does not seem to be the case.

Response: Sorry for the misleading sentence. We intended to say that peaks identified from all the stages locate in gene body (only the number of significant peaks varies). We now erased this misleading information and re-wrote the sentence into “*In detail, out of all the high quality H3K27ac peaks identified at different developmental stages*”

(Supplementary Table 1), more than 95.0% were found in gene bodies, whereas only about 2.5% were mapped to distal intergenic regions (>1 kb from the TSS; Fig. 1d - left panel)."

- Figure 1C. The legend does not specify the % clearly enough. Do you mean the expressed genes were divided in equal groups (equal nr of genes) or are particular thresholds taken associated with particular expression levels?

Response: Thank you for pointing out this issue. We meant to say equal number of genes in each group. We have made this clear in the figure legend.

- Figure 1D. How are the peaks defined and assigned to specific genomic regions? This needs to be further detailed in M&M. What if peaks overlap multiple categories? For example if the CDS and intron are part of a region showing H3K27ac, is this region assigned to one of them or split in two categories?

Response: Only one category is considered if a peak overlaps with multiple categories based on the following priority: 5' UTR > CDS > introns > 3' TUR > promoters > distal intergenic regions. We now stated the strategy in more details under the section "Genomic distribution of peaks" of the Materials and Methods section.

- Fig1E. Is this referring to all peaks or only distal peaks? If referring to all, how does the data look like for distal peaks specifically? And for the promoter/CDS peaks?

Response: The old Figure 1E is referring to all peaks. We used the same color code as in Figure 1D, which could lead to confusion. In this revision, we have moved this part to Supplementary Fig. S7a due to a reorganization of the content, since our main aim now is to analyze the function of enhancer dynamics in developmental stage-specific gene regulation.

- 152. What do you mean with 'highly induced flower tissues'? Please specify. And how does this fit with the statement that they gained more stage-specific H3K27ac sites? There are more unique peaks in S0 than in S2-S4. I would expect S8 (or S2 or S4) to be the highly induced tissues, instead of S0.

Response: by “highly induced flower tissues” we meant “later stages after induction” but we do agree that this formulation is not clear. Here, we only focused on the developmental stages after induction (from S2 to S8). Sorry for the misleading. We thank the reviewer for pointing out this lack of clarity. In order to focus more on the enhancer analysis, we shortened this section and the corresponding figures: *“Furthermore, we found that although most H3K27ac sites remained stably enriched during flower development (Supplementary Fig. 7), 23.2% (2913/12579) of them displayed at least a 1.5-fold change in H3K27ac peak intensity within the four time points.”*

- 153-154. Is likely that the correlation between H3K27ac and expression is not just true for flower tissue. Would leave ‘flower development’ out in this sentence.

Response: Thank you for the comment. We changed the sentence to *“Together, our results support and extend previous research³⁶, showing that H3K27ac is highly correlated with gene activation during floral meristem specification and organ development”*.

- 154-156. This has been observed by others as well. See e.g. papers cited above. In addition, the function of open chromatin and acetylation marks in transcription is well studied. The sentence suggests this is not the case.

Response: In the mentioned publications, we do not see the results that H3K27ac and DHS have very different patterns of locations in intergenic regions. In Zhu et al, they described that H3K27ac is enriched in “5-kb regions flanking each predicted enhancer” and that H3K27ac is positively correlated with tissue-specific enhancers.

We agree that the correlation between open chromatin/acetylation marks and active gene transcription is well studied. However, we found that across flower development, H3K27ac hardly appears in distal intergenic regions where around one quarter of DHSs locates. This indicates that H3K27ac and DHSs play different roles in activation of gene expression. In order to more clearly express this idea, we rewrote the sentence as *“Besides, the difference in localization of H3K27ac sites and DHSs with respect to the TSS, especially in distal intergenic regions, indicates that they may activate gene expression via distinct mechanisms.”*

- 167-168. The text indicates that the dynamic enrichment and change in gene expression levels generally follow a similar trend. This seems to be true for some examples (SVP and SEP3 for instance) but clearly not for all. Rather than showing cherry picked examples, it would be good to show, as presented in Figure 2B, the trend in expression for the different clusters. In addition we question the reasoning behind the decision to have 10 clusters. Some seem very redundant (cluster 2, 4 and 6 for instance).

Response: The dynamics of every differentially expressed gene in each cluster is shown in old Figure 2A while old Figure 2B describes the general trend of H3K27ac change in each cluster. Regarding the old Figure 2C, it was meant to provide details of H3K27ac modification and expression for some examples to assist with understanding the figures. Due to a reorganization of the content in the revision, we have moved the whole old Figure 2 to Supplementary Fig. 8. Following the reviewer's suggestion, we have removed the old Figure 2C and plotted the trend of expression change together with the H3K27ac dynamics in Supplementary Fig. 8c. The optimal number of clusters was calculated based on three different methods and was set to 4 (see below and also Supplementary Fig. 8a). Accordingly, we have rewritten this part (see page 6) based on the new Figures.

Supplementary Fig. 8: Dynamically changed H3K27ac-associated regions across flower development. (a) Determining the optimal number of clusters. Three different methods (elbow, Silhouette and gap statistic) were used to compute the optimal number of clusters ($k=4$ in all cases as highlighted in red). (b) Heatmap showing the clustering analysis of 2913 H3K27ac peak regions that dynamically change more than 1.5-fold across the four stages (from stages 0 [S0] to 8). These regions are clustered into four groups (in different colors) with distinct patterns of H3K27ac enrichment. The blue bars on the left of heatmap indicate the nearest DHSs for the corresponding H3K27ac peaks have more than 1.5-fold change (dDHS). The pink bars indicate the nearest genes are differentially expressed (DE). Flowering genes are labeled on the right of heatmap. (c) Summary of H3K27ac enrichment in the four groups as colored in (b). Grey curves indicate

the median scaled expression level of genes in the corresponding groups. Correlation coefficient values between H3K27ac signal intensity and gene expression are shown. (d) Gene ontology (GO) enrichment analysis for each gene clusters as represented in (b).

- 176-184. The messages in this section are contradictory. Correlation between H3K27ac and expression is first described as positive (174-176), then questioned given the result of some specific clusters (176-181) and then positive again (182-184). In addition, we miss the data on correlation between DHS and expression. This cannot be deduced from Figure 3A.

Response: The overall correlation between H3K27ac enrichment and active gene expression is clear but we also observed that in some individual cases, the trend is not fully followed, and initially pointed this out in order to be objective. However, we think that these individual cases do not change the general conclusion that H3K27ac is highly correlated with active gene expression. For the sake of clarity and upon this reviewer remark, we thus decided to convey a simpler message in the new version of the manuscript. This message is strongly supported by the Pearson's correlation coefficients we provided (Supplementary Fig. 8c). We added a likely explanation in the revision: *“The occasionally observed differences between changes in H3K27ac and in gene activities (e.g., for genes in groups 2 and 4; Supplementary Fig. 8c) can be partly explained by the fact that the data were not generated from individual cells or cell types, but correspond to ‘average’ signals from whole meristems or organ differentiation stages. For example, strong gene activation in some of the cells may be associated with an overall reduction in H3K27ac levels, if in most other cells the gene is not activated or becomes repressed.”*

- 178. You measured RNA steady state levels, not transcription (would need to do Gro-seq experiments or something similar). So you cannot use the word transcriptional. The use of the word transcription should be changed throughout the text.

Response: We do agree with the reviewer that transcription is not the proper word for commenting our results. We have used the word of *“expression”* to replace *“transcription”* throughout the text.

- 189-191. Why between stages 2 and 8? You forget to mention this is done for sites that are accessible, rather than all sites.

Response: Thank you for the remark, indeed we compared only the loci that are open. Since we re-organized and shortened the manuscript in order to focus more on the enhancer analysis, this sentence has been eliminated in the new version of the manuscript.

- 193-194. The authors first have to mention where the ChIP data on the TFs is coming from.

Response: Thank you for the comment. The sentence to which this reviewer comment refers to has been eliminated from the text. We nevertheless use the TF ChIP-seq data in the enhancer analysis, and references for the datasets were comprehensively cited in the Methods section (“ChIP-seq data analysis”). The references from which we obtained the ChIP-seq data are mentioned after each protein name.

- 194. Maser? You mean master?

Response: Sorry for the mistake. It should be “master”.

- 194-198. The text mentions that the ChIP-seq datasets had been generated with plant tissues of different developmental stages. Still there is a statement saying that proteins functioning mainly at early stages are mainly bound to stage 2 specific DHSs. It is however unclear if the ChIP-seq has been done in stage 2 and stage 8, so if this statement can be made to start off with. Looking for TF motifs is in no case an indication of TF binding and in a sense assuming binding of a TF due to the presence of a specific motifs could be seen as more bias than exploiting ChIP seq data in different tissues.

Response: Thank you for the comments. We have to point out that, due to rearrangement of the manuscript, we omit this analysis with H3K27ac sites in the revision. However, we agree that difference in tissue or stage is a problem. In the new version of the manuscript, we only consider AP1 and SEP3, for which stage specific binding profiles are available (obtained by us in a former study). We compared the target sites of these two regulators with stage-specific distal DHSs and found that AP1 and SEP3 bind to stage-specific

enhancers in accordance with dynamic chromatin accessibility (see Fig. 6f in this revision).

We agree with the reviewer that the existence/overrepresentation of a certain motif does not prove the existence of an *in vivo* TF-binding event. We found that the enrichment of predicted TF binding sites in proximal and distal DHSs is significantly higher than that in their flanking regions or random intergenic non-DHS regions (Supplementary Fig. 13 in this revision). Therefore, our data suggest that when TF ChIP-seq datasets are not available for specific tissue type or specific developmental stages, motif scanning allows to predict candidate TFs that may play roles in stage-specific gene regulation.

- 209-210. ‘Interestingly, these stage-specifically overrepresented regulators also showed stage-specific expression patterns ‘. For the proteins mentioned in the previous sentence, this only holds for TFPD and TCP15. So ‘These’ is inappropriate. Furthermore, it is required to specify in which direction this stage-specific expression is. The overrepresentation of particular binding sites can be associated with a higher expression of the regulators, but also with a lower expression level.

Response: The reviewer is right that the word “These” used here is inappropriate. In the revision, this sentence has been removed from the manuscript due to a reorganization of the content in order to focus more on the enhancer analysis.

- 211-212. ‘Collectively, our results showed that the dynamic change of H3K27ac at the two analyzed stages largely...’

Response: We agree that it is better to specify that we only draw the conclusion from the two analyzed stages. In the revision, this sentence has been removed from the manuscript due to a reorganization of the content in order to focus more on the enhancer analysis.

- 213. Here proximal DHS sites are mentioned. In the section this is not specified that the sites monitored in this section are proximal sites. Are they all proximal? IF so that needs to be specified at the start of the section.

Response: Thank you for pointing this out. We have reorganized and shortened this section in this revision. To answer the reviewer's question: yes, the analysis was done based on proximal DHSs.

- 217. Again, the message about H3K27ac being associated with active enhancers in ref 28 is not that clear.

Response: We got this information by reading the sentences of *"To investigate the possible association of these histone modification marks with enhancers in A. thaliana, we developed chromatin immunoprecipitation followed by sequencing (ChIP-seq) data sets for H3K27ac and H3K27me3 using both leaf and flower tissues that were at the same developmental stages as the tissues used for DNase-seq (see Methods). We examined ChIP-seq profiles along 5-kb regions flanking each predicted enhancer. These surrounding regions were clearly enriched in both histone modification marks (Supplemental Figure 1A). In contrast, the same regions were depleted of H3K27me1 and H3K9me2, two classical marks associated with heterochromatin in Arabidopsis (Luo et al., 2013) (Supplemental Figure 1A).....We then examined the histone modification patterns around tissue specific enhancers. The predicted leaf-specific enhancers and flanking regions were more enriched in leaf tissue-derived H3K27ac than in flower tissue-derived H3K27ac (Figure 2B). Similarly, the predicted flower-specific enhancers and flanking regions were more enriched in flower tissue-derived H3K27ac than leaf-tissue-derived H3K27ac (Figure 2D). Thus, H3K27ac was positively correlated with tissue-specific enhancers (Supplemental Figure 1B)."*, as well as by observing Figure 2 B-E and Supplemental Figure 1 in Zhu's paper.

- 225-227. From Figure 4A and B it is difficult to understand how DHS peaks can be located upstream of the TSS while they locate in similar position as H3K27ac peaks that are described as being downstream of the TSS. If the absolute value for distances was used, it should be mentioned in the legend.

Response: Yes, we used absolute value to calculate the distance by taking TSS sites as origin of coordinate. We have added this information by "Absolute value was used to calculate distance in a and b" to the figure legend.

- The data shown in S11 for H3K4me2 at distal DHSs in rice are different from the results shown in Zhang et al 2012 pp 151. In Fig 7 they show there is H3K4me2 enrichment around intergenic DH sites (panels D and H).

Response: We thank the reviewer for this comment. We have updated this figure (now Supplementary Fig. 12) by concerning the orientation of the data relative to the transcription direction (as suggested by the reviewer in the next comment). Now we found the results are consistent with Zhang et al 2012: there is indeed H3K4me2 enrichment around distal intergenic DH sites.

- 246-252. Fig 4D, It is unclear to which DHS sites this is plotted. Generated in the exact same tissue as the tissue in which the histone modifications have been examined? If not derived from the same tissue samples one cannot compare these data sets. Also, which DNase-seq data are shown on the vertical axis and which at the ‘X-axes’ (at the top)? See also comments below at Fig4 concerning the orientation of the data relative to the transcription direction. In addition, Oka et al showed that in maize, at distal DHS sites, H3K9ac is asymmetrically localized. You need to check if this is the case as well in Arabidopsis and rice and if so, orient the data such that the highest value is oriented towards only one specific site. The way the data are presented in 4D-F is in addition more suitable for a Supplemental Figure. Here data that are already published are analyses again. E.g. Zhang et al 2012 and Zhu et al 2015 already showed plots like in S11 in their paper. Luo 2013 mentions: ‘This result suggests that the vast majority of histone modification peaks are found within the annotated genic regions rather than the intergenic regulatory sequences for Arabidopsis.’

Response: Thank you for the comments. Both DHS (DNase-seq) and histone modification ChIP-seq data for Figure 4D (Fig. 2d in the revision) are from seedlings as annotated in the parenthesis. For figure E and F, the data used was from rice seedlings and human K562 cells, respectively. In order to improve clarity, we added more information on tissue types in the figure legend.

The reviewer raised an interesting point regarding the orientation of the enrichment of histone modifications. Following the reviewer’s comment, we performed similar analysis as done in maize by Oka et al and we found that the enrichment of H3K9ac and H3K27ac

is asymmetric at proximal DHSs in both rice and Arabidopsis. However, as shown in Figure 2d-g, H3K9ac is hardly co-localized with distal DHSs in Arabidopsis and in rice. By re-analyzing the publically available data, we aim to show in a systematic cross-species comparison that modifications that are co-localized with either proximal or distal DHSs differ in the different model species. This provides information that the histone hallmark of enhancers is not conserved among plant species. Thus, this adds a cautionary note for enhancer mapping in various model systems. We would like to show this information in the main text. Regarding the old Supplementary Fig. 11 (now Supplementary Fig. 12), it was indeed to reanalyze the data that had been published, but we intended to show the data in the context of proximal and distal DHSs, and to systematically compare the patterns in different species (Arabidopsis, rice and human).

- 4G. We wonder if the authors took into account that different antibodies have different kinetics, plus different antibodies will be of different quality. The figure kind of implies one can say there is less H3K4me2 in Arabidopsis and rice, than there is H3K4me3. The ChIP signal levels of different antibodies can however not be compared.

Response: We agree that it is hard to compare the enrichment level of a certain modification in different species, due to the reasons mentioned in the comment. However, the difference in color intensity showed in the old Figure 4G (Fig. 2g in the revision) was only to refer the difference of co-localization ratio between DHS and modification but not to compare the ‘overall’ enrichment level. We focus on the spatial distribution of histone marks in the manuscript text, not on differences in ChIP signal levels.

- 260, S11A. Not sure you can say ‘depletion’ of H3K27ac at distal DHS sites. There are small peaks visible in S11A.

Response: Thank you for pointing this out. We changed the previous sentence to *“however, the Arabidopsis distal DHSs barely exhibited H3K27ac modification in their flanks.”*

- 262. Epigenetic should be chromatin signature. It is not shown that histone acetylation can be inherited through mitosis or meiosis.

Response: Thank you for the suggestion. “Chromatin signature” is a more precise term to use here. We have changed the text accordingly.

- 275-276. Would be useful to mention how many of each category exist.

Response: We thank the reviewer for this comment. We have now included the number of genes in each category in the figures.

- 277-286. It is unclear what the added value of this analysis is. State 2 of ref 46 is also associated with coding sequences. It is associated with very low transcript levels. State 1 with high levels. Both 1 and 2 are associated with promoters. State 4, in which the distal DHSs are enriched with H3K27me3, so polycomb silenced chromatin. So those could represent silenced regulatory sequences, but this has not been tested. The authors should take this into account.

Response: We thank the reviewer for this comment. The aim of this analysis was to further support the distinct chromatin ‘properties’ of proximal and distal DHSs, in order to justify their classification into core promoter and enhancer elements. Since the chromatin states in ref 46 were defined based on data from seedling tissues, they do not imply a similar configuration in floral meristems. The prevalence of state 4 in distal DHSs suggests that they are silenced by the PcG system in seedlings (aboveground mixed tissues). Indeed, the enhancers associated with chromatin state 4 showed significantly reduced H3K27me3 enrichment in floral tissues compared to leaf tissues (analysis was performed using ChIP-seq data from Engelhorn et al 2017 Epigenomes; see Fig. R1-a below). Accordingly, potential target genes that are associated with enhancers of chromatin state 4 are usually downregulated in the leaves compared to the flowers (using RNA-seq data from Zhang et al 2012 Plant Cell; see Fig. R1-b below). In order not to confuse readers because of the different tissues used, we removed this part of analysis in the revised version of the manuscript.

Figure R1. Comparison of H3K27me3 enrichment and gene activity for distal DHSs associated with chromatin state 4. (a) Enrichment of H3K27me3 at distal DHSs associated with chromatin state 4. H3K27me3 ChIP-seq data in leaf and floral tissues were obtained from Engelhorn et al 2017. (b) Gene expression patterns of target genes that were associated with distal DHSs of chromatin state 4. RNA-seq data were obtained from Zhang et al, 2012.

• 286. The authors identified less candidates than have been identified in Zhu et al (ref 28), despite defining shorter promoter regions (1kb instead of 1.5kb). Comments on why this is the case appear in the discussion, but an analysis of the overlap between ref 28 candidates and the ones identified in this analysis would be a good addition to the paper. Also, Zhu et al experimentally validated some of those enhancers. Having a large overlap between the two datasets would be a good way to support the quality of the prediction in this paper.

Response: Thank you for the suggestion. Zhu et al. considered the DHSs with distance > 1.5 kb from the TSS (regardless their peak centers) as enhancers, although some of these “distal” DHSs may still locate within 1 kb apart from the TSS because some DHSs can span several kilobases. However, we predicted much less intergenic enhancers than Zhu et al., mostly due to the different criteria used in peak calling. We applied our analytical protocol on the same dataset used by Zhu et al.. We observed a very high overlap between our prediction and the predicted enhancers mapped by Zhu et al. (>93% of our

prediction was supported by Zhu et al.; data not shown). To further support the power of our prediction, we functionally validated 22 enhancer candidates in either forward or reverse, or both directions (in total, 30 constructs). The results showed that 27 of 30 (90%) reporter lines showed ability of triggering GUS gene expression. This result together with the high rate (90%; **Fig. 4**) of functional validation support the reliability of our enhancer prediction.

• 291. Why widespread?

Response: Thank you for raising this question. We have explained it in the revision: *“Computational motif mining from a collection 750 distinct TFs revealed that all the predicted enhancers contained at least one predicted TF-binding site. The enrichment of predicted TF binding sites in putative enhancers was significantly higher than in flanking regions or random intergenic non-DHS regions (p -value $< 2.2e-16$ by Welch's t -test; Supplementary Fig. 13), indicating that these enhancers are subject to widespread TF regulation.”*

Supplementary Figure 13. Enrichment of predicted transcription factor (TF) binding sites in putative distal enhancers. For comparisons, enrichment was also calculated in enhancer flanking regions (at both sides with a matched size), random intergenic non-DHS regions (n = 10000), promoters (proximal DHSs) and promoter flanking regions. Enrichment score was calculated as the number of TF binding sites per bp in each region. Significance was determined by Welch two sample t-test. DHSs, DNase I hypersensitive sites.

- 292. Can you define precisely what the stage-specificity of promoters and distal DH sites is based on. This is missing in the main text and legend.

Response: Stage-specific promoters or enhancers mean the called DHSs were only identified in one specific stage. We have made this clear in the revision (both in the main text and figure legend).

- 292-293. This indeed makes sense. Regulatory sequences are known to provide tissue-specificity to promoters. Also refer to such findings in literature.

Response: Thank you for the comment. We have added reference here.

- 299. Compared to which type of sequences are the GWAS SNPs enriched in the distal and proximal DH sites? Please define this.

Response: Thank you for pointing this out. We have improved the description to make this clear. The corresponding sentence was changed as followed: “*These SNPs were associated with flowering-related phenotypes by a genome-wide association study (GWAS)⁴⁸*”.

- 302. Define in the legend or methods what the flanking regions consist of (how much of the sequence around do you consider flanking, and what type of sequence do you use/not use)? What does the control in Figure 5F consist of? Would it be possible to include such control for figure 5E-I.

Response: We thank the reviewer for the comments. The old Figure 5 has been reordered as Figure 3 in the revision. We plotted phastCons score around the peak centers of proximal (promoters) or distal DHSs (enhancers). The whole plotting regions cover 3 kb (1.5 kb in each side of the center). The median length of predicted promoters and enhancers is 564 and 465 (<1% regions exceeding 3 kb were cropped into 3 kb regions

based on peak centers), and the remaining regions were considered as flanking regions. We have added this information in the figure legend.

In Fig. 3e (old Fig. 5F), as a control, 10000 random intergenic non-DHS regions were used to calculate the distribution of phastCons score. We have also added this in the figure legend. We also included a control for Fig. 3d,f,g (old Fig. 5E,G,H in the revision) as the genome-wide average values. However, we failed to find a good control for Fig. 3h (old Fig. 3I).

- 310, 5I. Where does the lncRNA data come from (reference is missing)? How are they defined?

Response: We thank the reviewer for pointing this out. We are sorry that the reference is missing here. We have added the references from where we obtained the lncRNA data.

- 326. It is not clear if you picked two particular clusters out of more than two or if the data was clustered in 2 clusters. Also, this is not clear from the legend of Figure 6.

Response: The data were grouped into two clusters. We wrote this sentence in the revision: *“These dynamic enhancers were dominantly grouped into two distinct clusters based on their behavior in the time-course experiments (Fig. 6a; Supplementary Fig. 14).”*

- 330. What does proximal genes mean? Both up- and downstream? Only downstream? Only target genes?

Response: Here, “proximal” gene refers to the nearest neighboring gene. The gene can be up- or downstream. The revised description is *“Overall, the dynamics of enhancers in the two clusters coincided well with the expression changes of their nearest neighbor genes (Fig. 6b; Supplementary Fig. 15a).”*

- 337. The distal region is indeed upstream, but not far upstream.

Response: Thank you for pointing this out. We removed the word “far-”

- Both Figure 3E and 6D are about motif enrichment when comparing stage 2 and 8. It has to be clear from the start what the differences are between those two figures.

Response: Thank you for the suggestion. The original Figures 3E and 6D are TF binding site prediction based on motif scanning and both of the two analyses took the data from stage 2 and stage 8. In Figure 3E, stage-specific active core-promoters featured by proximal DHSs with H3K27ac were analyzed, while Figure 6D represents the predicted TF binding ability in stage-specific distal DHSs. For the sake of clarity and focus, we reorganized the section and removed Figure 3 (as mentioned in replies to previous comments). Regarding Figure 6D (now Figure 6e), we added the sentence “We focused on the distal DHSs that showed at least 1.5 times difference in accessibility between stage 2 and stage 8.” to introduce this information.

- 350-353. Twice stage 2 is addressed, with a ‘while’ in between. Should second ‘stage 2’ be ‘stage 8’?

Response: Here, the information is two LBD proteins are the top two predicted TFs but the top gene families (means more members from this family) are AP2 and WRKY. So, stage 2 is right. The highly enriched genes and the mostly appearing gene family are different at the same stage. We re-phrased these sentences to "*For the highly active enhancers at 2 DAI, two LATERAL ORGAN BOUNDARIES DOMAIN (LBD) proteins, LBD18 and LBD13, were among the top-enriched binding TFs. In more general terms, AP2-EREBP and WRKY TF families bound most predominantly at enhancers active at 2 DAI.*"

- 366. Function is not shown, but indicated. This is true for accessibility. For H3K27ac a correlation is shown.

Response: Thank you for the comment. We have removed the corresponding sentence in the revised version of the manuscript for simplification.

- 373-376, 6B. Is this about the proximal plus distal regulatory elements? How is defined that particular genes do not carry regulatory elements? They can also be in the proximal

promoter, downstream, within the gene, further away; they may be active in tissues not examined, etc. If a gene is not constitutively expressed it is regulated by REs.

Response: Thank you for pointing this out. Yes, it about the proximal plus distal regulatory elements. Genes without regulatory elements were taken into consideration as well. Regulatory elements were assigned to their closest genes based on the distance between peak summits and the TSS. However, we have to point out that the whole paragraph has been removed from this revision due to re-organization of our manuscript.

- Figure 6B: It is unclear to us if trends seen in gene expression and enhancer accessibility can really be considered as correlated. Especially from an average analysis. An analysis per target gene would be more relevant than a general trend comparison.

Response: We thank the reviewer for pointing out this issue. Following the reviewer's suggestion, we performed the correlation analysis based on "per target gene". As shown in Figure 6c, the median correlation coefficients between gene expression and chromatin accessibility at enhancer regions is 0.42 and 0.57 for genes in groups 1 and 2, respectively. We have also added some description about this point in the revision. *"Overall, the dynamics of enhancers in the two clusters coincided well with the expression changes of their nearest neighbor genes (Fig. 6b; Supplementary Fig. 15a). This observation was further supported by an analysis per target gene, in which the correlation between gene expression and chromatin accessibility at enhancer regions is significantly higher than that at promoters (Paired Student's t-test < 0.05; Fig. 6c; Supplementary Fig. 15b)."*

Figure 6b-c. (b) The median expression level (in dashed arrow) and enhancer accessibility (in solid arrow) across developmental stages for the genes in the two groups as indicated in (a). Pearson correlation coefficients r between gene expression and enhancer accessibility for each group are shown in parentheses. (c) Violin plots (with box plots inside) showing the Pearson correlation coefficient between expression level and chromatin accessibility for each gene in different groups. Promoters with respect to the enhancers in each group were used for comparison. The number of genes used in the analysis is indicated in parenthesis. The median correlation coefficient is shown above the violin plot.

Discussion

The discussion needs to be revised based on the comments made on the main text.

Response: We extensively re-wrote the discussion section in the revised manuscript. Firstly, we discuss about the chromatin features and species-specific characteristics of plant enhancers. Next, we discuss the stage specificity and dynamics of enhancers during flower development, and the links between enhancer activity and the stage-specific gene expression pattern. Finally, based on some new results from the revision, we discuss the roles of enhancer clusters and intronic enhancers in re-shaping transcriptome dynamics in flower development. More details information can be found in pages 15-17 of the revised manuscript.

We made just a few comments below:

- 396-398. There are more genome-wide studies on regulatory elements in plants than the one cited here. See also our comment above. Please place the findings in a broader perspective.

Response: Thank you for the suggestion. We have added the information from the studies as well as other related reports in the third paragraph of introduction part. What we would like to discuss here is the function of enhancer dynamics in flower development. To emphasize this point and to follow the suggestion from the reviewers, we rewrote the first paragraph as *“The general features and cell-type specific activities of animal enhancers have been intensively studied. In contrast, much less is known about dynamic activities of enhancers during plant development.”*

- 399. Indicate the mapping is defined on DH sites.

Response: Thank you for the suggestion. We re-wrote the discussion part, hopefully now this information is clear in our newly written paragraph.

- 404. Intergenic should be changed to more than 1kb upstream of TSS.

Response: Thank you for this comment. We have changed "intergenic" into "more than 1kb upstream of the TSS".

- 408-413. Compactness of different genomes does not explain why a specific histone mark would be lacking an association with distal enhancers in Arabidopsis.

Response: It is true that there is no report linking the distribution of H3K27ac and genome compactness. We saw different H3K27ac distribution pattern between Arabidopsis and rice and genome compactness is one very obvious difference between the two. So, we questioned whether this is the reason. However, considering the situation in Drosophila which has a compact genome but takes H3K27ac as enhancer marker, we agree with the reviewer that genome compactness might not be the reason why H3K27ac hardly appears in distal region in Arabidopsis. In this regard, we further discussed about this situation by *"This indicates that H3K27ac does not mark enhancers in Arabidopsis thaliana. This cannot be explained as a simple consequence of genome compactness, since e.g. the Drosophila genome is also compact, but H3K27ac marks its enhancers^{67,68}.*

An important question is now to understand how the diverging H3K27ac patterns are established in different plant species. In animal cells, H3K27ac deposition in enhancers is mainly mediated by the CREB-binding protein (CBP) and p300 histone acetyltransferases^{69,70}. An Arabidopsis thaliana CBP/p300 homolog was reported to promote flowering by affecting expression of FLOWERING LOCUS C (FLC), but not via H3K27ac⁷¹. Plant and animal CBP/p300 proteins share highly conserved C-terminal segment, which is necessary for acetyltransferase activity⁷². Other domains have diverged between plant and animal homologs, suggesting divergent mechanisms of recruitment to their genomic target sites⁷³. This does however not explain the differences in H3K27ac patterns observed in Arabidopsis thaliana and rice/maize. One possible explanation is that the interactions among different types of chromatin modifications including histone acetylation, methylation, and DNA methylation are plant species specific. It has been shown that in rice DNA methylation at non-CG sites positively correlates with H3K27me3 in euchromatic regions, which is the opposite situation in Arabidopsis thaliana⁷⁴. Whether H3K27ac is involved in interactions with other chromatin features and the interaction modes differ between plant species is an open question to be addressed."

- 417. H3K27me3 is positively associated with poised enhancers and negatively associated with active enhancers.

Response: We have added the missing word and changed the wrong word.

- 439. The functional relevance between H3K27ac and open chromatin is quite well established. At least in other systems. The cascade of events and general mechanisms leading to transcription in Arabidopsis is expected to be similar.

Response: We thank the reviewer for pointing this out and we agree with the reviewer about the points here. Since this finding is as expected, we change to focus on discussing the enhancer function by removing H3K27ac/open chromatin part of discussion in the revision.

- Title 451. "hidden driver" is a bit strong. Reformulate.

Response: We have changed "hidden driver" to "associated with"

- 463-465. Sentence is stating a well-established concept. Please refer to relevant papers.

Response: Thank you for the suggestion. We have rewritten the whole discussion section in the revision (see pages 15-17) and this sentence has been removed.

- 470-472. The sentence suggests this has been shown in this study while it has not been demonstrated.

Response: We thank the reviewer for pointing this out. This sentence was reshaped from existing studies but not from our study. To avoid misleading, we have removed this sentence in the discussion.

Figures

- 1B. Arrow heads indicating TSS are very small.

Response: Thank you for the suggestion. We have made bigger arrows.

- Figure 2A. Why are the groups not numbered in ascending/descending order? Or based on the nr of sites within a group.

Response: Thank you for pointing this out. We now numbered the groups in an ascending order.

- Figure 2C. Please indicate the TSS and mention what the bars below the peaks represent (we assume the peaks called by X program). Indicate in the legend that examples are shown from different groups depicted in A.

Response: We thank the reviewer for pointing this out. This figure has been moved to Supplementary Fig. S8 in the revision. We has improved the figure as well as the description in the figure legend.

- 3C: bewteen->between

Response: We thank the reviewer for pointing out this mistake. We have corrected this. However, due to the reorganization of the manuscript, this figure does no longer exist.

- 3E: What is the meaning of blue versus red?

Response: Blue means more stage 2-specific and red more stage 8-specific. We have clarified this in the figure legend. However, in the revised version of the manuscript, we focus on the regulatory role of enhancer dynamics, and this figure is not anymore involved.

- 4B. How is the density calculated? 25% of the peaks are distal. If the peaks would have the same strength, the average density of the distal peaks would be lower than those close to the TSS.

Response: Thank you for pointing out this. For visualization purposes, the x-axis is log10 transformed. We have stated this in the legend.

- 4C, panels on the right. H3K9ac is enriched downstream of the TSS, see Figure 1C. So when examining proximal DHSs, you should see a peak of H3K27ac on one side, rather than two sides. You may have forgotten to orient the data relative to the transcription direction. It is not clear what this panel adds to the figures/panels already shown.

Response: In our analysis, we took the genome orientation as the orient of DHSs. It is not easy to assign the orientation of DHSs relative to the transcription direction especially for a DHS located at the middle to two genes (the DHS may be responsible for both genes in this case). In figure 1C, H3K27ac peaks is shown according the orient of genes (from TSS to TES). Therefore, Figure 4c and Figure 1c are shown the data in two different ways. However, we followed the reviewer's suggestion and took the direction of the nearest genes as the DHS's orientation. The results are shown in Figure 2c and Supplementary Figure 10 in the revision.

- 4E. The yellow for H3K4me2 in rice is quite difficult to see.

Response: We have improved this figure (in a higher resolution).

- 5B. The first D in the legend should be B.

Response: We thank the reviewer for pointing out this mistake. We have corrected this mistake in the revision.

- 5C. Enhancers should be changed into putative enhancers, also in the Legend.

Response: We have changed this according to the reviewer's suggestion.

- 5F. What is meant with below and above parts in F? Scores for individual DHSs are not visible. The legend does not seem to fit the figure panel!

Response: We are sorry for the incomplete figure legend. We have improved the figure legend in the revision.

- 7C, how should the multiple line features be read (3, 5, 7)? What is the difference between 4 and 9, etc? It seems 8-11 only picks out particular genes? But this is not mentioned anywhere. Please add a detailed description of the figure in the legend.

Response: We thank the reviewer for pointing out this. A more detailed description in the figure legend would be *“For tracks 3, 5 and 7, multiple line features divide values (e.g., intensity difference among the four stages) into five different levels. Tracks 4 and 6 are the distribution of all peaks, while tracks 8-11 are peaks associated with flowering genes. In track 12, only flowering genes with dynamic REs and/or differential expression are labeled.”* However, since now we focus on the regulatory role of enhancer dynamics in flower development, previous Figure 7 has been removed in this new manuscript.

- Suppl Fig3A. Binding intensity is not the appropriate wording. H3K27ac does not bind. ‘ChIP signal intensity’ is one alternative.

Response: Thank you for the comment. We have corrected this.

- S4: ‘related to Fig 1’ should be between brackets, and not be part of the title.

Response: This has been corrected.

- S5, legend. Accumulative fraction of which peaks?

Response: This has been corrected.

- S6: Peaks from all the four stages were firstly merged for reference. Then the peaks from a specific stage were counted based on the merged peaks. Legend has to be clarified.

Response: We have improved the description.

- S8C. This panel is not very clear. It seems that numbers and % are mixed. Percentages could be added in a separate table?

Response: We have improved the figure.

- S10B is indicated as rice data in the legend. I however think it is showing Arabidopsis data. Similarly, S10C would show human data, but the panel itself indicates leaf and flower. Are the legends of S11 and S10 swapped?

Response: Thank you for pointing out this and we are sorry for this mistake. These two legends are indeed swapped. We have corrected this mistake.

- S11. Same comment as for 4C. Is the data oriented relative to transcription direction? It should.

Response: Thank you for this comment. We took the genome orientation as the orientation of DHSs. It is not easy to assign the orient of DHSs relative to the transcription direction especially for DHSs located at the middle to two genes. In figure 1C, H3K27ac peaks is shown according the orient of genes (from TSS to TES). Therefore, Figure 4c and Figure 1c are shown the data in two different ways. However, we followed the reviewer's suggestion and took the direction of the nearest genes as the DHS's orientation. The results are shown in Figure 2c and Supplementary Figure 12 in the revision.

Tables

- Suppl Table 1: explain where the score refers to.

Response: We have explained this in the revised manuscript.

- Suppl Table 2-Motif Analysis: why are there NA's (Not applicable?) in the RNA-seq column?

Response: The reviewer is right, “NA” for not applicable / not available.

Typos/textual/reference issues

- 27. ‘With’ should be added before both. 50. activate should be activate.

Response: Thank you for pointing out and we are sorry for the mistakes. We have corrected these mistakes.

- 31. Would use another word than ‘derived’. Derived does not seem appropriate. Also holds for rest of the text.

Response: Thank you for the suggestion. We now use the phrase "enhancers predicted based-on DHS information”.

- 50.to activate

Response: Thank you for pointing out this mistake. We have corrected it in the revision.

- 51.enhancers are devoided

Response: Thank you for the suggestion. However, we think “are devoid of ” is correct in grammar. Therefore, we would like to keep it as it is.

- 53. Mbs

Response: Thank you for pointing out this mistake. We have corrected it in the revision.

- 53. ‘thus’, should be deleted. There is no causal relationship with the previous sentence.

Response: Thank you for the suggestion. We agree with the reviewer and have deleted "thus" in the revised manuscript.

- 56. Similarly, ‘accordingly’ should be deleted.

Response: Thank you for the suggestion. We agree with the reviewer and have deleted "accordingly " in the revision.

- 61-62. When saying ‘More and more reports’ one should list more than 1 reference.

Response: We thank the reviewer for pointing this out. We have added references in the revised manuscript.

- 68. Leading to the discovery

Response: Thank you for pointing this out. We have corrected this mistake in the revision.

- 74. Of enhancer’s function

Response: Thank you for the suggestion. However, here we want to state the link between genes and enhancers, more than to emphasize a spatial relationship. Therefore, we would like to keep it as it is.

- 75. Have should be has. Or analysis should be changed into analyses.

Response: We thank the reviewer for pointing this out. We have changed "analysis" into "analyses" in the revised version of the manuscript.

- 93. 'that' should be 'which', or rephrase it otherwise.

Response: Thank you for the suggestion. we have rephrased this sentence as "*Here, we used flower development as an ideal system to study enhancer dynamics, assess their roles in the control of stage-specific gene expression and address whether their dynamics correlates with H3K27ac changes.*".

- 93. Here, we took advantage of flower formation as an ideal system...

Response: Thank you for the suggestion. we have rephrased this sentence as "*Here, we used flower development as an ideal system to study enhancer dynamics, assess their roles in the control of stage-specific gene expression and address whether their dynamics correlates with H3K27ac changes.*".

- 103. 'does' -> 'is'

Response: We thank the reviewer for the pointing out the mistake and the suggestion. We have changed "does" into "is" in the revision.

- 108 largely -> highly

Response: Thank you for the suggestion. We changed the word "largely" into "highly" in the revised version.

- 109. To drive -> , which is associated with

Response: We thank the reviewer for the suggestion. We modified the sentence following the suggestion.

- 118: add 'a' before dynamic. And add an s after modification. There are a lot more of these errors in the text. Missing 'the', plural where it should be singular or vice versa, etc. Sometimes even a verb is missing. I won't indicate the rest. Please track them yourself and correct the text. It hampers smooth reading of the text quite a bit. A few more are listed below, but there are several more in the text.

Response: Thank you very much for your careful reading and kindly pointing these out. We have checked throughout the text and have tried to improve the grammar.

- 132-133. at all of the four developmental stages

Response: Thank you for the suggestion. We have changed the phrase in the revision.

- 134. ‘Regulatory architecture’: we would use more explicit terminology: H3K27ac and DHS profiles.

Response: We thank the reviewer for the suggestion. We have used "H3K27ac and DHS profiles" in the revision.

- 137. Coordinating assumes a causal relationship, which your experiments in Figure 1 do not show.

Response: We agree with the reviewer that "coordinating" is not a proper word here. We have used "correlate with" in the revision.

- 139. ‘the regions’ should be specified.

Response: Thank you for the suggestion. We have rephrased the sentence by "within regions surrounding the TSS (Fig. 1c)."

- 141. ‘genetic regions’: the entire genome is ‘genetic’. This sentence is not complete. It lacks a verb (“are” in between “sites” and “located”).

Response: We thank the reviewer for pointing this out and we are sorry for the mistake. Here, it should be "intragenic". We have corrected this mistake in the revision.

- 143. Specifically, of the approximately 6000 and 11160...

Response: Thank you for the suggestion. We have rephrased the whole sentence as "*In detail, out of all the high quality H3K27ac peaks identified at different developmental stages (Supplementary Table 1), more than 95% were found in gene bodies, whereas only about 2.5% were mapped to distal intergenic regions (>1 kb from the TSS; Fig. 1d - left panel).*"

- 145. Apart should be away.

Response: We apologize for the mistakes and thank you very much for pointing these out. We have changed the text based on the suggestions from the reviewer.

- 148. Reference to 1C is not correct. No distal peaks are visible.

Response: We thank the reviewer for pointing this out. We removed the citation of Fig. 1c here and only cited the Fig. 1d. Please note that the Fig. 1d has been improved in the revision.

- 160 & 187-188. The authors state they examine the role and contribution of H3K27ac to stage specific gene expression. The experiments performed examine correlations, not more than that. Mutants in H3K27ac are not used. The text has to be adapted.

Response: Thank you for the comments. Yes, we just showed the co-relationship. Since in the revised version of the manuscript, we focus on the regulatory role of enhancer dynamics in flower development, the correlation between H3K27ac and active gene expression, which had been discussed in other studies is not the major focus of this manuscript. We re-wrote this part in the revision and these sentences are not there anymore.

- 184-186. It is not clear why 'it was not surprising....'. Would delete this start of the sentence. Alternatively you have to explain why it was not surprising.

Response: Thank you for pointing this out! Due to the reorganization of the manuscript, this sentences is not there anymore.

- 194. Maser?

Response: Thank you for pointing this out and we apologize for the mistake. It should be "master".

- 213-214. Sentence is incomplete/incorrect.

Response: Thank you for the comments. Due to the reorganization of the manuscript, this sentences is not there anymore.

- 217-218. Incorrect sentence.

Response: Thank you for pointing this out. We have rewritten this sentence as "*Recent work showed that H3K27ac co-localizes with distal intergenic DHSs in Arabidopsis thaliana, indicating that this mark associates with predicted enhancers³⁶.*"

- 224-225. H3K27ac peak centers mostly presented...? Second part sentence also lacks a verb.

Response: Thank you for pointing out the mistake. We have improved these sentences in the revision.

- Figure 4B. Why is the upstream peak not visible in 2C? The axis extends up to -3kb and upstream peaks at -1.6kb should therefore be visible on this figure!

Response: We guess the reviewer meant “4C” instead 2C here. In Figure 4C, we indeed see that there are more H3K27ac peaks located at >1.6kb away from the center of distal DHSs. Please note that “1.6kb” is the over representative distance between TSSs and H3K27ac peak summits but not the distance between distal DHSs and H3K27ac peak summits.

- 228. What is different with the data shown in FigS5? That figure already shows distances between DHSs and H3K27ac peaks.

Response: Thank you for pointing out this. The reviewer is right that these two figures show similar data but in a complemented way. We also cited Fig. S5 here.

- 229-235. Twice the same message is written down.

Response: Thank you for pointing out this. We have rewritten this part.

- 4C, legend: above parts, below parts...should be top panels, bottom panels.

Response: We thank the reviewer for pointing out this mistake. We have changed this in the revision.

- 292: at all of the four stages

Response: We have corrected this mistake. Than yous.

- 297-298. ...occurring in promoters defin

Response: Thank you for the comments. We have reshaped this sentence.

- 298 predicated -> predicted.ed by....?

Response: We have made this change. Thank you.

- 365. Charting?

Response: Thank you for the comments. Due to the rearrangement of the manuscript, this part does no longer exist.

Reviewer #2 (Remarks to the Author):

Overall the study appears to be quite strong and its conclusions fairly well supported. It will be a good resource for others to use in the future as and when ChIP-seq and expression data becomes available for other TFs. There might have been some question

with inferring the TF binding through motif but the strategy to narrow down the TF motifs being observed seems sound. The major issue appears to be the clarity of some of the figures that are structured or labeled in a way that might not be immediately clear to some of the readers. It might be nice to have some more in depth ideas about why there were discrepancies between some of the paper's major findings and previous literature. You could also provide a short explanation of how your seedling assays are suitably comparable to your earlier floral data and why you didn't just use floral tissue all around. I understand that it is a lot more work in generating all these data sets in ap1;call1, but at the same time, comparisons made with leaves and other tissues may lead to misinterpretations. Would chromatin conformation data matched to the other Seq studies you performed would be more informative as opposed to obtaining it from other studies? It would be interesting to see more possible future directions discussed. For example, why H3K27ac does not mark enhancers? H3K27ac and DHS may cooperate together in determining expression. How do they come together/communicate? Do you have any plans on how to strengthen the support for your conclusions even further or new directions on which to take your findings?

Response: We thank the reviewer for the comments. Following the suggestion from the reviewer (as well as other reviewers), we largely revised our manuscript by including experimental validations (see Fig. 4) and new analyses (see Figs. 5 and 7) to focus more on the key results, e.g., the dynamics and functions of *Arabidopsis* enhancers. We also rewritten the whole discussion part. For example, we extended our discussion about the point of why H3K27ac not marking *Arabidopsis* enhancers in the revision: "... *This indicates that H3K27ac does not mark enhancers in Arabidopsis thaliana. This cannot be explained as a simple consequence of genome compactness, since e.g. the Drosophila genome is also compact, but H3K27ac marks its enhancers*^{67,68}. *An important question is now to understand how the diverging H3K27ac patterns are established in different plant species. In animal cells, H3K27ac deposition in enhancers is mainly mediated by the CREB-binding protein (CBP) and p300 histone acetyltransferases*^{69,70}. *An Arabidopsis thaliana CBP/p300 homolog was reported to promote flowering by affecting expression of FLOWERING LOCUS C (FLC), but not via H3K27ac*⁷¹. *Plant and animal CBP/p300 proteins share highly conserved C-terminal segment, which is necessary for*

*acetyltransferase activity*⁷². Other domains have diverged between plant and animal homologs, suggesting divergent mechanisms of recruitment to their genomic target sites⁷³. This does however not explain the differences in H3K27ac patterns observed in *Arabidopsis thaliana* and rice/maize. One possible explanation is that the interactions among different types of chromatin modifications including histone acetylation, methylation, and DNA methylation are plant species specific. It has been shown that in rice DNA methylation at non-CG sites positively correlates with H3K27me3 in euchromatic regions, which is the opposite situation in *Arabidopsis thaliana*⁷⁴. Whether H3K27ac is involved in interactions with other chromatin features and the interaction modes differ between plant species is an open question to be addressed.”

Comments

1. Even though you reference Hi-C analysis, perhaps **matched chromatin structure** assays could be informative along with the RNAseq, DNase-seq, and ChIP-seq.

Response: We thank the reviewer for the comment. We agree that additional chromatin structure assays would be informative to further study this phenomenon. Unfortunately such data are not yet available for the analyzed tissues, so this remains an important area of research for future investigations.

2. Line 151-151 if most H3K27ac sites remain static throughout development and only a fraction are stage specific what could the function of most of them be?

Response: Thank you for pointing out this. We now included a GO analysis of these stage-specific loci. Results showed that GO terms such as ‘regulation of gene expression’, ‘meristem maintenance’ and flower developmental-related processes are enriched in these stage-specific target genes (see Supplementary Fig. 8d).

3. Line 167 Could you be a little more specific about just how well do genes from each cluster track with its associated H3K27ac binding pattern? Do the majority of genes of every single cluster follows the H3K27ac pattern?

Response: We thank the reviewer for pointing this out. We refined this part of analysis to determine the optimal number of clusters by an elbow method (see Supplementary Figure

8a in this reviewer reply). Based on our analysis, the number of clusters was set to four. Indeed, in each of the clusters, the dynamic of H3K27ac enrichment and change in gene expression levels showed positively correlated. Please find a new version of Supplementary Figure S8 for more details.

Supplementary Fig. 8: Dynamically changed H3K27ac-associated regions across flower development. (a) Determining the optimal number of clusters. Three different methods (elbow, Silhouette and gap statistic) were used to compute the optimal number of clusters ($k=4$ in all cases as highlighted in red). (b) Heatmap showing the clustering analysis of 2913 H3K27ac peak regions that dynamically change more than 1.5-fold across the four stages (from stages 0 [S0] to 8[S8]). These regions are clustered into four groups (in

different colors) with distinct patterns of H3K27ac enrichment. The blue bars on the left of heatmap indicate the nearest DHSs for the corresponding H3K27ac peaks have more than 1.5-fold change (dDHS). The pink bars indicate the nearest genes are differentially expressed (DE). Flowering genes are labeled on the right of heatmap. (c) Summary of H3K27ac enrichment in the six groups as colored in (b). Grey curves indicate the median scaled expression level of genes in the corresponding groups. Correlation coefficient values between H3K27ac signal intensity and gene expression are shown. (d) Gene ontology (GO) enrichment analysis for each gene clusters as represented in (b).

4. Line 184 What would be the reasons DHS and H3K27ac biologically combine together for higher expression in linked genes than DHS alone. Is the same true for H3K27ac alone?

Response: The reason for this observation in plants is still unknown. Inspired by the observation that H3K27ac-positive DHSs are indicative of active enhancers in mammalian studies (e.g., Maatouk et al., 2017), one explanation for our data could be that H3K27ac-marked DHSs are putative active promoters (since H3K27ac generally locates at TSS proximal regions), possibly because H3K27ac not only enhances recruitment, by also transcript elongation by Pol II, as has been seen in other systems (Zlotorynski 2014, 10.1038/nrm3889). Indeed, the same is true for H3K27ac alone (see Supplementary Fig. 4 in the revision).

5. Line 201 Selected motifs maybe show certain patterns but do you think this is enough to generalize to developmentally important factors in general?

Response: We collected motif models for Arabidopsis TFs from public databases, including JASPAR (Mathelier et al, 2014), UniPROBE (Hume et al, 2015) and Athamap (Steffens et al, 2005), as well as from studies (Franco-Zorrilla et al, 2014; O'Malley et al, 2016; Sullivan et al, 2014; Weirauch et al, 2014). This collection covered more than 750 distinct TFs in a diverse set of TF families. We believe that the selected motifs are generally sufficient for the purpose of this analysis. We would like to point out that we shortened the part of the manuscript to which the comment refers in order to focus more on the enhancer analysis (for details, please see the reply to reviewer 1). However, this comment and its reply remain relevant for the results shown in Figure 6.

6. Line 217 what do you think is a reason for the discrepancy between the previous study and your data concerning H3K27ac? Maybe the discrepancy is due partly to the stages you chose to process or different tissues that you used.

Response: We thank the reviewer for pointing this out. From our in depth investigation, the discrepancy may be due to the strategy used in the analysis. In Zhu et al's study, they examined H3K27ac profiles around the predicted enhancers in a window of 5 kb (Figure 2 and Supplemental Figure 1 in Zhu et al's paper). Due to the compact Arabidopsis genome, 5-kb flanking regions already cover surrounding genes. This can create bias in the analysis. For example, the proposed 'enhancer' shown in Zhu et al's Supplemental Figure 1B is in fact not a putative enhancer but the promoter region of *MIR156a*. In contrast, when plotting the relative distance between H3K27ac-marked regions and the TSSs of proximal genes (Figure 2), we instead observed that nearly all the H3K27ac peaks located in TSS-proximal regions. The same results were obtained from a reanalysis of Zhu et al's data (Supplementary Figure 11 in this revision). Therefore, we don't think the discrepancy is due to different tissues used.

7. Line 230 Perhaps the reason why there so many H3K27ac sites are near DHSs but not the other way around is simply DHSs are more widely distributed and common?

Response: This could be one of the explanations but it is not likely to be the only reason because we only observed this in Arabidopsis but not in rice, and the H3K27ac pattern is specific to Arabidopsis. More specifically, 12.7% (2070/16323) of all H3K27ac sites are located in intergenic regions in rice, while 2.8% (248/8763) of all H3K27ac sites are located in Arabidopsis intergenic regions. In contrast, the fractions of intergenic DHSs are comparable in both species: 28.4% (6307/22192) in rice and e.g. 19.7% (3805/19352) in Arabidopsis at 8 DAI. We included some discussion on this point in the revision, e.g., *"This cannot be explained as a simple consequence of genome compactness, since e.g. the Drosophila genome is also compact, but H3K27ac marks its enhancers^{67,68}. An important question is now to understand how the diverging H3K27ac patterns are established in different plant species. In animal cells, H3K27ac deposition in enhancers is mainly mediated by the CREB-binding protein (CBP) and p300 histone acetyltransferases^{69,70}. An Arabidopsis thaliana CBP/p300 homolog was reported to*

promote flowering by affecting expression of FLOWERING LOCUS C (FLC), but not via H3K27ac⁷¹. Plant and animal CBP/p300 proteins share highly conserved C-terminal segment, which is necessary for acetyltransferase activity⁷². Other domains have diverged between plant and animal homologs, suggesting divergent mechanisms of recruitment to their genomic target sites⁷³. This does however not explain the differences in H3K27ac patterns observed in Arabidopsis thaliana and rice/maize. One possible explanation is that the interactions among different types of chromatin modifications including histone acetylation, methylation, and DNA methylation are plant species specific. It has been shown that in rice DNA methylation at non-CG sites positively correlates with H3K27me3 in euchromatic regions, which is the opposite situation in Arabidopsis thaliana⁷⁴. Whether H3K27ac is involved in interactions with other chromatin features and the interaction modes differ between plant species is an open question to be addressed.” Please read more information on page 16 of the revised manuscript.

8. Line 247 Is there enough reason to believe the seedling tissue is directly comparable to the flower tissue results?

Response: We thank the reviewer for this interesting comment. If we compare the data for H3K27ac and DHSs between seedling tissues (Fig. 2d and Supplementary Fig. 12a) and flower tissues (Fig. 2c), we actually see no difference of the global peak distribution patterns. Therefore, we believe the patterns are directly comparable between different tissues.

9. Line 272 Could it be clarified whether the 1kb cutoff for intergenic DHSs is arbitrary?

Response: The 1 kb cutoff was assigned based on the genome-wide distribution pattern of DHSs. As shown in Figure 2b, most of the distal DHSs were found >1 kb away from a TSS.

11. Line 276 So having both proximal and distal DHS sites implies that the distal site is an enhancer since the proximal DHS site represents open chromatin around the gene while the distal DHS is open chromatin around its associated enhancer?

Response: Yes, this is indeed the conclusion that was drawn based on the data.

12. Line 290 The correlation between motifs and open chromatin is interesting but its hard to be sure without more direct evidence. That is, stage specific CHIP data of the TF.

Response: Yes, we agree with the reviewer comment. Generating additional, stage-specific ChIP-seq data for a larger set of TFs is highly demanded in plant field but the analysis we did here is an alternative way to show the data.

13. Line 299 If you later say that enhancers are under increased conservation, what would you say is the biological significance of increased SNPS in the promoters and enhancers?

Response: Thank you for this important remark. Regarding evolutionary conservation, the increased conservation of promoters and enhancers is relative to intergenic non-DHS regions (“control” in Fig. 3e). Although highly conserved, promoters and enhancers can still harbor many SNPs. Regarding the biological significance of this, the comparison is made between promoters and enhancers (Fig. 3d-g). In the revised version, we additionally added a genome-wide control to show that promoters and enhancers have more GWAS SNPs than a random control (see Fig. 3d-g). To answer the reviewer comment: the enrichment of GWAS SNPs suggests that evolutionary changes in the analyzed promoters and enhancers suggests that they play roles in regulatory divergence of flowering-related developmental traits in Arabidopsis.

14. Line 305 By surrounding regions do you mean enhancers or just surrounding DNA in general? If you mean non-enhancer DNA of little to no epigenetic function, what would be the explanation of promoters being less conserved than that?

Response: We thank the reviewer for pointing this out. We mean the surrounding DNA in general. We supposed that this may be due to the effect of surrounding gene bodies. To dig out this issue, we plotted the distribution of PhastCons score in a strand-specific manner. Indeed, positive-strand promoters showed a phantom peak of conservation only at their downstream while negative-strand promoters had a conservation peak at their upstream (Fig. 3e). We made this clear in this revision.

15. Line 320 conformation instead of confirmation?

Response: We apologize for the mistake and thank you for pointing out this. We have corrected the mistake.

16. Line 323 Based upon the Hi-C analysis, most but not all the enhancers were associated with their nearest genes. Maybe it would be better to partition how you handle the enhancers based upon this study rather than assume all operate on the nearest neighbor.

Response: We thank the reviewer for this nice suggestion. We indeed assigned enhancers to their nearest target genes based on the linear genomic distance rather than the Hi-C analysis. Based on our analysis, more than 75% of enhancers located at their target genes within 3 kb from the TSS (between TSSs and enhancer centers; see Figure R2 below). There is no obvious difference of our results when we partitioned enhancers based on their distance to targets (i.e., < 3 kb vs > 3 kb). Since we mainly focused on the 810 dynamically changed enhancers, the distance effect can be ignored since only a few enhancers are far away (e.g., > 3 kb) from their putative targets. Thus, we would like to keep the analysis in its current status.

Figure R2. Density plot showing the proportion of enhancers far away from an indicated distance with respect to the TSS of their nearest target genes.

17. Line 328 Maybe more than just two significant clusters could have been extracted if the parameters of the clustering algorithm were set differently. There seems to be a lot of variety in how those enhancers are behaving across the different stages within each of those two clusters.

Response: We thank the reviewer for raising this point. The reviewer is right that there is some variety within each of the two clusters. However, these two clusters are the most representative ones as determined by the optimization analysis (see Supplementary Figure 14). Meanwhile, we performed a similar analysis with a sub-optimal number of clusters, e.g., $k = 3$ or $k = 4$. We still observed two main trends of enhancer activity in the time-course experiments. Thus, we would like to use two clusters to present our results in the Figure 6.

Supplementary Figure 14. Enhancer dynamics across flower development. (a) Determining the optimal number of clusters using three different methods (elbow, Silhouette and gap statistic). The optimal number of clusters is highlighted in red. (b,c) Similar to Figure 4a,b, the enhancers are grouped into three (b) or four (c) classes. RPM values for enhancer activity, and FPKM for gene expression. From the boxplots, two major clusters are observed for clustering analysis with $k=3$ or $k=4$. Thus, we considered $k=2$ as the optimal number of clusters.

18. Line 376 When you are referring to the percentages of 20.6% and 33.6% it appears to be unclear whether you are referring to flowering genes, all genes, or TF genes.

Response: Thank you for pointing at the lack of clarity. The percentages refer to flowering genes. However, we have removed this part of results in the revision due to re-organization of the manuscript.

19. Line 454 corrected->correlated?

Response: We are sorry for the mistake and have corrected it.

20. Line 427 Could you go into detail about how your classification of the DHSs might be more preferable than Zhu's?

Response: We thank the reviewer for this comment. Zhu et al. considered the DHSs with distance > 1.5 kb from the TSS (regardless their peak centers) as enhancers, although some of these “distal” DHSs may still locate within 1 kb apart from the TSS because some DHSs can span several kilobases. However, we predicted much less intergenic enhancers than Zhu et al., mostly due to the different criteria used in peak calling. We applied our analytical protocol on the same dataset used by Zhu et al.. We observed a very high overlap between our prediction and the predicted enhancers mapped by Zhu et al. ($>93\%$ of our prediction was supported by Zhu et al.; data not shown). To further support the power of our prediction, we functionally validated 22 enhancer candidates in either forward or reverse, or both directions (in total, 30 constructs). The results showed that 27 of 30 (90%) reporter lines showed ability of triggering GUS gene expression. This result together with the high rate (90%; Fig. 4) of functional validation support the reliability of our enhancer prediction.

Figure Comments

1. 2B Do all 10 clusters of H3K27ac behavior have biological significance? Only a fraction are discussed in the paper.

Response: Thank you for the comments, we have optimized the number of clusters as four. In this revision, all the four groups were discussed in the manuscript (see Supplementary Fig. 8 in the revision).

2. Fig3C Figure is confusing and could use more clear markings. They appear to show the DHS and H3K27ac peaks but what does the x axis represent? Are they just all 805 peaks lined up with the DHS results between Stage 2 and 8?

Response: Sorry for the confusing. We improved the schematic representation in Fig. 3C and its legend (see Figure R3 below). In Fig. 3C, x axis in the browser track is genomic position of DHS and H3K27ac peaks. However, the whole Fig. 3 has been removed from this revision in order to focus more on the key results, namely the (floral) enhancer analysis.

Figure R3. Schematic representation of the strategy to identify stage-specific active promoters at stages 2 and 8. Promoter regions are defined by DHSs. Activeness of promoters are marked by H3K27ac modification.

3. Fig3E Were these transcription factors the only ones examined for enrichment? How were certain ones chosen for displaying their motif logos?

Response: No, we actually included > 750 transcription factors in the analysis. Only the top enriched TFs were shown in the figure. In the revised version of the manuscript, this Figure was omitted in order to focus more on the key results, namely the (floral) enhancer analysis.

4. Fig 4 The scale for D, E, and F is the same as that for C? (mentioned in text as the same)

Response: Yes, the scale is the same. We made it more clear in the Figures. Due to rearrangement of the manuscript, the original Fig. 4D-F is now Fig. 2d-f.

5. Fig 5b is mislabeled in the legend as 5d. .

Response: Sorry for the mistake and thank you for pointing out this. We have corrected this mistake. Due to rearrangement of the manuscript, the old Fig. 5d is now Fig. 3d.

6. Fig 6A: would you happen to have an explanation why there are much more genes active in the later stage? (Cluster 2) rather than the active in the earlier stage? (cluster 1)

Response: Thank you for pointing out this observation. The number of enhancers in cluster 1 represents enhancers that are either significantly enriched at timepoints 0, 2 or 4, while enhancers in cluster 2 are specifically enriched at timepoint 8. So this could be a technical explanation for the observed differences in numbers.

7. 6B Could you elaborate on why RPM and FPKM are ideal for the particular cases in which they are used? From the figures saying perfectly coincided may be a little too strong since there looks like there are very slight discrepancies between the trends of some of the boxplots.

Response: Thank you for this interesting points. FPKM is equally termed RPKM in our data since we used single-end RNA-seq. By definition, the only difference between RPKM and RPM is that RPKM is further subjected to normalization for the length of the gene. However, since protein-coding genes often include intron and thus may have several isoforms, and the length of different genes varies a lot, RPKM is a popular solution to remove these effects. In contrast, H3K27ac-marked regions generally have similar peak size. RPM works well (and is widely used) in this case. For the second point, we toned down our statement to respond to this comment.

8. 6c Do you have examples of several other genes where the DNAase and TF binding results line up as well as this sample that are good enough to be main figures?

Response: We thank the reviewer for pointing this out. We included more examples in the Fig. 4.

9. 7C is somewhat confusing because it is difficult to understand what each of the circles represent and how they relate to the list in the middle.

Response: We thank the reviewer for this comment. We could add more description in the figure legend. However, this Figure was omitted in order to focus more on the key results, namely the enhancer clusters and intronic enhancers.

Reviewer #3 (Remarks to the Author):

The paper entitled “Dynamic control of enhancer activity drives stage-specific gene expression during flower morphogenesis” describes the genome-wide localization of histone H3 lysine 27 acetylation (H3K27ac) and compares the data with DNase I hypersensitivity sites (DHSs) and RNA-seq at four representative stages of Arabidopsis flower development. They show that H3K27ac dynamics and DHSs are highly correlated with stage-specific gene expression. They also claim that DHS-derived enhancers are highly stage-specific and associated with SNPs in GWAS for flowering-related phenotypes. The same group previously reported the relationship of DHSs, gene expression and binding of two MADS-domain transcription factors APETALA1 and SEPALLATA3 in Arabidopsis flower development using the same floral induction system (Pajoro et al. *Genome Biology* 2014, 15:R41). The other group reported genome-wide analyses of DHSs and H3K27ac, and showed by the reporter assay that ten of the 14 (71%) candidate enhancers with distal DHS really function for leaf- or flower-specific expression (Zhu et al. *Plant Cell* 2015, 27: 2415–2426). This study by Yan et al. provides thorough analyses of genome-wide dynamism of H3K27ac in flower development and its correlation with DHSs, expression profiling and genome-wide binding profiles of representative transcription factors. Both papers of Zhu et al. and Yan et al. generally came to a similar conclusion, but Yan et al. further conclude that distal enhancers do not contain H3K27ac from series of genomics analyses. My major concern is that this manuscript does not contain any biological evidence to show the regions actually act "enhancers" nor any new positive correlation with other histone marks.

In summary, this manuscript brings together a useful information for some specialists in the community of flower development. I believe that more specialized genomics journals could be a better home for this paper.

Detailed comments:

1) It is very confusing to use the term “stage” to describe the days after dex induction, since their stage 0, 2, 4 and 8 do not seem to perfectly correlate with the conventional flower developmental stages in Arabidopsis. At least it is not clear from the images shown in Fig. 1A.

Response: This whole inflorescence image contains hundreds of synchronized flower bud, it is not easy to show the detail of each single flowers. However, based on previous results and how we performed the experiment, stage 0 of this system represents the inflorescence with 2-5cm height and after 2, 4 and 8 days’ induction, the flowers comes to the stages floral meristem specification (stage 2), floral organ specification (stage 4-5), and floral organ differentiation (stage 7-8). To be more clear, we changed “stages” to “days after induction (DAI)” throughout the manuscript.

2) In page 5, line 123-123, “2 days after induction, stage 4-5” should be “4 days ,,,,.”

Response: Sorry for the mistake. We corrected the mistake.

3) In Fig. 1B, the AP1 locus is shown as a representative gene to show the correlation between H3K27ac, DHS and expression. Please justify to study the endogenous AP1 locus in the ap1 mutant background with an inducible construct of AP1.

Response: Thank you for pointing this out. We have added this information to the figure legend as “*note here the expression of AP1 represents the effect from both endogenous AP1 promoter and AP1 promoter in the an inducible construct of AP1*”

4) In page 10, line 275, “that” should read “than”.

Response: Thank you for pointing this out. We corrected the mistake.

5) In figure 1 legend, line 2 from the bottom, “Promoters were defined as 1kb upstream and downstream of the TSS.” does not make sense to me.

Response: We thank the reviewer for pointing at this lack of clarity. It is better to use “TSS-proximal” instead of “Promoters” in this context. We corrected this in the figure and also in the legend.

6) In figure 5 legend, line 4, “D” should be “B.”

Response: Thank you for pointing at this error. We have corrected the mistake.

Reviewer #4 (Remarks to the Author):

The manuscript by Yan et al., provides solid evidence that enhancers activity is driving stage-specific gene expression. The methodology used is adequate, the results solid and the figures neatly presented. I only have comments concerning the text itself that I believe need to be addressed:

A source of concern comes from the lack of description of the plant material used to perform the various assays. On line 484 the authors state that pAP1:AP1-GR *ap1-1 cal-1* plants were used for mapping stage-specific targets of H3K27ac in flowers, while on line 120 the authors refer to a previously described floral induction system that makes use of a 35S:AP1-GR *ap1-1 cal* line (ref 33; I believe that reference 34 should have been used here). This needs to be clarified, as the use of a strong constitutive promoter would be inappropriate to conduct the work described in the present manuscript and would alone justify rejection.

Response: We thank the reviewer for pointing at the lack of clarity. We indeed used pAP1:AP1-GR *ap1-1 cal-1* line for mapping H3K27ac targets. We cite ref33 because it represents the first use of the DEX-based induction method in the *ap1 cal* background for collection of synchronized flower tissue, which we believe should be appreciated. However, to avoid confusion, we now cite ref34 instead of ref33 here.

In addition, there is no mention of the plant material used for the RNA-seq and DHN-seq assays in the Plant Material and Growth Conditions paragraph. Annoyingly, the authors ended up including this information in the subsections describing the respective assays.

Response: Thank you for pointing out this important lack of information. We have rewritten this part in the Methods section: “*An inducible line in which API protein fused with glucocorticoid receptor (GR) is expressed under API promoter in ap1-1 cal-1 double mutant background (pAPI:API-GR ap1-1 cal-1) was grown for mapping stage-specific targets of H3K27ac in flowers^{31,88}. The same material had been used in the group previously for generating stage-specific open chromatin data in flowers by DNaseI-seq³¹. The seeds were directly sown to soil. After cold treatment for two days, the plants were transferred to phytotrons with long day conditions (16 h light, 8 h dark) at 20°C, at light intensity of 100 $\mu\text{mol}/\text{m}^2/\text{s}$.”*

In its present form I don't see how this manuscript deepens our knowledge of flower morphogenesis other than providing several transcriptomes and landscapes of histone marks and DHSs at several stages of flower development. The manuscript fails to interpret the genomic data in the context of flower morphogenesis. For instance, why is it important for an Arabidopsis plant to have a high frequency of regulatory elements occurrence in flowering genes than on any other genes? On line 386 the authors reported that “Strikingly, binding sites of MADS-domain containing proteins, represented by the master regulators of flower development, were overrepresented in both the proximal and distal DHSs” but failed to discuss the implication of this result. In my opinion, the Discussion should be rewritten in order to better bridge the genomics findings reported in this study and the biological aspects behind flower development. Alternatively the title could be modified by omitting the words “during flower development”, which would better reflect the focus of this manuscript on understanding the role of enhancers in regulating stage specific gene expression in plants.

Response: We thank the reviewer for these useful comments. Following the suggestion from the reviewer (as well as other reviewers), we largely revised our manuscript by including experimental validations (see Fig. 4) and new analyses (see Figs. 5 and 7) to focus more on the key results, e.g., the dynamics and functions of Arabidopsis enhancers.

For example, we stated that “21 of the annotated flowering-time genes⁷⁹ were associated with dynamic enhancers, and 17 (81.0%) of them were differentially expressed during flower development (**Supplementary Table 5**). More than half (53.3%) of the enhancers were bound by at least one of the 15 floral master regulators (**Supplementary Table 5**). These results highlight the importance of enhancers for re-shaping developmental transcriptome dynamics, preferentially via regulation of floral master regulators.” We have rewritten the whole discussion part based on our new results. Firstly, we discussed about the chromatin features and species-specific characteristics of plant enhancers. Next, we discussed about the stage specificity and dynamics of enhancers during flower development and the link between enhancer activity and the stage-specific gene expression pattern. Lastly, based on some new results from the revision, we discussed about the role of enhancer clusters and intronic enhancers in reshaping transcriptome dynamics in flower development. More details information can be found in pages 15-17.

Minor comment:

Line 320 confirmation should read conformation

Response: We apologize for the mistake and thank you for pointing this out. We have corrected the mistake.

Reviewer #5 (Remarks to the Author):

The paper by Yan is well written and the analyses performed are mostly sound.

The authors use H3K27ac signatures and RNAseq combined with prior analyses such as DNase HS to try to define possible enhancers in Arabidopsis during flower patterning. They find that H2K27ac associates only with promoter proximal DNase HS regions, while distal DNase HS regions are devoid of this (or other) histone modifications. The authors go on to suggest (based on DNA sequence conservation, presence of predicted TF binding sites and link to DE genes) that the distal DNase HS sites – but not the proximal ones- are likely to be enhancers.

The study is original, but in its current form not yet fully convincing. My main concern is that none of the newly identified proposed distal enhancers were functionally tested, this could easily have been done with reporter constructs. This needs to be remedied in a revised version.

Response: We thank the reviewer for pointing this out. Following the reviewer's suggestion, we performed experimental validation using a reporter assay. The results showed that 27 of 30 (90%) reporter lines showed ability of triggering GUS gene expression. Please find the section of "Validation of the predicted enhancers" in the Results and a new Fig. 4 for more details.

More specific comments

1. Changes in the level of histone modifications such as H3K27ac could be triggered by addition or removal of the acetyl groups to H3, or by histone octamer removal/turnover. To avoid looking at H3K27ac changes that simply reflect altered nucleosome occupancy, the H3K27ac levels need to be normalized over an independent H3 histone ChIP. This is critical, as TF binding can lead to nucleosome loss.

Response: We thank the reviewer for this very useful comment. In the revision, we analysed H3 ChIP-seq datasets in inflorescences at stages 0 (S0) and 2 (S2), corresponding to H3K27ac ChIP-seq data for S0 and S2, respectively. We also included wild-type inflorescence H3 ChIP-seq data for comparison (**Fig. R4**). From the **Figure R4**, one can see that the level of H3K27ac was dynamically changing between S0 and S2, while the level of corresponding H3 remained the same between the two stages. Accordingly, we found no correlation in peak changes for the mostly significantly changing H3 and H3K27ac (**Supplementary Fig. 17**). Moreover, we observed that overall H3 and H3K27ac dynamics are not correlated (**Supplementary Fig. 18a**), suggesting that changes in H3K27ac are independent from changes in H3 level. Finally, genome browser views of H3K27ac and H3 signals at selected loci, for S0, S2 and inflo time points illustrate that no change in H3 occupancy can account for the observed changes in H3K27ac (**Supplementary Fig. 18b**). Therefore, H3 ChIP-seq data were not used to normalize the H3K27ac level. Due to reorganization of the manuscript, the results on H3K27ac dynamics can be found in **Supplementary Fig. 8**.

Figure R4. Scatterplots showing the relationship of H3 and H3K27ac levels at stages 0 (S0) and 2 (S2; *ap1cal pAPI::API-GR*). H3 data in wildtype inflorescences (WT_inflo) were used for comparison.

Supplementary Fig. 17. The dynamics of H3K27ac is independent on the dynamics of H3. (a) Correlation of the change of H3 levels and H3K27ac levels between stages 0 (S0) and 2 (S2). All H3K27ac peak regions were used in the analysis. (b) Similar to (a). Only the top regions (n=177) with highest changing H3 and H3K27ac were used in the analysis.

Supplementary Fig. 18. Comparison of H3K27ac and H3 ChIP-seq data. (a) Scatterplots showing the relationship of H3 and H3K27ac signals at stages 0 (S0) and 2 (S2; *ap1cal pAPI::API-GR*). H3 data in wildtype inflorescences (WT inflo) were used for comparison. Only the top regions (n=177) with highest changing H3 and H3K27ac were shown. (b) Genome browser views of H3K27ac and H3 signal at selected loci.

2. Computationally predicted TF binding sites are not very useful for predicting factor binding. Only a very small fraction of these are bound in vivo. Most are small and occur on average every 2 kb in the genome. Because of that they are not a good metric for defining enhancers or DNase HS site function as enhancers. If they are to be used, genome-matched random regions (non DNase HS sites at similar genomic locations) should be used to test for significance of enrichment of predicted TF binding sites in the putative distal enhancers.

Response: We thank the reviewer for raising this important point. We now included two types of regions as a control to calculate the significance of enrichment of predicted TF binding sites. For the first control set, we took the direct flanking regions of distal DHSs with the same size. For the second control set, we randomly chose 10000 size-matched regions from the intergenic non-DHS locations. The results show that putative enhancers have more enrichment of TF binding sites than both control sets but slightly less enrichment than promoter regions (Supplementary Fig. 13).

Supplementary Figure 13. Enrichment of predicted transcription factor (TF) binding sites in putative distal enhancers. For comparisons, enrichment was also calculated in enhancer flanking regions (at both sides with a matched size), random intergenic non-DHS regions (n = 10000), promoters (proximal DHSs) and promoter flanking regions. Enrichment score was calculated as the number of TF binding sites per bp in each region. Significance was determined by Welch two sample t-test. DHSs, DNase I hypersensitive sites.

3. The authors use a large number of prior studies (for CNS for example), but do not always properly highlight in the main text that these were prior studies.

Response: We thank the reviewer for pointing out this and we are sorry for the mistakes. We have properly cited the work from prior studies in this revision. For example, we added references where we got the CNS and lincRNA data.

Reviewers' comments:

Reviewer #1 (Remarks to the Author):

In general the paper is improved quite a bit and more focused. In the current manuscript there is some redundancy in the data described (see comments below). Due to the redundancy in combination with the compactness of the text and the enormous amount of panels and suppl figures a reader may get lost in the data, hence the main conclusions. Taking out the redundancy should improve the readability and focus of the paper.

There are still quite a number of unclarities, errors, etc present that should be improved/corrected.

Small typos that are present in the text are not listed in the review. I did not go through the methods section as the rest of the text, figures and tables already took much more time than I had planned spending on the manuscript.

Feedback

- Response at previous line 82. You refer to a previous study by Bao et al. Which study do you mean? There is no reference to Bao et al. in your manuscript.
- 22/27 Please add 'development' before 'stage-specific'
- 32 Please add distal to enhancers
- 73/74. Do you mean extrinsic signal-dependent TFs? The current wording is quite complicated/confusing.
- 87 Reference 36 and 37 should also be included here. They also use DNaseI assays.
- 87. 'for the first time' is true for plants, but not if you take other organisms into account as well.
- 90 'half' should be 'nearly half' (42-45%)
- 92. A ref is missing at end of sentence referring to 'the same group later identified.....'. It is not reference 30. We assume it is a paper showing data on TF binding sites.
- 93. Reference 30 is incorrect. This paper is not about AP1 and SEP3. So it also does not fit in the series of data from the same group.
- 103. Per definition the role of an enhancer is to increase expression of its target gene, so you can't say that the role has not been elucidated. The role for enhancers is in their definition. Same for line 107.
- 151. in link should be 'in line'. I assume 'level' is missing after expression.
- 156. I would change marks into 'enrichment', as it is unclear if marks refers to a verb or a subject.
- 164. It is unclear why you use 'in accordance'.
- Suppl Figure 6 also shows curves that denote the accumulative fraction of H3K27ac peaks. This is however not discussed in the text. It seems to indicate that the distance to the closest H3K27ac peak differs per TFBS. Any idea why this would be?
- 169. Should 'within' be 'between'?
- 169. To which sites does 'these sites' refer? the ones showing the 1.5 fold change or all 12579 sites?
- Suppl Fig8b. In the legend there is a '7' between two words. At the Yaxis at the right in panel it reads 'scaled gene expression level'. in the legend for c it refers to 6 groups, while there are 4. There is no meristem maintenance GO term in panel d.
- 172-182. The explanation mentioned for the lack in correlation between H3K27ac and gene expression levels at some stages in groups 2 and 4 may indeed be an explanation. At the same time, one can then wonder if the observed correlation between H3K27ac and gene expression levels at other stages and groups should also be taken with a grain of salt. Groups 2 and 4 contain many genes. Are a few genes in these groups responsible for the observed patterns? Or are more genes within these groups show these 'aberrations' in the correlation? Furthermore, at which location is the H3K27ac level measured that is taken along in the correlation. At the peak just 3' of the TSS? Elsewhere? Or is the average signal over the entire gene used for the analysis? Taking the peak value at the TSS would be the preferred method of choice.

- Figure 1 legend. If the expression of AP1 represents the endogenous and transgenic AP1 sequence, is the DNase-seq and H3K27ac ChIP-seq also representing both the endogenous and transgenic locus? Or partially? That affects the data interpretation. It may be best to show another example than AP1.
- Figure 1C. In the response you indicated you now say equal number of genes in each group, but I do not see this in the text in the legend. It reads 'divided into 10 groups'.
- Figure 1e. I would spell out stage 2 and Stage 8 instead of mentioning S2 and S8.
- 194-196. I am not convinced by the reasoning in these sentences. The difference in localization at the TSS is logic. At a DHS at the TSS there will be no H3K27ac, as there is a depletion of nucleosomes to allow the transcription machinery to initiate transcription. Hence a concentration of H3K27ac downstream of the TSS. Overthere, it facilitates transcription initiation/elongation. DHSs and H3K27ac are part of the same mechanism, allowing transcription. They do not act separately. Distal of the TSS another histone modification, such as H3K9ac may be more prevalent.
- Suppl Table 1, unlike suggested by the text, does not mention distal intergenic peaks separately.
- 203. 'with a previous study' is lacking after 'The difference'.
- 205. Would be nice to indicate the definition of core promoters between brackets.
- Figure 2A and B in the manuscript are not the same as those in the response to reviewers. Irrespective of that, in 2a you cannot see that the H3K27ac peak is intragenic. The X-axis only reads distance from the TSS, which can also be upstream. In 2B the X-axis is exactly the same as in A, and then it represents the region upstream of the TSS at least for intergenic. What about the intragenic DHS sites shown? These are not discussed in the text. Are the peaks shown in this part of the pannel only present downstream of the TSS? At more or less the same positions? That seems quite unusual I would say. Would these genes also have a DHS just 5' of the TSS? These panels need to be clarified.
- Suppl Figure 9c. The % below would indicate non-coexistence regions. How can these % be below 10% while about half of the DHS sites are not overlapping with H3K27ac regions (in the same bar graph).
- 216-218: This conclusion can already be drawn from Suppl Fig 9a/b. It is not clear what the additional value is of showing 2c. At least the panels in S9a/b and the top panels at 2c are showing the same results. In addition 1C, S5 and S10 show similar results for the location of DHSs and H3K27ac relative to the TSS. There is too much redundancy between the figures.
- 219-221. You need to specify the presumed contradictory component.
- Suppl Fig 11c,d: you say this is similar to Figure 2a,b. That is not the case. 2a, b look quite different, split up in stages.
- 224. Zhu et al report in the legend of Figure 2 a window size of 50 bp, not of 1 kb. They determined the occupancy along +/-5 kb regions, which is something else than a window size of 5 kb. It is actually unclear to me how you would get the graphs you show with a window size of 1 kb. The resolution in the graphs (e.g. S9a,b, S10, S11c-f) is higher than 1 kb. Maybe the 50bp is a too high resolution to see a clear peak?
- Figure 2. Please indicate in a and b panels if the distance is upstream or downstream of the TSS.
- Figure 2d-f (and Suppl Fig 12). Are the signals at the distal DHSs oriented towards the nearest gene? Not oriented at all? It may make sense to orient them such that highest signal for a modification is at one and the same site, downstream to be in line with the proximal DHSs.
- Figure 2f. Are the DHS data for human also oriented towards the nearest gene?
- Figure 2f/g. I am confused there is no H3K36me3 marking at the gene body at the proximal DHSs. This is generally observed in mammals (see e.g. Fig3E in <http://www.genesdev.org/cgi/doi/10.1101/gad.313973.118>; this is the first example I found, there are plenty more), so I assume that the dataset used does not have proper H3K36me3 ChIP-seq data (e.g. due to a low quality antibody).
- 259-261. It is not clear why you would examine genes 'with distal DHSs alone' (fig 3a). It makes sense those are not highly expressed. Only genes proximal DHSs are likely to be expressed. The proximal DHS indicates the binding of the transcription machinery. Realize that distal DHSs may

target a gene that is not the closest gene.

- Thinking about the assumption that the closest gene is the target gene, how many of the distal DHS sites are downstream vs upstream of the presumed target gene?
- Suppl Table 2 first page, motif analysis: it should read 'TF binding sites enrichment'. Without 'sites' one suggests that ChIP-seq data is generated for all these TFs.
- Suppl Table 3. It would be good to indicate that the genes listed are the closest genes. They are not necessarily the target gene. That should actually also be clear in the text, that this assumption is used.
- 267-269. A reference to a table/Figure is missing.
- 269-272/ Suppl Fig 13. The promoter proximal DHSs seem to be even more enriched for TF binding sites. Would be good to mention this in the text.
- 272. What is meant with widespread TF regulation? This is a bit vague. Regulation by multiple TFs?
- Fig3e. Please add 'DHSs' to the sentence. So 'peak centers of promoter...enhancer 'DHSs'.
- 285, 287. Fig3e. There are no top/bottom panels.
- 282-283. Here is mentioned that one examined sequence conservation across crucifer species, in the legend only Arabidopsis is mentioned, and in the figure panel (3e) it reads Arabidopsis. I assume that the main text should read that one examines sequence conservation between Arabidopsis and the crucifers, or in 3e it should not read Arabidopsis.
- 288-289. It is not clear why the high conservation of the gene body in the vicinity of the promoter would explain the relatively low conservation of Arabidopsis promoters.
- 289-293. It is not clear why plotting the distribution of PhastCons scores for promoters in a strand-specific manner that you do see peaks. Aren't the promoters oriented to begin with also for the left panel (e.g. always having the TSS 3' of the promoter). I assumed you did so.
- 302. You say that the distal DHSs have conserved functional features. I assume you want to refer to enhancers instead. In addition, the word functional can only be added after the validation.
- 308-309. 'dual orientated constructs (..Fig 4a)'. This wording is not entirely clear (suggests there are 2 orientations in one construct) and in addition suggests there is a schematic drawing of the dual orientated construct in Fig4a, which is not true. similarly 'dual orientated assay' in line 314. A GUS assay itself is not dual.
- 315. 'Identical' is overinterpretation of the data. You should say highly similar.
- 327. The use of candidate enhancers in the 3' regions of genes should already be mentioned at the bottom/top of page 10/11.
- Fig4a. The legend mentions that the blue color intensity correlates (with) the percentage of transgenic plants with detected GUS expression. This seems to refer to the first column rather than the entire panel (and would that be independent transgenic plants rather than transgenic plants as such?). I assume that the intensity in the other columns refers to the intensity of the GUS staining?
- 4b. The picture that supposedly shows inflorescence with early flowers is not clear. Where are the flowers? I seem to see leaves instead.
- Fig 4g is showing the FT gene, not the JAG gene. 4h is not FT, but JAG. FT carries an enhancer 5 kb upstream. Why is this not visible in the profile.
- Fig 4. I miss the scale of the profiles shown (in other words, what is the size of the sequence regions looked at?). The scale is also important to know what is considered still promoter and what not.
- Figure 4b-i. There is data shown that is not discussed in the text. The DHS profiles are not mentioned. If showing the data, it should be discussed.
- Figure 4b-i. The TFBSs are shown in different colors, sometimes with indication of the relevant stage, sometimes without. In the latter case, which tissue is used for those analysis?
- Figure 4b-i. Seeing the GUS data together with the DHS data (please indicate at the graphs that this is DHS data!) also makes me wonder how the GUS expression looks like in the tissues harvested for the DHS analysis. Those stages are younger (Fig 1) than the flowers shown with GUS expression in Fig 4.
- Suppl Table 4. Orientation is written incorrect. What does the yellow background color and red letters mean? I assume the blue indicates the GUS intensity? Please clarify. The nrs in the blue

cells represent the % of plants of a transgenic showing Some of the nrs have too many digitals behind the comma to make sense.

- 345-347. Can you also provide the average nr of TF binding sites?
- Fig5a. BS of floral regulators sounds vague and is not specified in the legend. BS also stands for bisulfite...
- 342/Fig5b. Do all 1979 genes play some role in floral development? So does it e.g. make sense that 635 promoters are bound by BLR? The legend reports enrichment scores, what do these represent? enrichment of what relative to what? The color is not indicative for the numbers, hence my question.
- Fig5c & main text. How many stages are taken into account to determine if a gene is called DE? Does this refer to the 1.5 fold used before?
- 352. Every gene has a promoter, otherwise it cannot be transcribed. In addition please use a word like predicted in combination with enhancers. IS also true for e.g. line 374.
- 355-358. Please provide the average nr of bound regulators. The difference may be significant, but it seems the number does not double; 'Significant more' kind of suggests it does.
- 358. 'Support' and 'may' are a double uncertainty. Please delete 'may'.
- 364. 'regardless' is a bit overdone. The largest distance observed to my knowledge is 1.3 Mb.
- 365-366. This was also shown by Zhu before, and probably also by older, non genome-wide, papers.
- 371-373. Wasn't this (using the nearest neighbour strategy) already done for data shown earlier in the manuscript? For example, for the transgenics I also assume you already determined what would be the nearest neighbor.
- 376. 'dominantly grouped'?? What do you mean?
- Suppl14 legend. RPM would represent enhancer activity in reads? I do not know such measurement. Do you mean DNaseI sensitivity? Clarify. Also true for legend Fig 7e.
- Fig6,SupplFig8. What is the reasoning to do similar analysis on four groups and later on 2 groups? Why not focus on the latter analysis and delete the first from the manuscript? Which conclusions are drawn based on the first analysis that cannot be drawn based on the latter?
- Fig6c. How come that the numbers (266 & 329) do not add up to 810? Where are there enhancers/promoters missing?
- Line 385-386 do not connect to 386-389. It is unclear from the text why the latter exemplifies the first.
- 389-390. Sentence does not flow. N total there are three, so something like 'and another distal...'
- Fig 6d. What do the horizontal red bars below the plot refer to? I assume a potential enhancer (the upstream one is however lacking the bar). The main text seems to ignore one of the two element in the intron (bound by Sep3, AG etc). In addition, delete 'novel' in the legend and text. You do not show this enhancer just arose in a particular accession or something like that. The TFBS data is not discussed in the text. IF you show it, better also say something about it.
- 401-402, Fig 6f. Where does the chromatin accessibility come in at this figure? What does actually AP1, 2h refer to? The legend should make clear that the columns represent stage 2-specific enhancers and stage 8-specific enhancers. Now it takes a while before one understands the figure.
- 408. Why 'in more general terms'? LBD is also general.
- Suppl table 5. Target genes should read predicted target genes. Page Moif analysis: the nrs in the fold change column do not seem to be right. E.g. 10.6/0.6 is not 0.048 (top row). In the enhancer dynamics table the nrs in the log2 fold change column also do not seem to make sense. The explanation for the last page 'the number of bound floral regulators in each enhancer' does not fit the content of the page.
- 422. Delete 'distal'. That is commonly used for intergenic regulatory elements, not for intragenic.
- 426 and elsewhere should read predicted intronic enhancers (or something like that).
- 429. The text says 158 genes, the table lists 181 GO terms. How does 158 & 181 relate to each other? And which gene is connected to which row?
- Fig7. In a there are 724 intronic enhancers, in c there are 700. What happened with the 24 that

are missing in c? Similar Q for intergenic enhancers. legend for b should read predicted target genes. Why is 'or 3'UTRs' between brackets?

- Fig7e. The numbers in between brackets for the top and bottom plot differ. Are the nrs in () for the bottom panel actually also the nr of genes in each category? How is the enhancer dynamics measured in case of more than 1 enhancer? Do we look at the average 'activity' of e.g. 4 enhancer? Something else? Enhancer activity should be defined as well.
- Fig7f. How do the data translate in nr of TFBSs? Are these nrs similar as reported earlier in the manuscript for intergenic enhancers and promoters? How come that here there are 4844 intergenic enhancers again, and 724 intronic ones? (see point above as well). Which collection TFBS is looked at for this analysis? The text reads developmental master TFs. How many are these and how are these defined? Are these the same as used for Fig5? There is not a reference in the text referring to a specific paper.
- 443. You suggest that Fig7f shows that genes associated with multiple enhancers have a significant enrichment in binding sites for TFs. Fig7f however does not show the enrichment at enhancer elements that are part of a cluster vs single enhancer elements.

- Fig 7g. Different putative enhancers are indicated and numbered. Not all peaks having a bar are indicated/numbered. E.g. the one next to nr1. Please clarify. Please provide the scale to allow better interpretation of the figure.
- 449. 'stretched enhancer elements'? Personally I never heard this term. Super enhancer is much more commonly used.
- 449-451. What do you mean with similar arrays? Similar numbers? Compared to single enhancers? I assume you mean something else given you refer to a feature of super enhancers. Please clarify the text.
- 461-63. This is the choice made by which the enhancer candidates are identified (and includes accessibility (DHSs) as well). This was supported by finding known/putative enhancers back, but it has not been tested e.g. if DHSs without H3K9ac would be able to enhance expression. It would be appropriate to also e.g. cite Zhu et al. and Qiu et al.
- 468 & 473-475. I would delete 'compact genome'. I do not see any reason why a distal enhancer could not be marked by H3K27ac when the genome is more compact. Rice and maize are by the way monocots. I have no clue if there has been divergence in using this mark between monocots and dicots.
- 472. The 12.7% is this coming from your study? Or should there be ref added?
- 480. Han et al did not test any connection with H3K27ac, so you can't conclude the effect is not via H3K27ac. Mutants in HATs have in addition pleiotropic effects so promoting flowering may be an indirect effect.
- 487-490. I miss a reference for the data showing this is opposite in Arabidopsis.
- 486-491. It is not clear why the focus is on interactions/co-occupancy with other chromatin marks, rather than on divergence in the specific marks used to activate enhancer sequences. Acetylation makes chromatin more accessible, allowing access to the 'transcription machinery'. Maybe the chromatin at Arabidopsis enhancer sequences does not 'require' acetylation to open up chromatin; maybe the binding of a pioneering TF is sufficient to recruit chromatin remodelling proteins that in turn allow access to other proteins; or there is an acetylation mark added that is not identified yet.
- H3K4me1 is associated with both inactive and active enhancers in animals.
- 494-496. You elaborate on the intergenic localization of H3K4me3. How many of your predicted enhancers carry H3K4me3? In other words are there also DHSs detected at poised enhancers?
- Ref 78 lacks page numbering.
- 497-499. To study species-specific characteristics multiple species need to be studied. Not just Arabidopsis.
- 515-517. The first sentence is a general statement; in such case this will also be true for other TFs than TFs regulating flowering.
- 516-517. Are the master regulators a subset of the flowering genes? What is the criterium to be called a master regulator versus a regulator?
- 518. Formulation needs to be improved. Activity cannot be enriched by binding...

- 530-534. In figure 7 you show DHSs are also located downstream of the TSS, intronic, and also in the 3'UTR.
- 537. 'So' should be 'and'.
- 540. The use of 'Thus, they should' is not appropriate overhere. The previous sentence does not say anything about both activation and repression.
- 541. Similarly, 'in line with this' is not appropriate.
- 543-544 Last part of the sentence is a bit confusing. I would change it to ...gene expression through eRNA transcription.'
- 551. I am not sure this is still under debate 3 years from ref 88.
- 553. Not clear to what 'location' refers to.
- 554-555. You did not show that individual enhancers in a cluster bind more TFs than individual enhancers not being part of a cluster. You do however suggest overhere that you did.
- 558-559. Why not also for 3'UTR enhancers or single intergenic enhancers?
- 561-562. Accessibility is part of the activation process and it is triggered by binding of TFs and allows the binding of subsequent protein factors.
- 562-564. This has been reported by others before. The message is true focusing on

Reviewer #2 (Remarks to the Author):

Authors have revised or deleted most of the text related to the issues we talked about. We might have preferred that they integrate more of the responses they gave in the rebuttal into the actual manuscript. For example, more details comparing and contrasting Zhu et al enhancer prediction vs the current study, and seedling vs flower tissue comparisons. Otherwise, the rest of the revision looks fine.

Reviewer #3 (Remarks to the Author):

In the revised manuscript, the authors newly and nicely performed functional study of 22 candidate enhancers, and extended the bioinformatics analyses. Although they claimed that they have added experimental validation results for predicted enhancers, I still have one concern on the result of AP1 (Fig 3a and f). The AP1 enhancer lines with a 35S minimal promoter showed GUS expression in "joints" and sepals, but not in petals. Further, the majority of lines showed expression in stamens and carpels, which should have no AP1 expression. These results do not support an idea that the predicted AP1 enhancer is a real one. They obtained only 4 lines and their results may not reflect the real activity. This problem should be resolved.

Reviewer #5 (Remarks to the Author):

The authors did - for the most part- address my concerns. The enhancer tests in particular add to the study. There are two points in regards to this experiment that need addressing. Firstly, the size of the enhancer fragments tested in vivo should be included in the main text. Secondly, the authors should state in the main text how the expression pattern observed deviates from the endogenous expression pattern of the genes for each enhancer tested. It does deviate quite a bit and there are good reasons for this (additional regulatory information is missing), but the reader should be made aware of the partial recapitulation of the endogenous pattern.

Reviewers' comments:

Reviewer #1 (Remarks to the Author):

In general the paper is improved quite a bit and more focused. In the current manuscript there is some redundancy in the data described (see comments below). Due to the redundancy in combination with the compactness of the text and the enormous amount of panels and figures a reader may get lost in the data, hence the main conclusions. Taking out the redundancy should improve the readability and focus of the paper.

There are still quite a number of unclarities, errors, etc present that should be improved/corrected.

Small typos that are present in the text are not listed in the review. I did not go through the methods section as the rest of the text, figures and tables already took much more time than I had planned spending on the manuscript.

Feedback

- Response at previous line 82. You refer to a previous study by Bao et al. Which study do you mean? There is no reference to Bao et al. in your manuscript.

Response: Thank you for pointing out this mistake. The study we intended to mention here is by Bo Zhu et al. The citation has been corrected in the manuscript.

- 22/27 Please add 'development' before 'stage-specific'

Response: We have now added 'developmental' before 'stage-specific' in the revised version of the manuscript.

- 32 Please add distal to enhancers

Response: Thank you for the suggestion. We have added "distal" in the revision.

- 73/74. Do you mean extrinsic signal-dependent TFs? The current wording is quite complicated/confusing.

Response: Thank you for pointing this out. We have changed this sentence to "*Enhancers appear to function as integrated platforms for binding of multiple TFs, often including lineage- and signal-determining (intrinsic or extrinsic cue-dependent signal) TFs*". We hope that it is clearer now.

- 87 Reference 36 and 37 should also be included here. They also use DNaseI assays.

Response: Thank you for pointing this out. We have added these two references.

- 87. 'for the first time' is true for plants, but not if you take other organisms into account as well.

Response: We thank the reviewer for this comment. Yes, we wanted to state that this is the first report using DNaseI-seq to map open chromatin in plants. Therefore, we agree with the reviewer and have added the term of “in plants” after “for the first time”.

- 90 ‘half’ should be ‘nearly hal’ (42-45%)

Response: We appreciate the comment, which makes the description more precise. Accordingly, we have changed the sentence to “42.3% and 44.5% of DHSs in seedlings and in callus, respectively, reside in intergenic regions”.

- 92. A ref is missing at end of sentence referring to ‘the same group later identified.....’. It is not reference 30. We assume it is a paper showing data on TF binding sites.

Response: We think the reference 30 is proper here, since the authors of ref. 30 compared DHSs distribution and binding patterns of both AP1 and SEP3 and found that “*DH sites derived from flower tissue were found to be associated with 1843 (94.9%) of the 1942 AP1 binding sites (Figure 7A). Similarly, 3841 (89.7%) of the 4281 SEP3 binding sites overlapped with DH sites identified in flower tissue (Figure 7B).*” by using the binding sites data of AP1 and SEP3 from earlier reports of our group.

- 93. Reference 30 is incorrect. This paper is not about AP1 and SEP3. So it also does not fit in the series of data from the same group.

Response: Thank you for the comments. Indeed, the paper of ref. 30 is not about AP1 and SEP3 but the authors of this study mentioned the association between DHSs and AP1/SEP3 binding patterns. The sentence in line 93 was one of the results in the paper of ref. 30. The original sentences were “*DH sites derived from flower tissue were found to be associated with 1843 (94.9%) of the 1942 AP1 binding sites (Figure 7A). Similarly, 3841 (89.7%) of the 4281 SEP3 binding sites overlapped with DH sites identified in flower tissue (Figure 7B).*” We think it is proper to cite ref. 30 here. Both ref. 29 and ref. 30 are from Dr. Jiming Jiang group. To avoid confusion, we changed the sentence to “*The same group later identified a high overlap between TF occupancy and DHSs in Arabidopsis thaliana leaves and flowers and they found that around 90% of APETALA1 (AP1) and SEPALLATA3 (SEP3) binding sites overlap with DHSs³⁰*” in the revision.

- 103. Per definition the role of an enhancer is to increase expression of its target gene, so you can’t say that the role has not been elucidated. The role for enhancers is in their definition. Same for line 107.

Response: Indeed, the function of enhancer is to enhance expression of its target gene(s). However, how enhancer dynamics determine spatiotemporal gene expression patterns in plants is not clear, especially in genome-wide manner. In order to avoid misunderstanding, we changed the sentence into “*Although these studies spark the renewed interest in the investigation of plant enhancer elements, the mechanism of dynamic enhancer activities to trigger spatiotemporal gene*

expression, especially in genome-wide manner in plant development has not yet been elucidated.” The sentence at line 107 was changed to “... *to dissect their roles in the control of stage-specific gene expression patterns ...*”

- 151. in link should b ‘in line’. I assume ‘level’ is missing after expression.

Response: Thank you for the remark. Following the suggestions of the reviewer, we have changed it to “in line with” and have added “level” after “steady high expression” in the revision.

- 156. I would change marks into ‘enrichment’, as it is unclear if marks refers to a verb or a subject.

Response: Thank you for the suggestion. We have changed “marks” into “enrichment”.

- 164. It is unclear why you use ‘in accordance’.

Response: We thank the reviewer for the comment. “in accordance” might not be a proper wording here, we have used “Interestingly ” instead.

- Suppl Figure 6 also shows curves that denote the accumulative fraction of H3K27ac peaks. This is however not discussed in the text. It seems to indicate that the distance to the closest H3K27ac peak differs per TFBS. Any idea why this would be?

Response: Thank you for the comments. We have discussed this in the text according to the reviewer’s advice by adding the following sentences: “*However, the distance of binding sites to their closest H3K27ac peaks varies for different regulators (Supplementary Fig. 6a). Given that H3K27ac mainly marks TSS-proximal regions, the above observation indicates that different TFs have different binding preference in genomic locations with respect to the TSS.*”

- 169. Should ‘within’ be ‘between’?

Response: Thank you for the suggestion. We may want to use “among” here since we are comparing data from four time-points.

- 169. To which sites does ‘these sites’ refer? the ones showing the 1.5 fold change or all 12579 sites?

Response: We meant the 2913 sites that showed 1.5 fold change (as shown in Supplementary Fig. 8 and Supplementary Table 2). To avoid confusion, we have added the number of “2913” before “sites”.

- Suppl Fig8b. In the legend there is a ‘7’ between two words. At the Yaxis at the right in panel it reads ‘scaled ene expression level’. in the legend for c it refers to 6 groups, while there are 4. There is no meristem maintenance GO term in panel d.

Response: Thank you for pointing these mistakes out. We have corrected them according to the comments of the reviewer. Regarding the comment on “no meristem maintenance GO term”, we

apologize for not having changed the text accordingly after we performed new analysis based on the suggestions from the previous review comments.

- 172-182. The explanation mentioned for the lack in correlation between H3K27ac and gene expression levels at some stages in groups 2 and 4 may indeed be an explanation. At the same time, one can then wonder if the observed correlation between H3K27ac and gene expression levels at other stages and groups should also be taken with a grain of salt. Groups 2 and 4 contain many genes. Are a few genes in these groups responsible for the observed patterns? Or are more genes within these groups show these ‘aberrations’ in the correlation? Furthermore, at which location is the H3K27ac level measured that is taken along in the correlation. At the peak just 3’ of the TSS? Elsewhere? Or is the average signal over the entire gene used for the analysis? Taking the peak value at the TSS would be the preferred method of choice.

Response: We thank the reviewer for these comments. To address the concern of the reviewer, we would like to state that: 1) the observed patterns are representative of more genes within these groups as we plotted the median value in the Supplementary Fig. 8c.; 2) we indeed used the signal value of H3K27ac peaks at the TSS in our analysis (as mentioned in the figure legend in the revision).

- Figure 1 legend. If the expression of AP1 represents the endogenous and transgenic AP1 sequence, is the DNase-seq and H3K27ac ChIP-seq also representing both the endogenous and transgenic locus? Or partially? That affects the data interpretation. It may be best to show another example than AP1.

Response: Thank you for the comments. Indeed, all of the expression, DNase-seq and H3K27ac peak data represent both the endogenous and transgenic locus since the AP1 inducible construct contains both a 3.3 kb TSS upstream region and the whole genomic coding sequences, which cover the DHSs peak and H3K27ac enriched region at the *AP1* locus. The transgene reproduces the expression pattern and all the inducible regulation is just about the localization to the nucleus. Both transgene and endogenous locus should have very similar chromatin structures. Since we have already provided *AP3* as another example (besides *AP1*), we would like to keep the Figure 1b in the current way.

- Figure 1C. In the response you indicated you now say equal number of genes in each group, but I do not see this in the text in the legend. It reads ‘divided into 10 groups’.

Response: We have now provided the number (n=2083 genes in each group) in the legend of the revision. “*In the analysis, the expressed genes were equally divided into 10 groups (n=2083 genes in each group)*”

- Figure 1e. I would spell out stage 2 and Stage 8 instead of mentioning S2 and S8.

Response: Thank you for the suggestion. We have changed “S2 and S8” into “*stage 2 and stage 8 of flower development*”.

• 194-196. I am not convinced by the reasoning in these sentences. The difference in localization at the TSS is logic. At a DHS at the TSS there will be no H3K27ac, as there is a depletion of nucleosomes to allow the transcription machinery to initiate transcription. Hence a concentration of H3K27ac downstream of the TSS. Overthere, it facilitates transcription initiation/elongation. DHSs and H3K27ac are part of the same mechanism, allowing transcription. They do not act separately. Distal of the TSS another histone modification, such as H3K9ac may be more prevalent.

Response: Thank you for the insightful comment. We agree with the reviewer that TSS proximal DHSs and H3K27ac that are located downstream of the TSS could act via a common mechanism to regulate transcription. However, the different distribution patterns of DHSs and H3K27ac in the intergenic regions indicate that DHSs, especially the ones in distal regions, may have a different function compared with H3K27ac. To make it clearer, we have changed the sentence into “*In addition, the difference in localization of H3K27ac sites and DHSs with respect to the TSS in distal intergenic regions indicates that they may activate gene expression via distinct mechanisms*”.

• Suppl Table 1, unlike suggested by the text, does not mention distal intergenic peaks separately.

Response: Thank you for pointing this out. We have included distal intergenic peak information in the Supplementary Table 1.

• 203. ‘with a previous study’ is lacking after ‘The difference’.

Response: We have edited the sentence as “*This difference with the previous study led us to...*” in the revision.

• 205. Would be nice to indicate the definition of core promoters between brackets.

Response: We have included the definition of core promoter “within 1kb upstream of the TSS” in the revision.

• Figure 2A and B in the manuscript are not the same as those in the response to reviewers. Irrespective of that, in 2a you cannot see that the H3K27ac peak is intragenic. The X-axis only reads distance from the TSS, which can also be upstream. In 2B the X-axis is exactly the same as in A, and then it represents the region upstream of the TSS at least for intergenic. What about the intragenic DHS sites shown? These are not discussed in the text. Are the peaks shown in this part of the pannel only present downstream of the TSS? At more or less the same positions? That seems quite unusual I would say. Would these genes also have a DHS just 5’ of the TSS? These panels need to be clarified.

Response: To answer the questions of the reviewer, we would like to point out that: 1) the reviewer is right that the old Fig. 2a did not tell that the H3K27ac peak is intragenic. Given that most (>95%) H3K27ac peaks are intragenic (as shown in Fig. 1d), we thus only plotted the data

of intragenic H3K27ac peaks in the revised Fig. 2a. The corresponding sentence is rephrased as “Consistently, the center of most (>95%) H3K27ac peaks is present in intragenic regions (Fig. 1d) and these intragenic H3K27ac sites locate at approximately 300 bp (median value) downstream of the TSS (Fig. 2a)”; 2) in the old Fig 2a and 2b, the distance was always calculated as the absolute value. Intergenic DHSs locate upstream of the TSS while intragenic DHSs are downstream of the TSS. Intragenic DHSs tend to locate far away from the TSS while intergenic DHSs are either near the TSS (at ca. 120 bp) or far away from the TSS (>1 kb). To make it clear, we have now presented the distance for intergenic peaks as minus values while distance for intragenic peaks as positive values in the revised Fig. 2 (as shown in the figure below). We apologize that the label for intergenic and intragenic DHSs were wrongly marked in the old Fig. 2b. We have corrected this in the revised figure. We have also updated the corresponding description (including the text at page 8 and the figure legend) in the revision.

Figure 2a-c. (a) Distribution of the distance of H3K27ac peak summits from the transcript start site (TSS). Only intragenic H3K27ac peaks were used in the analysis. (b,c) Distribution of the distance of the center of DNase I hypersensitive sites (DHSs) from the TSS. Intragenic DHSs are downstream of the TSS (b) while intergenic DHSs are upstream of the TSS (c). (a-c) The number of H3K27ac peaks or DHSs is shown in parentheses. Note that the distance for intragenic peaks is presented as positive (+) values while the distance for intergenic peaks is shown as minus (-). The x-axis is log₁₀-transformed.

• Suppl Figure 9c. The % below would indicate non-coexistence regions. How can these % be below 10% while about half of the DHS sites are not overlapping with H3K27ac regions (in the same bar graph).

Response: The reviewer is right that the % in parentheses below the number of genes in the DHS category (green bar) or below the number of genes in the H3K27ac category (blue bar) indicate non-coexistence regions. This is exactly what we wrote in the figure legend "values in parentheses show the percentage of non-overlapping regions". To calculate the total % of non-overlapping regions, one has to add the % mentioned in the green and blue bars (the total non-overlapping % then reaches c.a. 55%-63%).

• 216-218: This conclusion can already be drawn from Suppl Fig 9a/b. It is not clear what the additional value is of showing 2c. At least the panels in S9a/b and the top panels at 2c are showing the same results. In addition 1C, S5 and S10 show similar results for the location of DHSs and H3K27ac relative to the TSS. There is too much redundancy between the figures.

Response: We thank the reviewer for pointing these potential redundancies out. Fig. 2 and Suppl Fig 9a/b are related but they are different. In Suppl Fig 9a/b, we intended to illustrate that nearly all the H3K27ac sites resided within 1 kb from a nearby DHS while a considerable proportion of DHSs appeared to be isolated more than 1 kb apart from H3K27ac sites. In Fig. 2c (as well as Suppl Fig 10), we intended to illustrate that these isolated DHSs tend to be located in TSS-distal regions. We agree with the reviewer that Fig. 2c and Suppl Fig 10 are indeed redundant (Fig. 2c for data at stage 0, while Suppl Fig 10 for all stages). In the revision, we thus removed Fig. 2c to avoid redundancy.

• 219-221. You need to specify the presumed contradictory component.

Response: We rephrased this sentence to "*These observations are contradictory to the former report from Zhang et al., where the authors found clear enriched H3K27ac modification surrounding enhancer regions using data generated from leaf and mixed flower tissues in Arabidopsis thaliana*³⁶".

• Suppl Fig 11c,d: you say this is similar to Figure 2a,b. That is not the case. 2a, b look quite different, split up in stages.

Response: In order to make it clearer, we have written the legend for Suppl. Figure 11c-d, as "(c) *Distribution of the distance between H3K27ac sites (midpoints) and the TSS of their nearest genes.* (d) *Distribution of the distance between DHSs (peak summits) and the TSS of their nearest genes.*"

• 224. Zhu et al report in the legend of Figure 2 a window size of 50 bp, not of 1 kb. They determined the occupancy along +/-5 kb regions, which is something else than a window size of 5 kb. It is actually unclear to me how you would get the graphs you show with a window size of 1 kb. The resolution in the graphs (e.g. S9a,b, S10, S11c-f) is higher than 1 kb. Maybe the 50bp is a too high resolution to see a clear peak?

Response: We thank the reviewer for the comment. We apologize that the "window size" may not be a proper wording here. We meant to say that the region size (5 kb) chosen by Zhu et al may be too broad to overlap with distal H3K27ac sites (as observed in their Figure 2 that shows enriched H3K27ac signal surrounding enhancers). However, using 1-kb-regions in our analysis, we did not observe such enrichment surrounding the enhancers. In addition, we also tried different WINDOW sizes (100 bp, 150 bp) as suggested by the reviewer, the conclusion remained the same. In the revision, the corresponding sentences were rewritten as "*The differences between our report and that of Zhang et al. likely result from the difference in the sizes of regions chosen for the analysis (5 kb in Zhang et al.³⁶ vs 1 kb in our study). A 5 kb*

enhancer flanking region may overlap with the H3K27ac signal from distal sites because the Arabidopsis thaliana genome is very compact with an average size of ~3 kb for intergenic regions.”

• Figure 2. Please indicate in a and b panels if the distance is upstream or downstream of the TSS.
Response: We thank the reviewer for the suggestion. We have added a “-” sign in front of the numbers for intergenic DHSs (upstream the TSS) and “+” for intragenic DHSs (downstream the TSS). We have made this clear in the revised figure legend.

• Figure 2d-f (and Suppl Fig 12). Are the signals at the distal DHSs oriented towards the nearest gene? Not oriented at all? It may make sense to orient them such that highest signal for a modification is at one and the same site, downstream to be in line with the proximal DHSs.

Response: Yes, all the signals had been oriented towards the nearest gene, as suggested by the reviewer in the previous revision.

• Figure 2f. Are the DHS data for human also oriented towards the nearest gene?

Response: Yes. The analysis is the same for human, Arabidopsis and rice.

• Figure 2f/g. I am confused there is no H3K36me3 marking at the gene body at the proximal DHSs. This is generally observed in mammals (see e.g. Fig3E in <http://www.genesdev.org/cgi/doi/10.1101/gad.313973.118>; this is the first example I found, there are plenty more), so I assume that the dataset used does not have proper H3K36me3 ChIP-seq data (e.g. due to a low quality antibody).

Response: We appreciate the reviewer’s comment. We downloaded the processed ChIP-seq data (BED and bigWig files) from the ENCODE project and generated all the plots in the same way. We have ruled out any bias in the analysis. We agree with the reviewer that the unusual pattern of H3K36me3 signal may be due to the problems of the original ChIP-seq data, e.g. poor quality of H3K36me3 antibody. We have mentioned this issue in the revised figure legend.

• 259-261. It is not clear why you would examine genes ‘with distal DHSs alone’ (fig 3a). It makes sense those are not highly expressed. Only genes proximal DHSs are likely to be expressed. The proximal DHS indicates the binding of the transcription machinery. Realize that distal DHSs may target a gene that is not the closest gene.

Response: Thank you for the comments. In the aim of leading a complete analysis, we wanted to look at the impact of both distal and proximal DHSs on gene expression, which had never been addressed in *Arabidopsis* before. As commented by the reviewer, it turned out that genes with distal DHS alone are lowly expressed. This could be (i) because the distal DHS plays an enhancer function on its closest gene, thus mediating spatial and/or temporal control of expression but not directly determining its level (to this regard, the specificity of the enhancer turning on transcription in specific cells only may explain the low expression level of the gene);

or (ii) as suggested by the reviewer, because the distal DHS targets another gene than the closest one.

- Thinking about the assumption that the closest gene is the target gene, how many of the distal DHS sites are downstream vs upstream of the presumed target gene?

Response: We thank the reviewer for bringing up this interesting point. We looked into our data again and found that about two-thirds of distal DHSs are upstream of their target genes and one-third downstream of their target genes. This information has been added in the revision.

- Suppl Table 2 first page, motif analysis: it should read ‘TF binding sites enrichment’. Without ‘sites’ one suggests that ChIP-seq data is generated for all these TFs.

Response: We thank the reviewer for the suggestion. We have renamed “motif analysis” to “TF binding sites enrichment” in the Suppl Table 2.

- Suppl Table 3. It would be good to indicate that the genes listed are the closest genes. They are not necessarily the target gene. That should actually also be clear in the text, that this assumption is used.

Response: We thank the reviewer for the suggestion. We have changed the “Gene” column to the “Closest Gene” in Suppl Table 3 and mentioned that this assumption is used in the revision.

- 267-269. A reference to a table/Figure is missing.

Response: We thank the reviewer for pointing this out. We have included the data in the Suppl Table 3 and added a reference to this table.

- 269-272/ Suppl Fig 13. The promoter proximal DHSs seem to be even more enriched for TF binding sites. Would be good to mention this in the text.

Response: We thank the reviewer for the comment. Yes, it is true that proximal DHSs are slightly more enriched for TF binding sites than distal enhancers. Following the suggestion of the reviewer, we have added this information in the text.

- 272. What is meant with widespread TF regulation? This is a bit vague. Regulation by multiple TFs?

Response: We agree that “Regulation by multiple TFs” is a better term here. We have changed the text accordingly.

- Fig3e. Please add ‘DHSs’ to the sentence. So ‘peak centers of promoter...enhancer ‘DHSs’.

Response: Thank you for the suggestion. We have modified the whole sentence following the suggestions of the reviewer.

- 285, 287. Fig3e. There are no top/bottom panels.

Response: We have changed “top/bottom” to “left/right”, respectively in the text.

• 282-283. Here is mentioned that one examined sequence conservation across crucifer species, in the legend only Arabidopsis is mentioned, and in the figure panel (3e) it reads Arabidopsis. I assume that the main text should read that one examines sequence conservation between Arabidopsis and the crucifers, or in 3e it should not read Arabidopsis.

Response: Thank you for the comments. Yes, we examined conservation across crucifer species, taking Arabidopsis as focus. We apologize for the unclear description in the legend. We have modified both the text and figure legend accordingly.

• 288-289. It is not clear why the high conservation of the gene body in the vicinity of the promoter would explain the relatively low conservation of Arabidopsis promoters.

Response: Thank you for the comment. We have rephrased this sentence to “... we observed that *Arabidopsis thaliana* promoters identified in this study were less conserved than their surrounding regions (Fig. 3e), due to the common existence of protein-coding genes or natural antisense transcripts in promoter surrounding regions.” We hope that it is clearer now.

• 289-293. It is not clear why plotting the distribution of PhastCons scores for promoters in a strand-specific manner that you do see peaks. Aren't the promoters oriented to begin with also for the left panel (e.g. always having the TSS 3' of the promoter). I assumed you did so.

Response: The reviewer is right that the promoters were not oriented for the left panel since DHS peaks have no orientation by definition. However, it is a great suggestion from the reviewer to plot the PhastCons score in an orientation-dependent manner. For this, we considered the orientation of proximal DHSs as the orientation of their target genes and regenerated the Figure 3e. We also changed relevant description (including main text and figure legend) in the revision.

Figure 3e. Distribution of phastCons score around the peak centers of promoter (blue) and enhancer (orange) DHSs in Arabidopsis. The distribution of phastCons score for promoter and enhancer DHSs was plotted in a strand-specific manner, using the orientation of their target genes. As a control, 10000 random intergenic non-DHS regions were used to calculate the distribution of phastCons score (grey). PhastCons scores between Arabidopsis

and other crucifers were taken from ref.⁵³ for *Arabidopsis* (nine-way multiple alignment). Left panel, composite (average intensity) plots; right panel, heatmaps of conservation distribution per DHS.

• 302. You say that the distal DHSs have conserved functional features. I assume you want to refer to enhancers instead. In addition, the word functional can only be added after the validation.
Response: We thank the reviewer for the comments. Indeed, we concluded based on our observation that predicted enhancers (defined by distal DHSs) are more conserved than promoters across species and this conservation may lead to common functional features of enhancers. We have rephrased the sentence to “*Moreover, the conservation of predicted enhancers across species indicates that enhancers may possess conserved putative functional features.*”

• 308-309. ‘dual orientated constructs (..Fig 4a)’. This wording is not entirely clear (suggests there are 2 orientations in one construct) and in addition suggests there is a schematic drawing of the dual orientated construct in Fig4a, which is not true. similarly ‘dual orientated assay’ in line 314. A GUS assay itself is not dual.

Response: We thank the reviewer for this remark. We now have changed the text to “two independent constructs with opposite enhancer fragment” and “*two independent constructs in which the enhancer fragment was either in forward or reverse orientation*” in the corresponding text and figure legend.

• 315. ‘Identical’ is overinterpretation of the data. You should say highly similar.

Response: Thank you for the comment. We have changed the wording.

• 327. The use of candidate enhancers in the 3’ regions of genes should already be mentioned at the bottom/top of page 10/11.

Response: We thank the reviewer for the suggestion. We have now mentioned this information shortly after we described the overall results of GUS assay: “*Three enhancer candidates locating at the 3’ regions of genes (including JAGGED [JAG], FLOWERING LOCUS T [FT], and a shared region by REDUCED SHOOT BRANCHING 1 [RSB1] and SOCI) were also chosen for validation (Supplementary Table 4).*”

• Fig4a. The legend mentions that the blue color intensity correlates (with) the percentage of transgenic plants with detected GUS expression. This seems to refer to the first column rather than the entire panel (and would that be independent transgenic plants rather than transgenic plants as such?). I assume that the intensity in the other columns refers to the intensity of the GUS staining?

Response: Thank you for bringing up this question. The blue color intensity illustrates the percentage of transgenic plants with detected GUS expression, not the intensity of the GUS staining. The first column shows the overall percentage of independent transgenic plants with GUS signal. The GUS signal was scored for its appearance in specific organs and displayed as

percentage of the GUS-positive plants in the adjacent columns. For more comprehensive information and in order to avoid bias caused by T-DNA location, independent T1 plants were used to define the activity of each tested enhancer candidate. Moreover, to avoid bias due to analysis of different branches GUS activity was characterized only from the main inflorescences of T1 plants. We chose to count the number of plants showing a signal and to describe the signal pattern of the GUS-positive plants. The same strategy had been applied in a previous study by Bo Zhu et al.

• 4b. The picture that supposedly shows inflorescence with early flowers is not clear. Where are the flowers? I seem to see leaves instead.

Response: Thank you for the comment. We updated the figure for the negative and positive control to make it clearer that it displays an inflorescence with a central inflorescence meristem and arising young floral buds on the flanks.

b Negative control

Positive control

Figure 4b. (b) Pictures of inflorescence with early flowers, inflorescence and mature flowers from plants harboring minimal35S as negative control (left panel) or enhancer element for 35S promoter as positive control (right panel). Red arrows indicate the central inflorescence meristems.

• Fig 4g is showing the FT gene, not the JAG gene. 4h is not FT, but JAG. FT carries an enhancer 5 kb upstream. Why is this not visible in the profile.

Response: We thank the reviewer for this remark and apologize for the mistake. We have now changed the organization of the figure and have modified the text accordingly. Thank you for the reminding regarding the 5 kb upstream enhancer. We didn't find this enhancer in our data, presumably due to the tissue we used. Indeed, we used flower tissue while the 5kb upstream

enhancer seems to be very important for the proper expression of FT gene in leaves (Jessika Adian et al., the Plant Cell 2010; Liangyu Liu et al., Frontiers In Plant Science 2014; Cao et al., the Plant Cell 2014).

- Fig 4. I miss the scale of the profiles shown (in other words, what is the size of the sequence regions looked at?). The scale is also important to know what is considered still promoter and what not.

Response: Thank you for the comments. We actually placed a snapshot picture of the physical organization of each candidate region in the genome at the bottom of each Genome Browser image. We have now added the scale (1 kb) as well.

- Figure 4b-i. There is data shown that is not discussed in the text. The DHS profiles are not mentioned. If showing the data, it should be discussed.

Response: Thank you for the suggestion. We have added the sentence of “*In addition, a fragment covered by a distal DHS peak that locates upstream of SOCI gene also showed exhibited detectable GUS signal (Fig. 4j).*” We have now mentioned the DHS profile in the figure legend.

- Figure 4b-i. The TFBSs are shown in different colors, sometimes with indication of the relevant stage, sometimes without. In the latter case, which tissue is used for those analysis?

Response: Thank you for the comments. In fact, all the ChIP-seq data used in the analysis were the reanalysis results from Chen et al. (*Nature Communications* 2018, doi: 10.1038/s41467-018-06772-3). We have now provided more detailed stage and tissue information in the figure legend.

- Figure 4b-i. Seeing the GUS data together with the DHS data (please indicate at the graphs that this is DHS data!) also makes me wonder how the GUS expression looks like in the tissues harvested for the DHS analysis. Those stages are younger (Fig 1) than the flowers shown with GUS expression in Fig 4.

Response: We thank the reviewer for the suggestion. We have indicated in the y-axis DHS data (Fig. 4j). We obtained DHSs data by using pAP1::AP1-GR *ap1 cal* inducible line (Fig. 1), which allows us to have a developmental stage-specific view of dynamics of chromatin accessibility across flower development. To validate enhancer function of predicted enhancers, we transformed wildtype Col-0 plants. Microscopic images in Figure 4b,d-j display dissected inflorescences including a central inflorescence meristem and surrounding young floral buds (first picture). Here, stage 0 (the inflorescence meristem) and floral stages 2, 4 and 8 can be seen. However, floral stages in Col-0 background are not synchronized as it is the case in pAP1::AP1-GR *ap1 cal*. We decided to not stick to the pAP1::AP1-GR *ap1 cal* inducible line because the pAP1::AP1-GR *ap1 cal* system has the advantage to provide large amounts of tissue for stage-specific genome-wide analyses, but imaging of reporter gene activity in wildtype inflorescences serves as a fully independent method to monitor stage-specific enhancer activity. However, it is a nice idea to see more stage-specific DHS via GUS reporting assays in the pAP1::AP1-GR *ap1 cal*

cal line. We would be happy to consider this in a future work dissecting the function of certain enhancers in organ specification.

- Suppl Table 4. Orientation is written incorrect. What does the yellow background color and red letters mean? I assume the blue indicates the GUS intensity? Please clarify. The nrs in the blue cells represent the % of plants of a transgenic showing Some of the nrs have too many digitals behind the comma to make sense.

Response: We thank the reviewer for the comments and suggestions. We have corrected this spelling mistake. We reformatted this table and added a color legend. We also shortened the number of digits to 2 for easy reading.

- 345-347. Can you also provide the average nr of TF binding sites?

Response: We thank the reviewer for the suggestion. There are on average 3.1 TF binding sites in enhancers, compared with 2.3 in promoters. We have added this information in the text.

- Fig5a. BS of floral regulators sounds vague and is not specified in the legend. BS also stands for bisulfite...

Response: Thank you for the comment. BS here means “binding sites”. We have specified this in the figure legend.

- 342/Fig5b. Do all 1979 genes play some role in floral development? So does it e.g. make sense that 635 promoters are bound by BLR? The legend reports enrichment scores, what do these represent? enrichment of what relative to what? The color is not indicative for the numbers, hence my question.

Response: Thank you for the comments. The 1979 genes are genes that have both proximal DHSs and distal DHSs in early stages of flower development. Not all of the genes have been functionally characterized. BLR is a general regulator in meristem development, so it is not surprising that it has lots of binding sites (BLR binds to 7341 targets which is generally more than other regulators; Chen et al. *Nature Communications* 2018, doi: 10.1038/s41467-018-06772-3) and regulate genes that are highly expressed. Enrichment scores were calculated based on p-values from hypergeometric tests assessing whether the enrichment of target genes for a specific TF in the specific category (enhancer or promoter) is statistically significant relative to the genome background (i.e., using all the annotated protein-coding genes as background). The resulting p-values were $-\log_{10}$ transformed (i.e., the enrichment score) and visualized in the heatmap.

- Fig5c & main text. How many stages are taken into account to determine if a gene is called DE? Does this refer to the 1.5 fold used before?

Response: Thank you for the comments. All the four stages were taken into consideration to define genes that are differentially expressed. As mentioned in the Methods section of “*RNA-seq*

data analysis”, “genes were considered as differentially expressed if they showed at least two-fold changes with FDR < 0.05.”

- 352. Every gene has a promoter, otherwise it cannot be transcribed. In addition please use a word like predicted in combination with enhancers. IS also true for e.g. line 374.

Response: Thank you for the comments. We have modified the text based on the comments from the reviewer in the revised version.

- 355-358. Please provide the average nr of bound regulators. The difference may be significant, but it seems the number does not double; ‘Significant more’ kind of suggests it does.

Response: Thank you for the comments. We have put the average nr on top of the plot, removed “significant” from the sentence and also re-written the whole sentence in the revised version of the manuscript.

- 358. ‘Support’ and ‘may’ are a double uncertainty. Please delete ‘may’.

Response: Thank you for the comments. We have deleted “may” in the revision.

- 364. ‘regardless’ is a bit overdone. The largest distance observed to my knowledge is 1.3 Mb.

Response: We thank the reviewer for the comment. The whole paragraph has been re-written and been moved to the earlier part of the manuscript.

- 365-366. This was also shown by Zhu before, and probably also by older, non genome-wide, papers.

Response: Thank you for the comment. The whole paragraph has been re-written and moved to the earlier part of the manuscript.

- 371-373. Wasn’t this (using the nearest neighbour strategy) already done for data shown earlier in the manuscript? For example, for the transgenics I also assume you already determined what would be the nearest neighbor.

Response: Thank you for the comment. Indeed, we agree with the reviewer that it would be better to mention that we apply the “nearest neighbor strategy” in the beginning (see page 10 in the revision).

- 376. ‘dominantly grouped’?? What do you mean?

Response: Thank you for the comment. By ‘dominantly grouped’, we meant that there are two main groups but each group contains subgroups. Since we already said two “distinct” clusters, we have now deleted “dominantly” in the revision.

- Suppl14 legend. RPM would represent enhancer activity in reads? I do not know such measurement. Do you mean DNaseI sensitivity? Clarify. Also true for legend Fig 7e.

Response: Thank you for the comments. Yes, RPM (reads per million mapped reads) values represent the enhancer activity here, and they were calculated based on DNase-seq data. We have clarified this in the figure legend of Fig 7 and Suppl Fig. 14.

- Fig6, SupplFig8. What is the reasoning to do similar analysis on four groups and later on 2 groups? Why not focus on the latter analysis and delete the first from the manuscript? Which conclusions are drawn based on the first analysis that cannot be drawn based on the latter?

Response: With SuppFig 8, we investigated the correlation between H3K27ac dynamics and changes in gene expression while Fig 6 showed the association of enhancer (predicted by DHS) dynamics and gene activities. We think that they are two independent analyses to interpret the dynamics of gene expression across flower development, so we would like to keep both.

- Fig6c. How come that the numbers (266 & 329) do not add up to 810? Where are there enhancers/promoters missing?

Response: We thank the reviewer for the comment. In fact, genes with missing expression data (“NA” values) were removed from our analysis. We have now mentioned this in the figure legend.

- Line 385-386 do not connect to 386-389. It is unclear from the text why the latter exemplifies the first.

Response: Thank you for the comments. We have now placed the sentences on line 386-389 after line 379.

- 389-390. Sentence does not flow. N total there are three, so something like ‘and another distal...’

Response: Thank you for the comments. We have removed the sentence here.

Fig 6d. What do the horizontal red bars below the plot refer to? I assume a potential enhancer (the upstream one is however lacking the bar). The main text seems to ignore one of the two element in the intron (bound by Sep3, AG etc). In addition, delete ‘novel’ in the legend and text. You do not show this enhancer just arose in a particular accession or something like that. The TFBS data is not discussed in the text. IF you show it, better also say something about it.

Response: Thank you for the comments. Yes, the bars below the signal profiles represent the called DHS peaks at each stage. The upstream enhancer has the bar at stage 2 or stage 4, but not at stage 8. Since the enhancers from the 2nd intron of AG has been well studied (e.g., Deyholos et al, 2000) and in order to focus on more novel findings, we removed the Fig 6e in the revision.

- 401-402, Fig 6f. Where does the chromatin accessibility come in at this figure? What does actually AP1, 2h refer to? The legend should make clear that the columns represent stage 2-

specific enhancers and stage 8-specific enhancers. Now it takes a while before one understands the figure.

Response: We thank the reviewer for the comments. We apologize that this figure is not intuitive enough to understand. As suggested by the reviewer, we have added more information to the figure legend to indicate that the columns are stage-specific enhancers (as determined by chromatin accessibility data) while the rows represent TF binding sites (determined by ChIP-seq data at different developmental stages). “AP1, 2h” means the binding pattern of AP1 at 2 hours after induction. In the revised figure legend, we also added more detailed information about the data used in the analysis. Please note that the old Fig. 1f becomes Fig. 1e in the revision.

- 408. Why ‘in more general terms’? LBD is also general.

Response: Thank you for the comment. We intended to describe a general overview of the putative binding events. We admit that “in more general terms” may not be the proper wording and have deleted this phrase in the revision.

- Suppl table 5. Target genes should read predicted target genes. Page Motif analysis: the nrs in the fold change column do not seem to be right. E.g. 10.6/0.6 is not 0.048 (top row). In the enhancer dynamics table the nrs in the log2 fold change column also do not seem to make sense. The explanation for the last page ‘the number of bound floral regulators in each enhancer’ does not fit the content of the page.

Response: Thank you for the comments. We have changed the “Target genes” to “Predicted target genes”. We moved the column of “fold change” to the right on the page “Motif analysis”. The calculation of fold-change of motif enrichment is right (Stage 2 / Stage 8). Values in the enhancer dynamics table are log2-transformed. We have added more explanations in the table header. We also updated the explanation for the last page. We hope these changes make this part clearer now.

- 422. Delete ‘distal’. That is commonly used for intergenic regulatory elements, not for intragenic.

Response: We agree with the reviewer and have deleted “distal” in the revision.

- 426 and elsewhere should read predicted intronic enhancers (or something like that).

Response: Thank you for the comments. We have changed the text accordingly

- 429. The text says 158 genes, the table lists 181 GO terms. How does 158 & 181 relate to each other? And which gene is connected to which row?

Response: We thank the reviewer for the comment. We now added an additional column to indicate a gene list for a specific GO term.

• Fig7. In a there are 724 intronic enhancers, in c there are 700. What happened with the 24 that are missing in c? Similar Q for intergenic enhancers. legend for b should read predicted target genes. Why is ‘or 3’UTRs’ between brackets?

Response: We appreciate the reviewer for the comments. 724 is the number of intronic enhancers and 700 is the number of their target genes. Some genes may have multiple intronic enhancers. We have made this clear in the figure legend. We have changed the legend by adding “predicted” and by changing “intronic (or 3’UTRs)” into “intragenic”.

• Fig7e. The numbers in between brackets for the top and bottom plot differ. Are the nrs in () for the bottom panel actually also the nr of genes in each category? How is the enhancer dynamics measured in case of more than 1 enhancer? Do we look at the average ‘activity’ of e.g. 4 enhancer? Something else? Enhancer activity should be defined as well.

Response: The number in the bottom panel is the number of enhancers in each category while the number in the top panel is the number of predicted target genes in each category. We have improved the figure legend for this. Similar to the definition of gene expression changes, the enhancer dynamics was calculated as the change of chromatin accessibility (based on DNase-seq data) over the four developmental stages.

• Fig7f. How do the data translate in nr of TFBSs? Are these nrs similar as reported earlier in the manuscript for intergenic enhancers and promoters? How come that here there are 4844 intergenic enhancers again, and 724 intronic ones? (see point above as well). Which collection TFBS is looked at for this analysis? The text reads developmental master TFs. How many are these and how are these defined? Are these the same as used for Fig5? There is not a reference in the text referring to a specific paper.

Response: We thank the reviewer for the comments. In Fig. 7f, the number of TF binding sites for a catalogue of 15 developmental master regulators were counted in the regulatory elements under each category. Yes, the numbers for intergenic enhancers and promoters are the same as reported earlier in the manuscript. The numbers in Fig. 7f are the number of regulatory elements, while the numbers in Fig. 7b are the number of their predicted target genes (one gene may have multiple regulatory elements). The collection of TFBSs were obtained from our recent study (Chen et al. *Nature Communications* 2018, doi: 10.1038/s41467-018-06772-3) where a catalog of 15 master TFs were investigated. The reviewer is right that the same data was used for the analysis of Fig 5. We have added a reference in the text as well in the Methods part. Please note that the old Fig. 7f has been renamed as Fig. 7b in the revision.

• 443. You suggest that Fig7f shows that genes associated with multiple enhancers have a significant enrichment in binding sites for TFs. Fig7f however does not show the enrichment at enhancer elements that are part of a cluster vs single enhancer elements.

Response: Thank you for this very insightful comment. We re-organized the Fig 7 where the old Fig 7f is now directly after Fig. 7b since these two figures are more relevant. We changed the

corresponding sentence as “*Interestingly, we found that these predicted intronic enhancers (like predicted intergenic enhancers) were associated with significantly higher target gene expression dynamics than DHSs in 3’ UTR regions (Fig. 7b) and with significant enrichment in binding sites for a catalog of 15 developmental master TFs (Fig. 7c). This suggests that the predicted intronic enhancers are important regulatory elements for the control of developmental transcriptome dynamics.*”

- Fig 7g. Different putative enhancers are indicated and numbered. Not all peaks having a bar are indicated/numbered. E.g. the one next to nr1. Please clarify. Please provide the scale to allow better interpretation of the figure.

Response: We thank the reviewer for pointing this out. We admit that the one next to nr1 is indeed a true element for this enhancer cluster, but nr4 is closest to the next gene (*RAD51B*) according to our analyses. We apologize for this mistake. However, it still makes sense to assign nr4 to one enhancer element for the *TOEI* gene since nrs2-4 are very near to each other. We thus propose that *TOEI* has an enhancer cluster with five enhancer elements by manual checking. We renumbered the elements in the new figure. We have also added a scale in the figure.

- 449. ‘stretched enhancer elements’? Personally I never heard this term. Super enhancer is much more commonly used.

Response: Stretched enhancer elements were firstly described by Stephen C. J. Parker et al. (doi: 10.1073/pnas.1317023110) to describe a regions of more than 3 kb in length and containing contiguous segments marked as enhancer states. Later these enhancer clusters were also called super enhancers additionally featured by binding sites for master transcription factors. We agree with the reviewer that the term of “super enhancer” is becoming more commonly used, and changed the wording as suggested.

- 449-451. What do you mean with similar arrays? Similar numbers? Compared to single enhancers? I assume you mean something else given you refer to a feature of super enhancers. Please clarify the text.

Response: Thank you for your comment. Here, “similar arrays of transcription factors” means the protein groups binding to these regions are composed of the similar TFs. In order to avoid misunderstanding, we would like to change the term to “similar sets of transcription factors.”

- 461-63. This is the choice made by which the enhancer candidates are identified (and includes accessibility (DHSs) as well). This was supported by finding know/putative enhancers back, but it has not been tested e.g. if DHSs without H3K9ac would be able to enhance expression. It would be appropriate to also e.g. cite Zhu et al. and Qiu et al.

Response: Thank you for the comments and suggestions. Following the suggestions of the reviewers, we have re-written this part as “*Only recently, enhancers were mapped in plant*

genomes and the results revealed that those enhancer candidates correspond to accessible distal DNA fragments”.

- 468 & 473-475. I would delete ‘compact genome’. I do not see any reason why a distal enhancer could not be marked by H3K27ac when the genome is more compact. Rice and maize are by the way monocots. I have no clue if there has been divergence in using this mark between monocots and dicots.

Response: We thank the reviewer for the comments. We agree with the reviewer that compactness of the genome is not the reason why a distal enhancer can’t be marked by H3K27ac. We actually pointed this out previously by line 473-475 “*indicating that H3K27ac does not mark enhancers in Arabidopsis thaliana. However, this cannot be explained as a simple consequence of genome compactness, since e.g. the Drosophila genome is also compact, but H3K27ac marks its enhancers*”. In order to make it clearer, we have deleted “*with a less compact genome than Arabidopsis thaliana*” in the revised version of the manuscript. Regarding the comments on “*divergence in using this mark between monocots and dicots*”, we had discussed this possibility a bit in the previous version of the manuscript (see line 485-490) but essentially, this question needs to be addressed in future research.

- 472. The 12.7% is this coming from your study? Or should there be ref added?

Response: Thank you for the comment. It is the data from rice but was re-analyzed by us. We have made this clear in the revised version.

- 480. Han et al did not test any connection with H3K27ac, so you can’t conclude the effect is not via H3K27ac. Mutants in HATs have in addition pleiotropic effects so promoting flowering may be an indirect effect.

Response: Thank you for the comments and sorry for the over-interpretation of the message delivered by the paper. After checking more related work (He et al., 2003; Sung and Amasino, 2004), we figured out that Han checked both H4 and H3 (H3K9ac and H3K14ac) at the FLC locus and found there is no change of the modifications in the mutant. So, we have changed the sentence in the revision.

- 487-490. I miss a reference for the data showing this is opposite in Arabidopsis.

Response: We have added the reference.

- 486-491. It is not clear why the focus is on interactions/co-occupancy with other chromatin marks, rather than on divergence in the specific marks used to activate enhancer sequences. Acetylation makes chromatin more accessible, allowing access to the ‘transcription machinery’. Maybe the chromatin at Arabidopsis enhancer sequences does not ‘require’ acetylation to open up chromatin; maybe the binding of a pioneering TF is sufficient to recruit chromatin

remodelling proteins that in turn allow access to other proteins; or there is an acetylation mark added that is not identified yet.

Response: Thank you for the very interesting comments and ideas. The reviewer raises an interesting point here. Based on other studies, H3K56ac correlates with H3K4me2, H3K36me3 and H3K4me3 (Roudier et al, EMBO J 2011) and not with active transcription. Our analyses for co-occupancy and relative enrichment (Fig.2) show that these three marks are even less enriched at distal DHS than H3K27ac; as stated by Lauria & Rossi (2011) H3K56ac seems to be a mark of transcriptional competence (Lauria & Rossi, 2011). As of H3K14ac, the chromatin state to which it belongs differentiates from the rest as a different combination of open chromatin marks containing H2A.Z, H3K9ac and H3K4me3 (Sequeira-Mendes et al, 2014). Once again, these three marks do not show better correlation with DHS occupancy than H3K27ac. Moreover, else than acetylation marks, we did not find other chromatin marks for which ChIPseq data is available in Arabidopsis, that co-occupy/interact with DHSs better than H3K27ac (Fig 2).

We do agree that an explanation could be that the chromatin at arabidopsis enhancers does not require acetylation to be opened up; and that TF binding is sufficient to recruit chromatin remodeling proteins that in turn allow access to other proteins; however, one cannot exclude that another, yet unidentified, acetylation (or other) mark may contribute to chromatin accessibility. We could of course mention all these possibilities in the manuscript but believe that one would need more understanding on how exactly pioneering factors are recruited to DNA, or discovery of potential novel chromatin marks, to discuss these possibilities in a wiser manner.

- H3K4me1 is associated with both inactive and active enhancers in animals.

Response: Thank you for the comment. We agree that H3K4me1 can be also associated with inactive enhancers in animals and that is why we used the word “generally”. However, the main information by the sentence is that even though H3K4me1 could mark active enhancers, we found little co-localization between H3K4me1 and distal DHSs, which indicates H3K4me1 at least doesn't mark enhancer in Arabidopsis. To avoid misunderstanding, we have corrected the sentence to “*In animal systems, H3K27me3 correlates with poised enhancers, whereas H3K4me1 mark could correlate with poised and active enhancers*”.

- 494-496. You elaborate on the intergenic localization of H3K4me3. How many of your predicted enhancers carry H3K4me3? In other words are there also DHSs detected at poised enhancers?

Response: We assume that the reviewer intended to ask about H3K27me3 rather than H3K4me3 since we didn't mention H3K4me3 here. Anyway, the reviewer raised a very nice point, so we have performed analyses for both H3K4me3 and H3K27me3 and found that less than 1% intergenic DHSs carry H3K4me3 and 9.0% distal DHSs contain H3K27me3 but 31% of those H3K27me3 marked distal DHSs are actively changing across flower development. We have added this information in the revised manuscript.

- Ref 78 lacks page numbering.

Response: Thank you for pointing this out. We have added page number for this reference.

- 497-499. To study species-specific characteristics multiple species need to be studied. Not just Arabidopsis.

Response: Thank you for the comment. We have deleted “species-specific” in the revision.

- 515-517. The first sentence is a general statement; in such case this will also be true for other TFs than TFs regulating flowering.

Response: Thank you for the comment. We have changed the statement into “*These results highlight the importance of enhancers for re-shaping developmental transcriptome dynamics during flower development, preferentially via regulation of floral master regulators.*” Now, we focus on flower development process, for which all the data were generated and analyzed.

- 516-517. Are the master regulators a subset of the flowering genes? What is the criterium to be called a master regulator versus a regulator?

Response: Thank you for the comments. Yes, the master regulators mentioned here are also flowering genes. The reasons to call them “master regulators” lies in their characteristics that line up with the definition: they are “Transcription factors that trigger major developmental decisions in plants and animals. Such master regulators are classically seen as acting on the top of a regulatory hierarchy that determines a complete developmental program, and they usually encode transcription factors.” (Kerstin Kaufmann and Chiara A. Airoidi, 2018)

- 518. Formulation needs to be improved. Activity cannot be enriched by binding...

Response: Thank you for the suggestion. We have changed “binding” to “binding events” in the revised version of the manuscript.

- 530-534. In figure 7 you show DHSs are also located downstream of the TSS, intronic, and also in the 3’UTR.

Response: Thank you for the comments. By the sentences from line 530 to 534, we intended to talk about intergenic enhancer and discuss other intragenic enhancers in later parts of the manuscript. In order to be more focused on enhancer clusters and intronic enhancers, we have removed this paragraph from the manuscript.

- 537. ‘So’ should be ‘and’.

Response: Thank you for pointing this out. We have corrected the text accordingly.

- 540. The use of ‘Thus, they should’ is not appropriate overhere. The previous sentence does not say anything about both activation and repression.

Response: Thank you for pointing this out. We have changed the sentence to “*Intronic regulatory regions have been reported to be essential for determining the activation of several important developmental genes in distinct spatiotemporal patterns (e.g., see ref.84–86), which indicates that these intronic regulatory regions should contain repressive and enhancing elements, as was shown for the AG 2nd intron.*”.

- 541. Similarly, ‘in line with this’ is not appropriate.

Response: We have changed it to “*In addition*” in the revision.

- 543-544 Last part of the sentence is a bit confusing. I would change it to ‘...gene expression through eRNA transcription.’

Response: Thank you for the suggestion. We have made the change based on the suggestion of the reviewer.

- 551. I am not sure this is still under debate 3 years from ref 88.

Response: Thank you for the comment. Whether super-enhancer is just computationally defined or actually has biological function is under debate (Wong et al., *Molecules* 2018; doi: 10.3390/molecules23051057). To be more precise, we have changed the sentence as “*In animal systems, the functional relevance of ‘super enhancers’ is still under debate*”

- 553. Not clear to what ‘location’ refers to.

Response: We meant a cluster of closely-located enhancers. Sorry for the improper wording. We have changed to “*closely-located enhancer clusters*” in the revision.

- 554-555. You did not show that individual enhancers in a cluster bind more TFs than individual enhancers not being part of a cluster. You do however suggest overhere that you did.

Response: Thank you for the comment. We indeed have no evidence to show that individual enhancers in a cluster bind more TFs than individual enhancers not being part of a cluster (there is no significant difference between them based on our analysis). We have changed the sentence as “*Indeed, our identified enhancer clusters typically serve as binding platforms of developmental master TFs, and different individual enhancer within the cluster shows similar sets of TF binding sites*”

- 558-559. Why not also for 3’UTR enhancers or single intergenic enhancers?

Response: Thank you for the comment. Yes, it is interesting to dissect the function of any type of enhancers, for which not much is known in plants. To make this clearer, we have changed the sentences as “*Further research should aim to dissect the functions of individual enhancers or individual cis-regulatory components within enhancer clusters in plants, for example by making use of Cas9-based mutagenesis.*”

• 561-562. Accessibility is part of the activation process and it is triggered by binding of TFs and allows the binding of subsequent protein factors.

Response: Thank for the comments. We have now changed the sentences to *“In addition to the widely accepted concept that gene transcription is triggered by binding of pioneer TFs that open the chromatin of core-promoters, thereby allowing the binding of subsequent regulators, our work uncovered another regulation layer of developmental stage-specific gene transcription by the dynamic activities of enhancers in plants.”*

• 562-564. This has been reported by others before. The message is true focusing on

Response: Thank you for the comment. To the best of our knowledge, although similar work had been done in animals, this is the first report about the role of enhancer dynamics in gene expression control over a developmental course in plants. We assume that the missing word in the reviewer comment is “plants”. We have modified the following sentence accordingly: *“our work uncovered another regulation layer of developmental stage-specific gene transcription by the dynamic activities of enhancers in plants”*

Reviewer #2 (Remarks to the Author):

Authors have revised or deleted most of the text related to the issues we talked about. We might have preferred that they integrate more of the responses they gave in the rebuttal into the actual manuscript. For example, more details comparing and contrasting Zhu et al enhancer prediction vs the current study, and seedling vs flower tissue comparisons. Otherwise, the rest of the revision looks fine.

Response: We thank the reviewer for the comments. We have incorporated some of the points from our previous revision to the main text including the ones pointed out by the reviewer. For instance, we have discussed 1) why the genes with both H3K27ac and proximal DHSs exhibited higher expression; 2) the difference regarding enhancer number and H3K27ac patterns between the two studies; 3) why the results of seedling tissue could be applied to flowers; 4) what needs to be done to prove the conclusion obtained from motif scan.

Reviewer #3 (Remarks to the Author):

In the revised manuscript, the authors newly and nicely performed functional study of 22 candidate enhancers, and extended the bioinformatics analyses. Although they claimed that they have added experimental validation results for predicted enhancers, I still have one concern on the result of AP1 (Fig 3a and f). The AP1 enhancer lines with a 35S minimal promoter showed GUS expression in “joints” and sepals, but not in petals. Further, the majority of lines showed expression in stamens and carpels, which should have no AP1 expression. These results do not

support an idea that the predicted AP1 enhancer is a real one. They obtained only 4 lines and their results may not reflect the real activity. This problem should be resolved.

Response: We thank the reviewer for the insightful comments. In order to address the concern of the reviewer regarding the *API* enhancer, we identified another 24 independent transgenic GUS reporter T1 lines for the *API* enhancer candidate following the suggestion of the reviewer. These 24 new lines showed a similar GUS staining pattern in flower tissue to the signal observed in the four transgenic lines and largely confirm the previous results. Still, we observed very weak signal in the vasculature of mature petal but not in petals at early stages. Again we saw clear signal in sepal, stamens and carpels. All these data revealed that the fragment we tested has enhancer activity although the GUS pattern partly deviates from its neighboring downstream gene, *API*, which should be highly expressed in petals. The reason for this deviation could be that some other regulatory elements for *API* expression are missing, especially the core-promoter since a gene expression pattern is determined by the combinatory action of multiple regulatory elements, including core-promoter and other elements.

As stated in our answer to Reviewer #5, we have added one figure to show the correlation between GUS pattern of tested enhancer and expression profile of neighbor gene (Fig 4c). We have also discussed the results accordingly. Nevertheless, we discovered in our work that the dynamic activity of predicted enhancers highly correlates with the expression changes of its neighboring gene.

Reviewer #5 (Remarks to the Author):

The authors did - for the most part- address my concerns. The enhancer tests in particular add to the study. There are two points in regards to this experiment that need addressing. Firstly, the size of the enhancer fragments tested in vivo should be included in the main text. Secondly, the authors should state in the main text how the expression pattern observed deviates from the endogenous expression pattern of the genes for each enhancer tested. It does deviate quite a bit and there are good reasons for this (additional regulatory information is missing), but the reader should be made aware of the partial recapitulation of the endogenous pattern.

Response: We appreciate the suggestions from the reviewer. Following the suggestion of the reviewer, we have included information on the length of enhancer elements we selected for tests in both the main text and the **Supplementary Table 4**. In general, the region covered by the corresponding distal DHS peaks was isolated and the length of the fragments varies from 0.52 kb to 1.77 kb, with a median value of 1.07 kb. Indeed, we saw quite a bit of deviation between the GUS pattern of tested enhancer and the endogenous gene expression of its putative target. We totally agree with the reviewer that this deviation is largely due to other missing regulatory elements, especially in the core-promoter. We would like to also point out that although we found a tight link between dynamics of enhancer activity and changes in target gene expression, the determinant role of core-promoter on gene expression pattern should not be ignored. We

have added one figure to show the correlation between GUS pattern of tested enhancer and expression profiles of neighboring gene (see figure below; as well as **Figure 4c** in the revision). We have also discussed the results accordingly in the manuscript.

Figure 4c. (c) Pearson correlation (r) between GUS signals and expression profiles of the corresponding genes in the tested tissues. Tissue-specific gene expression data can be found in Supplementary Table 4. Data points were colored according to the GUS signals in (a).

REVIEWERS' COMMENTS:

Reviewer #3 (Remarks to the Author):

The authors addressed my concerns well.